# miRNA142-3p targets Tet2 and impairs Treg differentiation and stability in models of type 1 diabetes

Martin G. Scherm[1,2], Isabelle Serr[1,2], Adam M. Zahm[3], Jonathan Schug[3], Saverio Bellusci[4], Rossella Manfredini [5], Victoria K. Salb[1,2], Katharina Gerlach[6], Benno Weigmann [6], Anette-Gabriele Ziegler[7,8], Klaus H. Kaestner[3] & Carolin Daniel[1,2,9]*

In type 1 diabetes, the appearance of islet autoantibodies indicates the onset of islet auto-immunity, often many years before clinical symptoms arise. While T cells play a major role in the destruction of pancreatic beta cells, molecular underpinnings promoting aberrant T cell activation remain poorly understood. Here, we show that during islet autoimmunity an miR142-3p/Tet2/Foxp3 axis interferes with the efficient induction of regulatory T (Treg) cells, resulting in impaired Treg stability in mouse and human. Specifically, we demonstrate that miR142-3p is induced in islet autoimmunity and that its inhibition enhances Treg induction and stability, leading to reduced islet autoimmunity in non-obese diabetic mice. Using various cellular and molecular approaches we identify Tet2 as a direct target of miR142-3p, thereby linking high miR142-3p levels to epigenetic remodeling in Tregs. These findings offer a mechanistic model where during islet autoimmunity miR142-3p/Tet2-mediated Treg instability contributes to autoimmune activation and progression.

[1] Institute of Diabetes Research, Group Immune Tolerance in Type 1 Diabetes, Helmholtz Diabetes Center at Helmholtz Zentrum München, 80939 Munich, Germany. [2] Deutsches Zentrum für Diabetesforschung (DZD), 85764 Munich-Neuherberg, Germany. [3] Department of Genetics and Institute for Diabetes, Obesity and Metabolism, Perelman School of Medicine, University of Pennsylvania, Philadelphia, PA 19104, USA. [4] German Center for Lung Research, Excellence Cluster Cardio-Pulmonary System, Universities of Giessen and Marburg Lung Center, 35390 Giessen, Germany. [5] Center for Regenerative Medicine, University of Modena and Reggio Emilia, 41125 Modena, Italy. [6] Department of Medicine 1, University of Erlangen-Nuremberg, 91052 Erlangen, Germany. [7] Institute of Diabetes Research, Helmholtz Zentrum München, German Research Center for Environmental Health, 85764 Munich-Neuherberg, Germany. [8] Forschergruppe Diabetes, Technical University Munich, at Klinikum rechts der Isar, 80333 Munich, Germany. [9] Division of Clinical Pharmacology, Department of Medicine IV, Ludwig-Maximilians-Universität München, 80337 Munich, Germany. *email: carolin.daniel@helmholtz-muenchen.de

Type 1 diabetes (T1D) results from a breakdown of immunological self-tolerance to the insulin-producing islet beta cells, and consequently their destruction by auto-reactive T cells[1]. The incidence of T1D is increasing worldwide, especially in young children[2]. Longitudinal studies of individuals at risk for developing T1D show that the disease progresses through distinct identifiable stages prior to the onset of clinical symptoms[3,4]. The appearance of multiple islet autoantibodies indicates the onset of islet autoimmunity (pre-T1D), which can occur years before clinical symptoms arise[3]. Despite ongoing research efforts, the molecular mechanisms underlying the onset of islet autoimmunity and progression to clinical diabetes remain poorly understood.

Recent analyses have focused on the specific roles of different T cell subsets during the presymptomatic stage of T1D and their contribution to immune activation and autoimmunity[5,6]. High frequencies of insulin-specific Tregs were associated with a slow progression from islet autoimmunity to clinical T1D in children, suggesting a crucial role for regulatory T cells (Tregs) in delaying or possibly preventing the progression of islet autoimmunity[7]. Tregs are key players for the maintenance of peripheral immune tolerance, and defects in Treg induction and function are important contributors to autoimmune disorders like T1D[8–10]. Tregs are characterized by the expression of CD4, CD25, and the transcription factor Foxp3, which is required for their development and function[11,12]. The efficient induction of Foxp3+ Tregs from naive CD4+ T cells can be achieved by the application of a strong-agonistic T cell receptor (TCR) ligand under sub-immunogenic conditions[10,13–15]. Mutations in the FOXP3 gene have deleterious consequences, leading to autoimmune phenotypes in both mice (scurfy mice) and humans (IPEX—immuno-dysregulation, polyendocrinopathy, enteropathy, X-linked syndrome), highlighting the crucial role of Foxp3 for Treg function[16,17].

Epigenetic mechanisms such as altered DNA methylation patterns are a critical factor in the pathogenesis of several auto-immune diseases[18–20]. The Foxp3 gene itself is subject to changes in DNA methylation, controlling gene activity by altering the accessibility of the DNA to transcription factors[21,22]. The hypomethylated state of four "conserved noncoding sequences" (CNS) within the Foxp3 locus ensures proper Foxp3 expression in Tregs[23–25]. In particular, the CNS2 is a critical regulator of long-term stability of Foxp3 expression, and consequently the Treg phenotype: The CNS2 element is completely demethylated in Tregs but fully methylated in conventional T cells and in vitro-induced Tregs[23,25–27].

The establishment of hypomethylated regions is dependent on three members of the "ten eleven translocation" (Tet) family, Tet1, Tet2, and Tet3[28,29]. These enzymes are capable of oxidizing 5-methylcytosine (5mC) to 5-hydroxymethylcytosine (5hmC), which is an intermediate of DNA demethylation[30,31]. The Tet genes are critical for the differentiation of CD4+ T cells in mice[32] and humans[33], as well as Treg homeostasis and function[34–36]. Despite these insights, the molecular mechanisms that can regulate Tet gene expression in CD4+ T cells remain incompletely understood. Moreover, it is unknown whether aberrant Tet activity can impair Treg homeostasis during islet autoimmunity.

MicroRNAs (miRNAs) critically contribute to immune function and homeostasis[5,6,37–40]. Although these studies provide considerable insight into the role of miRNAs in immune homeostasis, their direct targets and affected signaling pathways remain poorly understood, especially in human T cells. In particular, a direct link between miRNA dysregulation and impaired Treg induction in the context of the onset of autoimmunity has not been reported yet.

Here, we identify a miRNA/Tet2 axis as a direct component of Treg regulation. We propose that aberrant miR142-3p expression in CD4+ T cells acting via Tet2 repression functions as one mechanism by which dysregulated DNA methylation at the Foxp3 locus mediates impaired Treg homeostasis, and consequently contributes to autoimmune activation.

## Results

**miR142-3p is highly abundant in RISC of human CD4+ T cells.** While profiles of total miRNA abundance in T cells have been reported previously[41], none have determined which miRNAs are actively engaged in mRNA regulation, and which mRNAs are specifically targeted. Therefore, we performed high-throughput sequencing of RNA isolated by crosslinking immunoprecipitation (HITS-CLIP) analysis of miRNAs and mRNA fragments present in the RNA-induced silencing complex (RISC) of human CD4+ T cells, following immunoprecipitation with an antibody against Argonaute 2 (Fig. 1a). Mapping of the sequencing reads to the human genome identified 271 unique miRNAs as present within the RISC in human CD4+ T cells and 7829 mRNA targets. The analysis of our sequencing libraries showed that miRNA binding occurs at comparable levels at the 3′ UTR and the coding sequence of the mRNA target (Fig. 1b), with only a slight preference for the 3′ UTR (Fig. 1c). This is in contrast to earlier findings, suggesting that the binding happens preferentially at the 3′ UTR[42].

The ten most abundant miRNAs in the RISC of human CD4+ T cells are shown in Fig. 1d. The abundance of individual miRNAs in the RISC varied greatly and miR142-3p was the most abundant active miRNA. Gene ontology analysis of the 500 most targeted mRNAs showed a significant enrichment of biological processes associated with immune activation, such as "T cell signaling" and "T cell activation" (Fig. 1e).

**Increased miR142-3p levels in islet autoimmunity and T1D.** A critical question is whether the abundance of individual miRNAs is linked to the activation of islet autoimmunity. Therefore, in a further set of experiments we isolated the miRNA fraction from activated CD4+ T cells (CD4+CD3+CD45RA−CD45RO+CD127+CD25intermediate; Supplementary Fig. 1a) of children with and without islet autoimmunity to screen for differentially expressed miRNAs by high-throughput sequencing. We identified multiple differentially expressed miRNAs, with both down- and upregulation of up to tenfold (Supplementary Fig. 2). One of the miRNAs specifically upregulated in activated CD4+ T cells from children with ongoing islet autoimmunity was miR142-3p, with a fold change of about 2 (Fig. 1f). The differential expression of miR142-3p was validated by quantitative polymerase chain reaction (qPCR) analysis of activated CD4+ T cells isolated from peripheral blood of individuals without T1D and with recent onset of T1D (Fig. 1g). Interestingly, we found that activated T cells (CD4+CD25−CD44high; Supplementary Fig. 1b) in non-obese diabetic (NOD) mice, a well-established model of T1D[43,44], with recent development of insulin autoantibodies (IAA) also showed elevated levels of miR142-3p expression compared to IAA− littermates (Fig. 1h). Furthermore, we analyzed an existing mRNA sequencing dataset of individuals with ongoing islet autoimmunity and healthy controls[6] for differential expression of predicted miR142-3p targets[45]. The majority of the predicted miR142-3p targets was downregulated in CD4+ T cells of individuals with ongoing islet autoimmunity (Supplementary Fig. 3), supporting the important role of this miRNA for the activation of T-cell-specific islet autoimmunity.

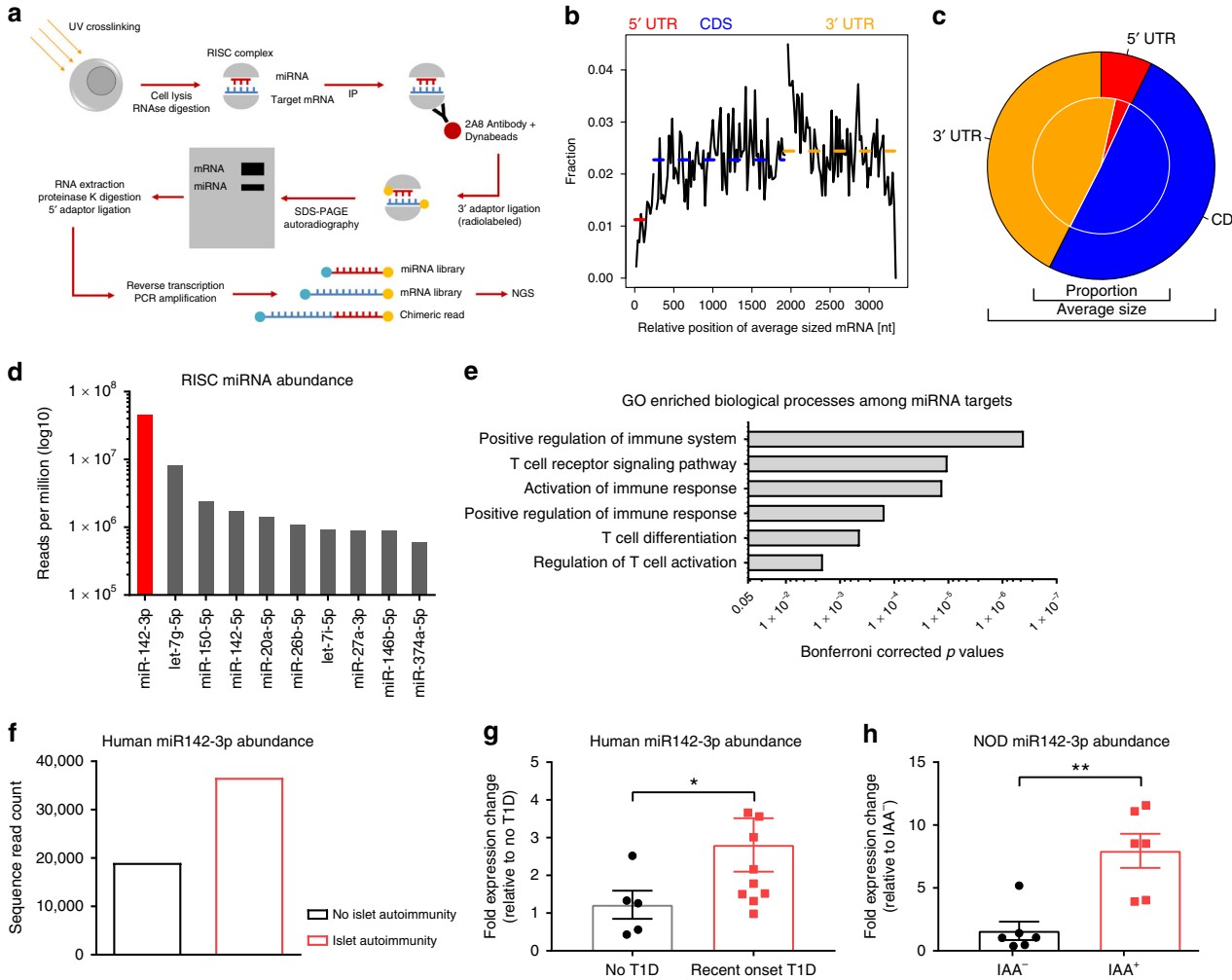

**Fig. 1 miR142-3p is highly abundant in CD4$^+$ T cells and upregulated in islet autoimmunity. a** Schematic illustration of the HITS-CLIP technique.
**b** Average read coverage of RISC-associated mRNA fragments over a standardized mRNA as revealed by HITS CLIP. Dashed lines show the average levels. CDS, coding sequence. **c** The outer pie shows the average size of the regions. The inner pie indicates the proportion of RISC-associated mRNA fragments found in each section. **d** The ten most abundant RISC associated miRNAs in human CD4$^+$ T cells, as revealed by HITS-CLIP. **e** Significantly enriched gene ontology (GO) biological processes in target genes of RISC associated miRNAs in human CD4$^+$ T cells, as revealed by HITS CLIP. **f** Levels of miR142-3p in activated CD4$^+$ T cells isolated from peripheral blood of children with and without islet autoimmunity, as revealed by miRNA sequencing of four pooled samples per group. **g** Levels of miR142-3p in activated CD4$^+$ T cells isolated from peripheral blood of children without T1D and with recent onset of T1D, as revealed by qPCR normalized to 5s rRNA, $n \geq 5$. **h** Levels of miR142-3p in activated CD4$^+$ T cells isolated from lymph nodes of NOD mice with and without islet autoimmunity, as revealed by qPCR normalized to 5s rRNA. $n = 6$. One data point represents one subject. Experiments were performed in three technical replicates per subject. Data are presented as means ± s.e.m., Student's $t$-test, *$P < 0.05$, and **$P < 0.01$. The source data are provided as a Source Data file.

**Inhibition of miR142-3p improves Treg induction in vitro.** Having established that miR142-3p is induced in CD4$^+$ T cells during onset of islet autoimmunity in both mice and human, we next investigated the potential specific contribution of this miRNA to autoimmune activation. We recently showed that insulin-specific Tregs are significantly less abundant in children with recent onset of islet autoimmunity[7]. We wondered if the differentially expressed miR142-3p might interfere with Treg induction from naive T cells, thereby promoting autoimmune activation and progression. To this end, we performed in vitro Treg induction assays using subimmunogenic stimulation of naive CD4$^+$ T cells (CD4$^+$CD25$^-$CD44$^-$; Supplementary Fig. 4a)[7,46] from NOD mice, with recent onset of islet autoimmunity. Treg induction capacity of naive CD4$^+$ T cells from NOD mice significantly decreased with the onset of IAA$^+$ autoimmunity (Fig. 2a). Next, we performed murine in vitro Treg induction

experiments in the presence of miR142-3p mimics or miR142-3p inhibitors (a representative flow cytometry staining is shown in Fig. 2b and Supplementary Fig. 5a). Increasing miR142-3p activity in the presence of a mimic significantly reduced Treg induction efficacy using naive CD4$^+$ T cells from non-autoimmune prone BALB/c mice (Fig. 2c), resembling the reduced Treg induction efficacy in IAA$^+$ NOD mice. Concordantly, blocking miR142-3p activity with a highly potent inhibitor significantly increased Treg induction (Fig. 2c).

The high efficacy and specificity of the miRNA inhibitor was demonstrated in several independent experimental settings. First, the inhibitor reduced miR142-3p abundance by >99% after 3 h of incubation (Supplementary Fig. 6a). Second, application of the miRNA inhibitor significantly derepressed luciferase activity of a miR142-3p activity sensor plasmid in Jurkat T cells (Supplementary Fig. 6b). Third, the inhibitor increased mRNA abundance of

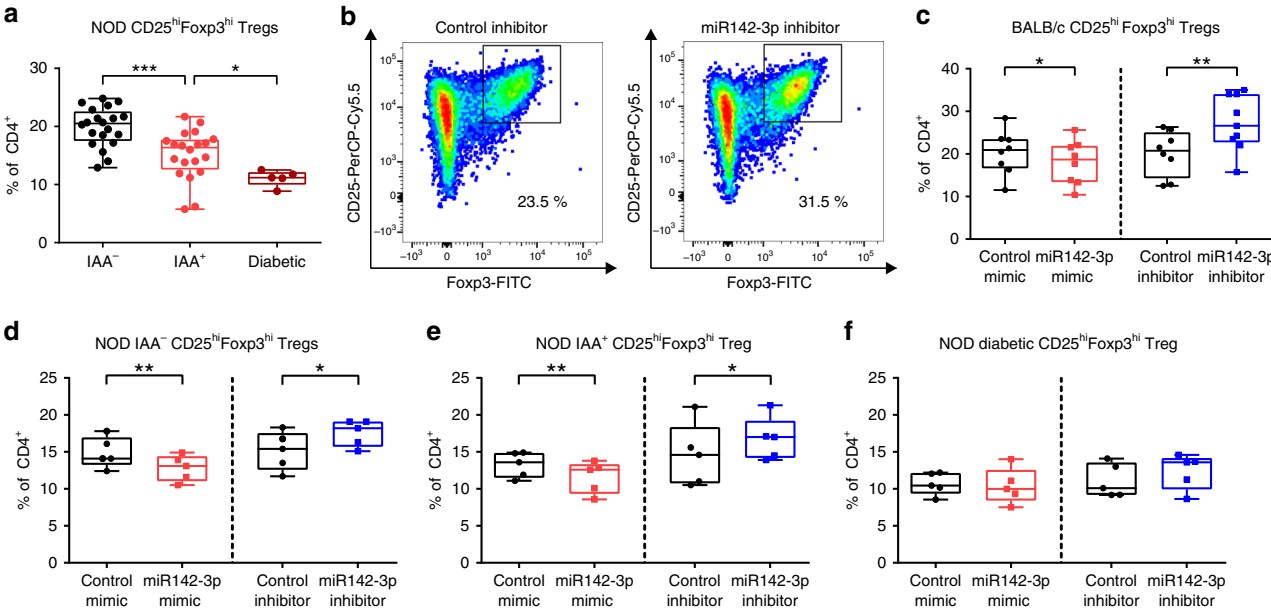

**Fig. 2 Inhibition of miR142-3p improves murine Treg induction in vitro. a** In vitro Treg induction assays using limited TCR stimulation of naive CD4[+] T cells isolated from lymph nodes of NOD mice with different stages of autoimmunity, as revealed by flow cytometry. $n = 20$ for IAA[−] and IAA[+]; $n = 5$ for diabetic. **b** Representative flow cytometry plots for in vitro Treg induction assays in presence of a miR142-3p inhibitor and a control inhibitor, respectively, using naive CD4[+] T cells isolated from lymph nodes of BALB/c mice. **c** In vitro Treg induction assays in presence of a miR142-3p mimic and a miR142-3p inhibitor, respectively, using naive CD4[+] T cells isolated from lymph nodes of BALB/c mice, as revealed by flow cytometry, $n = 8$. **d–f** In vitro Treg induction assays in presence of a miR142-3p mimic and a miR142-3p inhibitor, respectively, using naive CD4[+] T cells isolated from lymph nodes of NOD mice with different stages of autoimmunity, as revealed by flow cytometry, $n = 5$. One data point represents one subject. Experiments were performed in three technical replicates per subject. Data are presented as box-and-whisker plots with mean, 25% percentile, 75% percentile, minimum and maximum values. **a** Ordinary one-way ANOVA, Tukey's multiple comparisons test. **c–f** Student's t-test, $*P < 0.05$, $**P < 0.01$, and $***P < 0.001$. The source data are provided as a Source Data file.

Tgfbr1 and ATG16L1, two established target genes of miR142-3p[47,48], after 6 h (Supplementary Fig. 6c, d).

Modulating miR142-3p activity by a mimic or inhibitor likewise decreased or enhanced Treg induction efficacy, respectively with naive CD4[+] T cells from NOD mice that had not yet developed IAA (Fig. 2d). Importantly, inhibiting miR142-3p resulted in a significant improvement of Treg induction using NOD mice with recent development of IAA[+] autoimmunity (Fig. 2e), while no significant changes in Treg induction potential were observed using T cells from NOD mice with established T1D (Fig. 2f).

Next, to assess the relevance of miR142-3p activity for human Treg induction, we used naive CD4[+] T cells (CD4[+]CD3[+] CD45RA[+]CD45RO[−]CD127[+]CD25[−]; Supplementary Fig. 4b) from individuals without islet autoimmunity, with recent onset of T1D, and with established T1D. Comparing these groups without miR142-3p modulation, we identified an overall reduction of Treg induction efficacy compared to healthy individuals (Fig. 3a), as observed previously[6]. In naive CD4[+] T cells from subjects without islet autoimmunity, the miR142-3p mimic significantly reduced the frequency of induced Tregs, while the inhibition of miR142-3p improved Treg induction efficacy (Fig. 3c), just as we had observed in the mouse model (a representative flow cytometry staining is shown in Fig. 3b and Supplementary Fig. 5b).

Increasing miR142-3p activity did not further reduce Treg induction efficacy in naive CD4[+] T cells from children with recent onset of T1D (Fig. 3d) or individuals with established T1D (Fig. 3e), presumably because miR142-3p levels were already saturated in these cells. However, in both groups the inhibition of miR142-3p resulted in significantly higher frequencies of induced Tregs (Fig. 3d, e). These findings suggest that high levels of

miR142-3p limit in vitro Treg induction efficacy during islet autoimmunity and T1D in humans and mice, while miR142-3p inhibition was able to restore this impairment.

**Inhibition of miR142-3p improves Treg stability**. Having demonstrated that the inhibition of miR-142-3p improves murine and human Treg induction efficacy in vitro, we asked whether this improvement was accompanied by increased Treg stability. To assess phenotypic stability, we performed in vitro Treg induction assays as before and subsequently restimulated the induced Tregs for 30 h, both in the presence of the miR142-3p inhibitor or a control inhibitor (Fig. 4a). Remarkably, the addition of the miR142-3p inhibitor during restimulation resulted in a significantly higher maintenance of the Treg phenotype, as evidenced by high levels of Foxp3 and CD25 (Fig. 4b). There were no significant differences in cell viability and Ki67 expression after Treg induction or restimulation, excluding altered cell survival or proliferation as a contributing factor (Supplementary Fig. 7a–c). Thus, the beneficial effect of miR142-3p inhibition on Treg induction efficacy is accompanied by an increased stability of the Treg phenotype.

**T cell activation changes Foxp3 CNS2 DNA methylation**. To dissect the mechanism underlying the miR142-3p effect, we analyzed the methylation status of the Foxp3 gene at the conserved CNS2, as this epigenetic mark is a key player in regulating the Foxp3 locus. First, we employed high-resolution melting PCR and pyrosequencing to determine the methylation status of the Foxp3 CNS2 in in vitro-induced Tregs in the human and the murine systems. Although in vitro-induced human Tregs exhibit a methylated CNS2, it was shown that Treg induction using

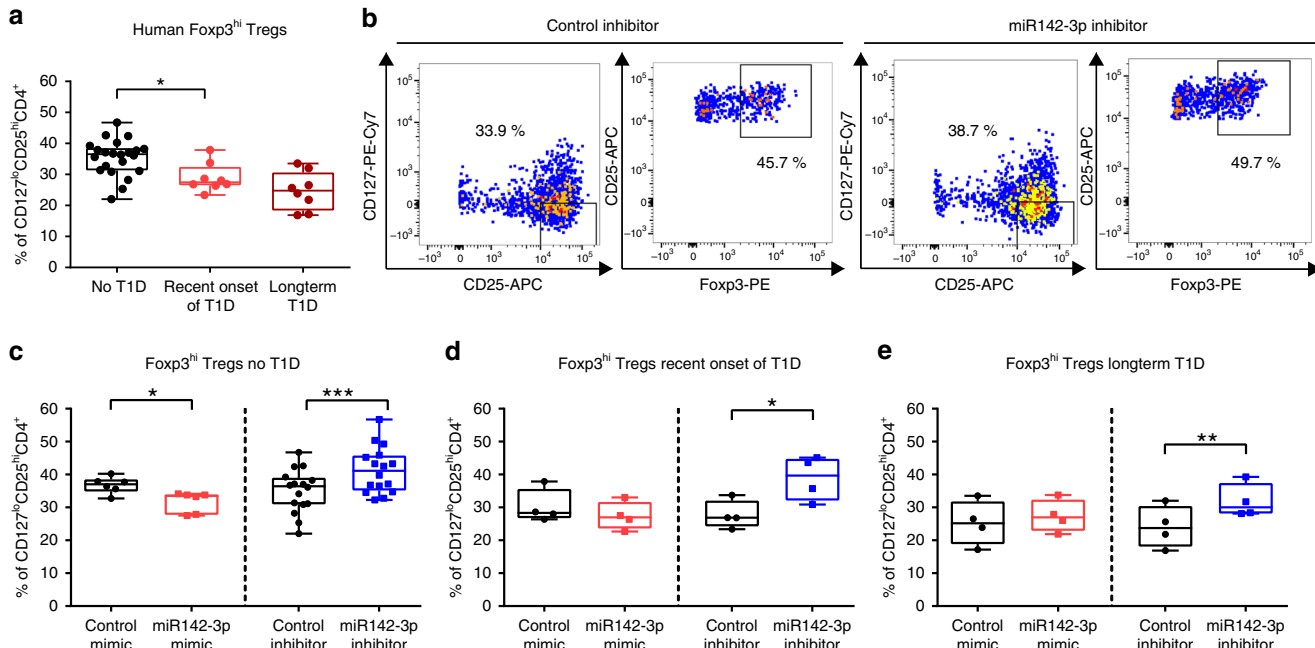

**Fig. 3 Inhibition of miR142-3p improves human Treg induction in vitro. a** In vitro Treg induction assays using limited TCR stimulation of naive CD4+ T cells isolated from peripheral blood of human subjects with different stages of T1D, as revealed by flow cytometry. $n = 22$ for no T1D $n = 8$ for recent onset and long-term T1D. **b** Representative flow cytometry plots for in vitro Treg induction assays in presence of a miR142-3p inhibitor and a control inhibitor, respectively, using naive CD4+ T cells isolated from human peripheral blood. **c-e** In vitro Treg induction assays in presence of a miR142-3p mimic and a miR142-3p inhibitor, respectively, using naive CD4+ T cells isolated from human peripheral blood, as revealed by flow cytometry, (**c**) no autoimmunity $n = 6$ for mimic $n = 16$ for inhibitor, (**d**) recent onset of type 1 diabetes $n = 5$, (**e**) long-term type 1 diabetes $n = 4$. One data point represents one subject. Experiments were performed in three technical replicates per subject. Data are presented as box-and-whisker plots with mean, 25% percentile, 75% percentile, minimum and maximum values. **a** Ordinary one-way ANOVA, Tukey's multiple comparisons test. **c-e** Student's t-test, *$P < 0.05$, **$P < 0.01$, and ***$P < 0.001$. The source data are provided as a Source Data file.

subimmunogenic TCR stimulation results in higher stability of induced Tregs[7]. To capture potential methylation dynamics during the early phase of murine and human Treg induction from naive CD4+ T cells, we analyzed Treg markers and Foxp3 CNS2 methylation as early as 12 h following TCR stimulation. We observed cells expressing Treg markers, especially high levels of Foxp3, with an increase over time as shown for 12 and 18 h in Fig. 4c (murine: CD4+CD25hiFoxp3hi) and Fig. 4e (human: CD4+CD127loCD25hiFoxp3hi).

The DNA methylation analysis of cells with a Foxp3hi Treg phenotype in this early phase of Treg induction showed a clear dynamic in Foxp3 CNS2 methylation, which are only partially captured by the in vitro differentiation system. Thus, initial Treg induction from naive CD4+ T cells causes rapid CNS2 demethylation (Fig. 4d, f) and Foxp3 activation as expected. However, longer culture leads to methylation at the CNS2, likely a result of the culture conditions. Of note, the demethylation was restricted to cells showing a Foxp3hi Treg phenotype, while no changes in CNS2 methylation were observed in Foxp3− cells (Supplementary Fig. 8a–d). These results support the notion that the initial demethylation of CNS2 is linked to the expression of Foxp3 and its downstream targets, and is not solely the effect of TCR stimulation on cell proliferation.

**miR142-3p targets Tet2, a modulator of DNA methylation.** Inhibition of miR142-3p increases Treg induction and stability in vitro, and this process is accompanied by a very rapid demethylation of the crucial Foxp3 CNS2. Therefore, we hypothesized that miR142-3p targets critical components in the DNA demethylation machinery. We scanned predicted target genes of

miR142-3p[45] and the mRNA targets that were associated with the RISC in human CD4+ T cells (Fig. 1) for genes that can be involved in demethylation. One interesting candidate was TET2, which contains two predicted miR142-3p-binding sites in the 3′ UTR[45] and exhibited multiple RISC footprints as shown by the respective mRNA fragments aligned to the TET2 transcript (Supplementary Fig. 9). The presence of TET2 mRNA fragments in RISC footprints of the HITS-CLIP data indicated that TET2 is targeted by miRNAs. In addition, the predicted miR142-3p-binding sites pointed to an involvement of this miRNA in TET2 regulation.

To dissect if Tet2 links high miR142-3p levels to the regulation of Treg induction efficacy and stability, we analyzed Tet2 gene expression during the early phase of Treg induction in vitro from BALB/c naive CD4+ T cells. We observed a strong increase in Tet2 mRNA levels after 3 h of TCR stimulation, followed by a decrease to baseline levels or below after 6 h (Fig. 5a). The activation of Tet2 upon TCR stimulation of naive CD4+ T cells was also reflected on the protein level, with increased protein levels after 6 h and an additional increase after 12 h (Fig. 5c; a representative flow cytometry staining including controls is shown in Supplementary Fig. 10a). Thus, the kinetics of Tet2 accumulation are consistent with a role for Tet2 in Foxp3 demethylation.

In order to provide further support for Tet2 as a direct target of miR142-3p, we performed Treg induction experiments using naive CD4+ T cells from BALB/c mice in the presence of the miR142-3p inhibitor or control inhibitor, and analyzed Tet2 mRNA and protein expression. Of note, inhibition of miR142-3p resulted in significantly higher Tet2 mRNA levels after 6 h of TCR stimulation (Fig. 5b), and increased Tet2 protein abundance after 12 h of TCR stimulation (Fig. 5d). While Tet1 and Tet3

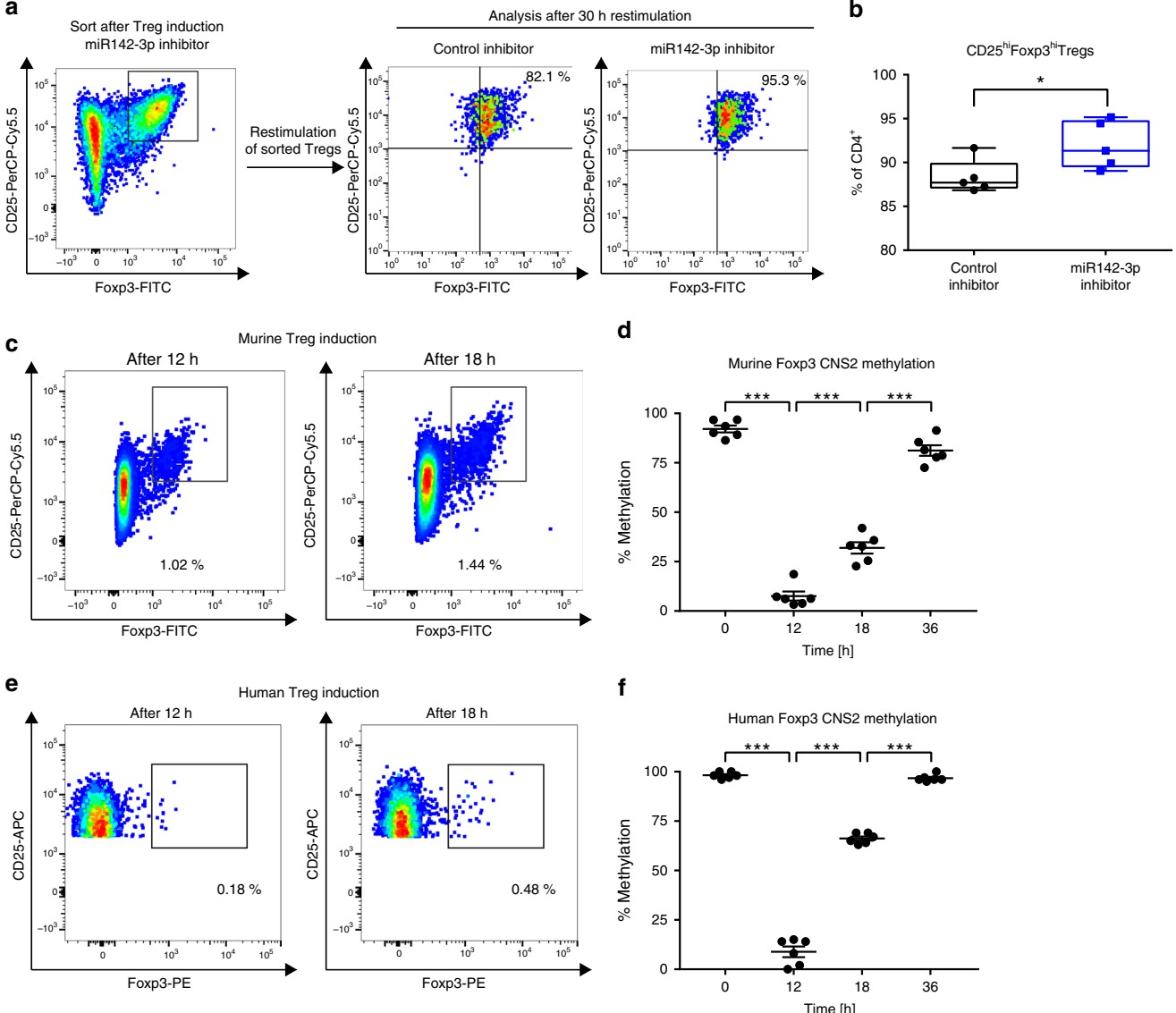

**Fig. 4 miR142-3p inhibition improves stability of in vitro-induced Tregs. a** Procedure and representative flow cytometry plots of restimulation experiments in presence of a miR142-3p inhibitor and a control inhibitor, respectively, using Tregs previously induced from naive CD4+ T cells isolated from lymph nodes of BALB/c Foxp3GFP reporter mice. **b** Quantification of restimulation experiments as described in **a**, as revealed by flow cytometry, $n = 5$. **c** Representative flow cytometry plot indicating CD4+CD25+Foxp3+ Tregs after 12 and 18 h of TCR stimulation of naive CD4+ T cells isolated from lymph nodes of BALB/c mice. **d** Methylation of four CpG sites in the murine Foxp3 CNS2 of CD4+CD25+Foxp3+ Tregs after 0, 12, 18, and 36 h of subimmunogenic TCR stimulation of naive CD4+ T cells isolated from lymph nodes of BALB/c mice, as revealed by pyrosequencing, $n = 6$. **e** Representative flow cytometry plot indicating CD4+CD127−CD25+Foxp3+ Tregs after 12 and 18 h of TCR stimulation of human naive CD4+ T cells. **f** Methylation of eight CpG sites in the human Foxp3 CNS2 of CD4+CD127-CD25+Foxp3+ Tregs after 0, 12, 18, and 36 h of subimmunogenic TCR stimulation of naive CD4+ T cells isolated from human peripheral blood, as revealed by pyrosequencing, $n = 6$. One data point represents one subject. Experiments were performed in two (**d**, **f**) or three (**b**) technical replicates per subject. Data are presented as box-and-whisker plots with mean, 25% percentile, 75% percentile, minimum and maximum values or as means ± s.e.m., **b** Student's *t*-test. (**d**, **f**) Ordinary one-way ANOVA, Tukey's multiple comparisons test. *$P < 0.05$, **$P < 0.01$, and ***$P < 0.001$. The source data are provided as a Source Data file.

expression was also dynamic following T cell activation there was no significant effect of miR142-3p inhibition, supporting the notion that miR142-3p specifically targets Tet2 (Supplementary Fig. 11a, b).

Moreover, a miR142-3p mimic induced a significant reduction in luciferase activity in HEK-293 cells transfected with a wild-type TET2 3′ UTR reporter construct while there was no effect of the mimic in cells transfected with a reporter construct containing the TET2 3′ UTR with mutated miR142-3p-binding sites (Fig. 5e, Supplementary Fig. 12a). These results support the notion that miR142-3p specifically targets Tet2.

**Loss-of-function models confirm Tet2 as a miR142 target**. In addition, we used two loss-of-function models to mechanistically validate Tet2 as a direct target of miR142-3p: Given the broad role of Tet2 in various cell types, we first used the non-hematopoietic 3T3 fibroblast cell line, which was previously shown to lack miR142-3p expression almost completely[49] (Supplementary Fig. 12b). As a second approach, we dissected the miR142-3p/Tet2 axis using T cells from miR142 knockout (miR142−/−) mice[50] (Supplementary Fig. 12c).

Specifically, we transfected 3T3 fibroblasts with a miR142-3p mimic or a control mimic to introduce miR142-3p levels given

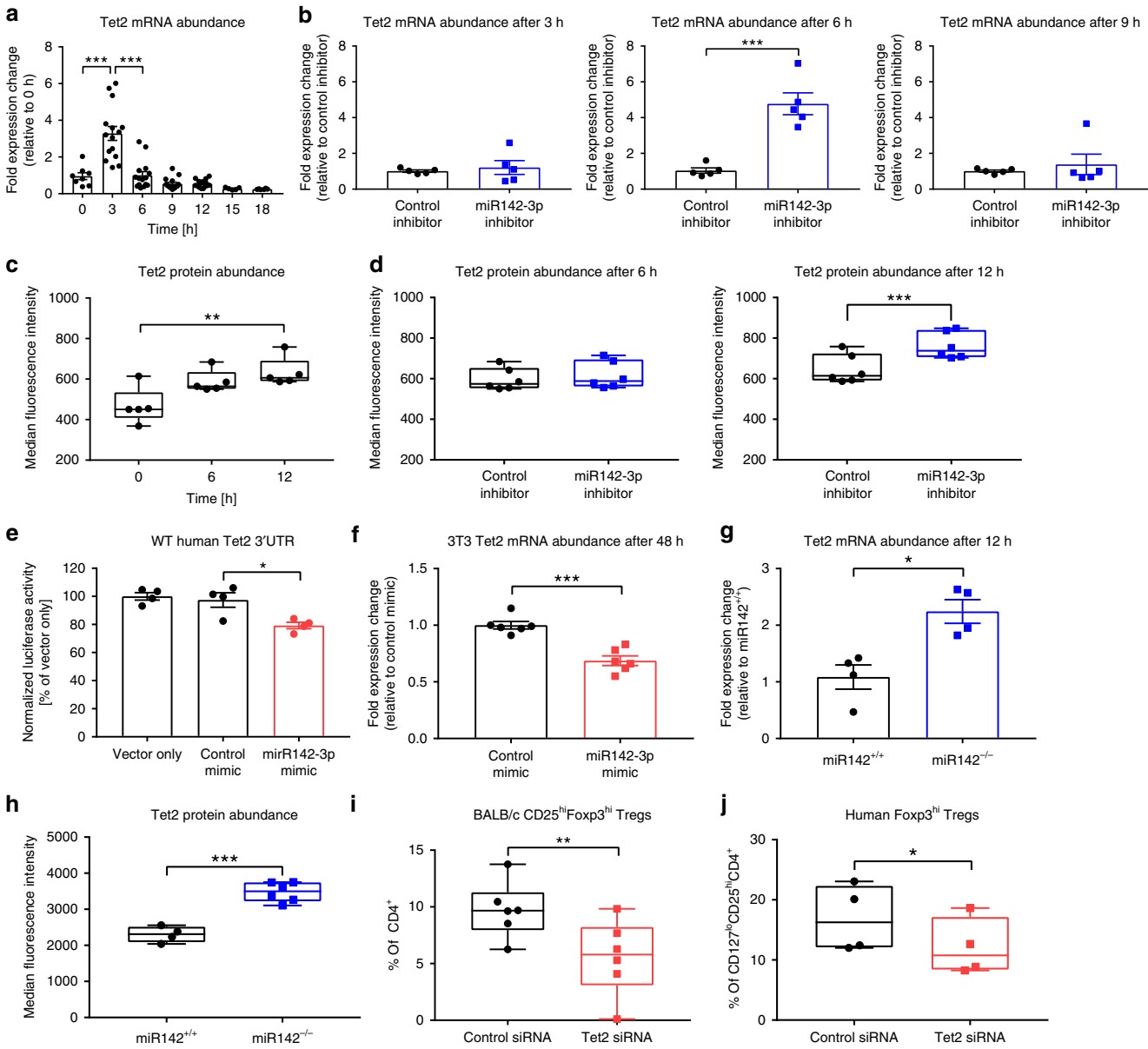

**Fig. 5 miR142-3p targets methylcytosine dioxygenase Tet2. a** Levels of Tet2 mRNA after TCR stimulation of CD4[+] T cells isolated from lymph nodes of BALB/c mice, as revealed by qPCR normalized to Histone mRNA, $n \geq 8$. **b** Levels of Tet2 mRNA after TCR stimulation of CD4[+] T cells as described in **a**, in presence of miR142-3p inhibitor, as revealed by qPCR normalized to Histone mRNA, $n = 5$. **c** Tet2 protein abundance after TCR stimulation of CD4[+] T cells as described in **a**, as revealed by flow cytometry, $n = 5$. **d** Tet2 protein abundance after TCR stimulation of CD4[+] T cells as described in **a**, in presence of miR142-3p inhibitor, as revealed by flow cytometry, $n = 6$. **e** Normalized luciferase activity of HEK-293 cells cotransfected with miR142-3p mimic and wild-type TET2 3′UTR reporter construct, $n = 4$. **f** Levels of Tet2 mRNA in 3T3 fibroblasts transfected with miR142-3p mimic, as revealed by qPCR normalized to Histone mRNA, $n = 6$. **g** Levels of Tet2 mRNA after TCR stimulation of CD4[+] T cells isolated from lymph nodes of miR142[+/+] and miR142[-/-] mice, as revealed by qPCR normalized to Histone mRNA, $n = 4$. **h** Tet2 protein abundance after limited TCR stimulation of CD4[+] T cells as described in **g**, as revealed by flow cytometry, $n = 6$. **i** In vitro Treg induction in presence of Tet2 siRNA, using naive CD4[+] T cells isolated from lymph nodes of BALB/c mice, as revealed by flow cytometry, $n = 6$. **j** In vitro Treg induction in presence of Tet2 siRNA, using naive CD4[+] T cells isolated from human peripheral blood, as revealed by flow cytometry, $n = 4$. One data point represents one subject. Experiments were performed in three technical replicates per subject. Data are presented as box-and-whisker plots with mean, 25% percentile, 75% percentile, minimum and maximum values or as means ± s.e.m. (**a**, **c**, **e**) Ordinary one-way ANOVA, Tukey's multiple comparisons test. (**b**, **d**, **f**–**j**) Student's *t*-test, *$P < 0.05$, **$P < 0.01$, and ***$P < 0.001$. The source data are provided as a Source Data file.

the absence of endogenous miR142-3p expression. The introduction of miR142-3p expression in these cells following mimic application resulted in a significant decrease of Tet2 mRNA levels (Fig. 5f). To further strengthen the mechanistic molecular evidence for the direct miR142-3p/Tet2 relationship, we used T cells from miR142 knockout animals. Stimulation of

CD4[+] T cells from miR142 knockout mice demonstrated significantly higher Tet2 mRNA levels when compared to T cells from miR142 competent mice (Fig. 5g). The direct targeting of Tet2 by miR142 was also validated on the protein level (a representative flow cytometry staining is shown in Supplementary Fig. 10b). Specifically, T cells from miR142-

deficient animals presented with significantly higher Tet2 protein expression following TCR stimulation when compared to T cells from miR142$^{+/+}$ mice (Fig. 5h). Furthermore, we stimulated CD4$^+$ T cells from miR142 knockout mice in the presence of a miR142-3p inhibitor or a control inhibitor in vitro. As expected, the inhibitor did not increase mRNA abundance of Tet2 or the established target genes of miR142-3p, Tgfbr1, and ATG16L1 (Supplementary Fig. 12d–f). These findings underline and confirm the direct effect of the miR142-3p inhibitor in miR142 competent mice and provide compelling evidence for direct targeting of Tet2 by miR142.

To directly test the importance of Tet2 for murine and human Treg induction, we purified naive CD4$^+$ T cells from BALB/c mice and human subjects without T1D, and performed Tet2 knockdown experiments using a Tet2 short interfering RNA (siRNA). Reduced Tet2 abundance during Treg induction resulted in a significantly attenuated Treg induction efficacy in both the murine and human setting (Fig. 5i, j). Taken together, these results suggest a critical role for Tet2 in T cell activation and Treg induction, and confirm the identification of Tet2 as an important direct target of miR142-3p.

**Islet autoimmunity changes Tet2 levels and CNS2 methylation**. Next, to investigate the relevance of these findings to islet autoimmunity, we analyzed if the onset of islet autoimmunity alters miR142-3p levels and Tet2 abundance to directly contribute to impaired Treg induction. To this end, we compared IAA$^+$ to control, IAA$^-$ NOD mice. IAA$^+$ NOD mice exhibited higher miR142-3p levels in CD4$^+$ T cells compared to IAA$^-$ mice (Fig. 1h), which correlated with reduced protein abundance of its target Tet2 (Fig. 6a). We confirmed the reduction in Tet2 abundance in CD4$^+$T cells by immunofluorescence microscopy of pancreatic cryosections from NOD mice with or without IAA$^+$ autoimmunity, which revealed significantly reduced numbers of pancreas-infiltrating CD3$^+$Tet2$^+$ T cells in IAA$^+$ mice (Fig. 6b, c and Supplementary Fig. 13a) despite an increase in the total number of pancreas-infiltrating CD3$^+$ T cells (Supplementary Fig. 13b). Furthermore, immunofluorescence microscopy of CD4$^+$ T cells purified from human peripheral blood showed a reduced abundance of CD3$^+$Tet2$^+$ T cells in subjects with recent onset of T1D compared to subjects without T1D (Fig. 6d, e).

If our proposed miR142-3p/Tet2/Foxp3 axis indeed exists, one would expect DNA methylation at the Foxp3 CNS2 to be altered by islet autoimmunity accordingly. Therefore, we analyzed the Foxp3 CNS2 methylation of Tregs from NOD mice with recent development of IAAs, and also in children with recent onset of T1D and compared them to controls without IAA/T1D. As shown in Fig. 6f, g, all individual CpG sites within the CNS2 of Tregs from individuals, whether mouse or humans, with islet autoimmunity showed higher DNA methylation levels, and this effect reached statistical significance at several sites. In addition, analysis of the entire Foxp3 CNS2 region demonstrated significantly increased DNA methylation (Fig. 6f, g).

To dissect the potential causative contribution of impaired Treg stability in promoting autoimmune progression, we assessed Foxp3 CNS2 DNA methylation in NOD mice <30 days of age with a very early onset of IAA$^+$ positivity. Of note, these very young IAA$^+$ NOD mice presented with distinctly increased Foxp3 CNS2 DNA methylation when compared to nonautoimmune prone BALB/c mice (Supplementary Fig. 13c), supporting the concept that early onset Treg instability can be involved in promoting autoimmune activation and progression.

Since the composition of the Treg pool with regard to thymic-derived and peripheral Tregs might have an effect on differential Foxp3 CNS2 methylation, we analyzed Tregs isolated from peripheral blood of human subjects without T1D and with recent onset of T1D. The expression of Helios, which is a marker of thymic-derived Tregs[51], was not significantly different between the two groups (Supplementary Fig. 13d). These results support a direct link between miR142-3p, Tet2 expression, and Foxp3 CNS2 methylation affecting Treg induction and stability during ongoing islet autoimmunity.

**miR142-3p inhibitor improves NOD islet autoimmunity in vivo**. To address the pathological relevance of these findings in vivo, we next analyzed the effect of miR142-3p inhibition in NOD mice with ongoing IAA$^+$ autoimmunity. We used an established LNA-miRNA-inhibitor, which has been shown to accumulate in tissues, including lymphoid tissues, when applied systemically[52] thereby facilitating miRNA silencing in vivo[6,53]. The miR142-3p inhibitor was applied at 10 mg/kg, intraperitoneally (ip) every other day for 14 days. Using a fluorescently labeled miR142-3p inhibitor, we confirmed the successful delivery of the inhibitor to CD4$^+$ T cells in relevant draining lymph nodes, including liver-draining lymph nodes, mesenteric lymph nodes, as well as pancreatic lymph nodes and directly in pancreas-residing CD4$^+$ T cells as assessed after 4 h (Supplementary Fig. 14a) and 24 h (Supplementary Fig. 14b). In addition, the successful delivery of the inhibitor to CD4$^+$ T cells was also demonstrated in lymph nodes at the end of the 14 days application period (Supplementary Fig. 14c, d). Moreover, the efficiency of the delivered miR142-3p inhibitor in pancreatic tissue and local immune cells was validated by demonstrating increased expression of Tgfbr1, a well-established target of miR142-3p[47], in pancreatic T cells (Supplementary Fig. 14e, f).

Of note, blockade of miR142-3p-reduced pancreatic T cell infiltration as shown by histopathological analyses of pancreatic sections (Fig. 7a, b). These effects of miR142-3p inhibition were accompanied by reduced IAA levels (Fig. 7c), while blood glucose levels or body mass were unaffected (Supplementary Fig. 15a, b). Importantly, immunofluorescence analyses revealed significantly enhanced frequencies of Foxp3$^+$ Tregs within the pancreas following blockade of miR142-3p in vivo (Fig. 7d, e and Supplementary Fig. 15c). In addition and in line with a direct targeting of Tet2 by miR142-3p its blockade in vivo resulted in a significant increase in Tet2 expression in pancreatic T cells of NOD mice (Fig. 7f, g). These results were accompanied by an elevated Tet2 expression (Supplementary Fig. 15d) and a reduced proliferation indicated by lower Ki67 expression (Supplementary Fig. 15e) in peripheral T cells of miR142-3p inhibitor-treated mice.

In order to directly link these observations with a modulation of Treg stability, we next investigated the effects of in vivo miR142-3p inhibition on Foxp3 CNS2 demethylation in Tregs. To this end, we analyzed CNS2 methylation of Tregs from pancreatic lymph nodes of inhibitor-treated IAA$^+$ NOD mice. All individual CpG sites within the CNS2 of miR142-3p inhibitor-treated mice exhibited decreased DNA methylation, and this effect reached statistical significance when the entire CNS2 region was considered (Fig. 7h).

To confirm that the observed effects of miR142-3p inhibitor application on Treg induction, stability and Tet2 in vivo were directly mediated by reduced miR142-3p activity, we applied the miR142-3p inhibitor to miR142 knockout mice. As expected, the inhibitor had no effect on Treg frequency (Supplementary Fig. 16a), Tet2 protein abundance (Supplementary Fig. 16b), or Foxp3 CNS2 methylation (Supplementary Fig. 16c) in miR142-deficient mice.

Finally, to assess a potential relevance of miR142-3p inhibition in human T cells from an established T1D environment, we used

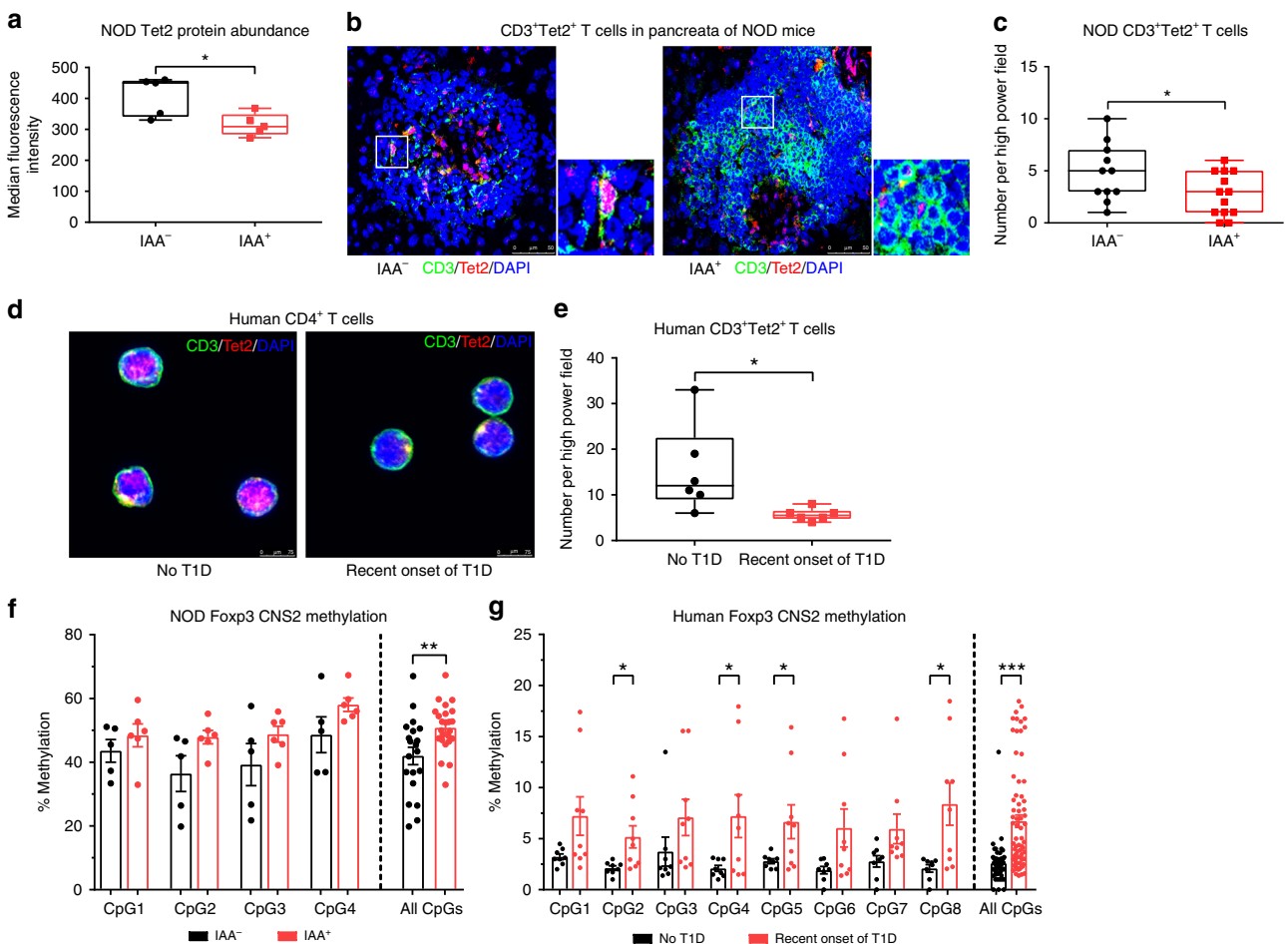

**Fig. 6 Tet2 abundance and Foxp3 CNS2 methylation are changed in islet autoimmunity. a** Ex vivo Tet2 protein abundance (median fluorescence intensity) in CD4[+] T cells isolated from pancreatic lymph nodes of NOD mice with and without islet autoantibodies, as revealed by flow cytometry, $n = 5$. **b** Immunofluorescence staining for CD3 (green), Tet2 (red), and DAPI (blue) in pancreas cryosections of NOD mice with and without islet autoimmunity. Scale bars: 50 µm. **c** Quantification of CD3[+]Tet2[+] T cells per high-power field in samples from **b**, $n = 12$. **d** Immunofluorescence staining for CD3 (green), Tet2 (red), and DAPI (blue) in cytospins of human CD4[+] T cells isolated from peripheral blood of individuals with and without recent onset of T1D. Scale bars: 75 µm. **e** Quantification of CD3[+]Tet2[+] T cells per high-power field in samples from **d**, $n = 6$. **f** Methylation of the Foxp3 CNS2 (four CpG sites and combination of all sites) in Tregs isolated from pancreatic lymph nodes of female NOD mice with and without autoimmunity, as revealed by pyrosequencing, $n = 6$. **g** Methylation of the Foxp3 CNS2 (eight CpG sites and combination of all sites) in Tregs isolated from peripheral blood of male human subjects with recent onset of T1D and healthy controls, as revealed by pyrosequencing, $n = 8$. One data point represents one subject/one high-power field. Experiments were performed in two (**f** and **g**) or three (**a**) technical replicates per subject. Data are presented as box-and-whisker plots with mean, 25% percentile, 75% percentile, minimum and maximum values or as means ± s.e.m. (**f**, **g**) Ordinary one-way ANOVA, Tukey's multiple comparisons test. (**a**, **c**, **e**) Student's $t$-test, *$P < 0.05$, **$P < 0.01$, and ***$P < 0.001$. The source data are provided as a Source Data file.

peripheral blood mononuclear cells (PBMCs) from individuals with T1D for a pilot experiment to reconstitute the murine major histocompatibility complex II (MHCII) deficient, HLA-DQ8 transgenic NOD.Cg-$Prkdc^{scid}$ $Il2rg^{tm1Wjl}$ (NSG) mouse model[7] for the establishment of humanized mice. Application of a miR142-3p inhibitor to these humanized NSG mice showed a trend toward increased frequencies of peripheral CD127[low]CD25[hi] Tregs and a significant increase in Tet2 expression in pancreatic T cells (Supplementary Fig. 17), suggesting that further dose titration or daily application of the inhibitor might be required.

## Discussion

The molecular underpinnings promoting both onset of autoimmunity and the highly variable progression to clinically overt T1D remain poorly understood. Recent studies have highlighted miRNAs as crucial regulators of immune homeostasis, and suggested that their dysregulation contributes to the onset of autoimmunity and/or the progression to symptomatic T1D[5,6,37–40]. In

silico target prediction tools predicted a multitude of potential targets, the relevance of which have remained unclear. The identification of specific miRNAs that contribute to the onset of autoimmunity and the identification of their relevant targets, and downstream pathways will considerably advance our understanding of autoimmune activation and facilitate the development of future intervention and prevention approaches.

Here, we identified increased levels of miR142-3p, which is the most abundant active miRNA in the RISC of human CD4[+] T cells, during the onset of islet autoimmunity and T1D in CD4[+] T cells of mice and humans. We employed a combination of various molecular and cellular approaches[45,54–56], including miR142-3p modulation and loss-of-function models, to identify Tet2 as a relevant target of miR142-3p. Moreover, we provide compelling evidence for a link of miR142-3p/Tet2 signaling to impaired Treg homeostasis and function.

The onset of islet autoimmunity and T1D in mice or humans is accompanied by a reduced capacity of naive CD4[+] T cells to be

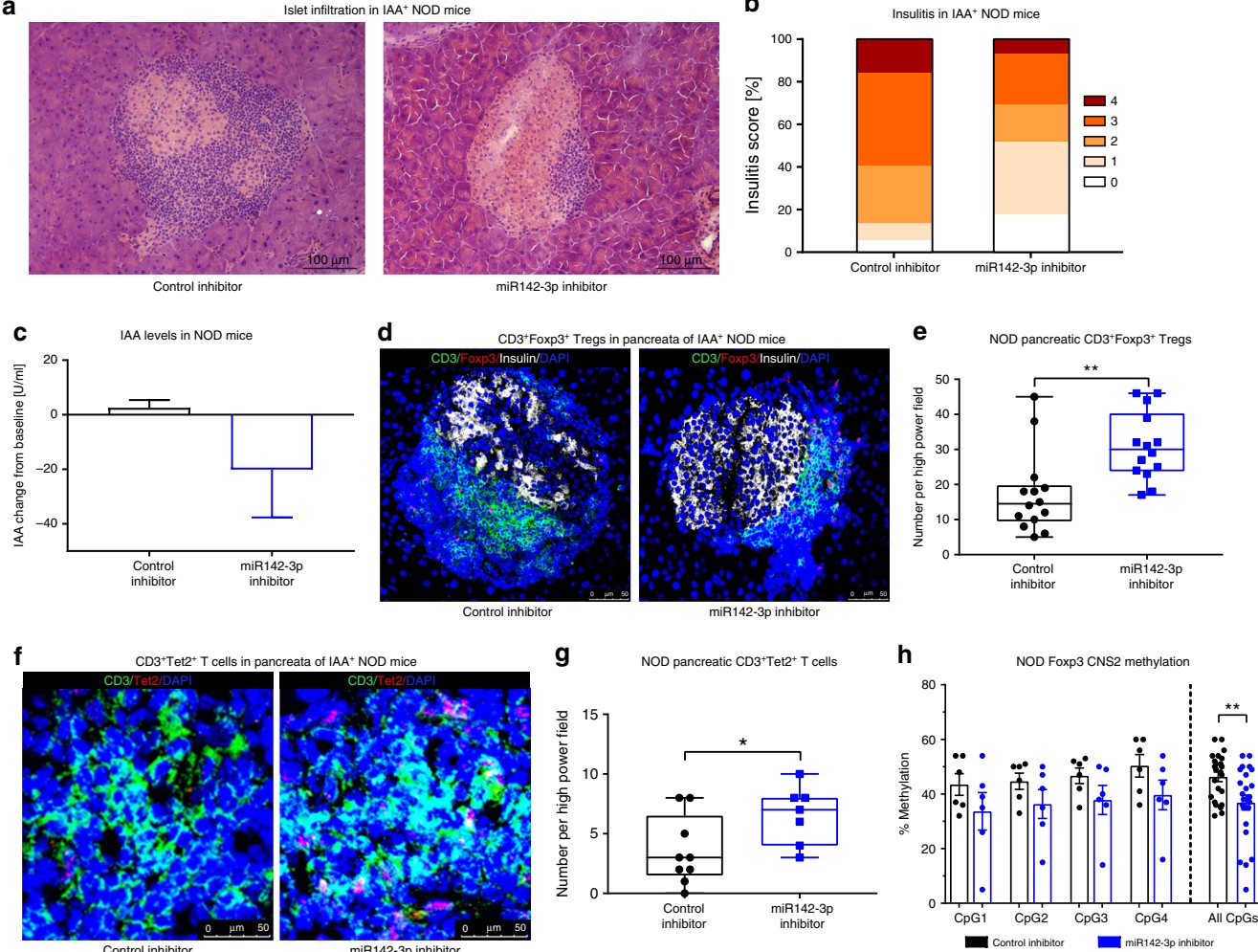

**Fig. 7 miR142-3p inhibition improves murine islet autoimmunity in vivo. a** Representative hematoxylin and eosin-stained pancreas cryosections from IAA$^+$ NOD mice treated with a miR142-3p inhibitor or control inhibitor for 14 days with 10 mg/kg ip every other day. Scale bars: 100 μm. **b** Grading of insulitis from mice treated as described in **a**, $n = 3$ per group. **c** IAA levels in serum of IAA$^+$ NOD mice after treatment with a control inhibitor or a miR142-3p inhibitor as described in **a** and revealed by ELISA. Data are shown as change from baseline, $n = 3$. **d** Immunofluorescent staining for CD3 (green), Foxp3 (red), Insulin (white), and DAPI (blue) in pancreas cryosections of IAA$^+$ NOD mice treated with a miR142-3p inhibitor or control inhibitor as described in **a**. Scale bars: 50 μm. **e** Quantification of CD3$^+$Foxp3$^+$ T cells per high-power field in samples from **c**, $n = 14$. **f** Immunofluorescent staining for CD3 (green), Tet2 (red), and DAPI (blue) in pancreas cryosections of IAA$^+$ NOD mice treated with a miR142-3p inhibitor or control inhibitor as described in **a**. Scale bars: 50 μm. **g** Quantification of CD3$^+$Tet2$^+$ T cells per high-power field in samples from **f**, $n \geq 7$. **h** Methylation of the Foxp3 CNS2 (four CpG sites and combination of all sites) in Tregs isolated from pancreatic lymph nodes of IAA$^+$ NOD mice treated with a miR142-3p inhibitor or control inhibitor as described in **a** and revealed by pyrosequencing. One data point represents one high-power field. Experiments were performed in two technical replicates per subject (**h**). Data are presented as box-and-whisker plots with mean, 25% percentile, 75% percentile, minimum and maximum values or as means ± s.e. m. **h** Ordinary one-way ANOVA, Tukey's multiple comparisons test. (**c**, **e**, **g**) Student's $t$-test, *$P < 0.05$, and **$P < 0.01$. The source data are provided as a Source Data file.

differentiated into Tregs in vitro. Critically, we could reproduce this effect by subimmunogenic in vitro Treg induction assays in the presence of a miR142-3p mimic, suggesting that the dysregulation of miR142-3p indeed contributes to impaired Treg induction. Conversely, the inhibition of miR142-3p during in vitro Treg induction improved Treg induction efficacy and was accompanied by enhanced Treg stability.

We proposed a miR142-3p/Tet2/Foxp3 axis as one possible mechanism to explain increased Treg stability upon miR142-3p inhibition. Bioinformatic prediction tools[45] as well as cellular loss function models and a broad set of molecular approaches, including a TET2 3'UTR luciferase reporter assay revealed the methylcytosine dioxygenase Tet2 as a direct target of miR142-3p. Tet enzymes are involved in the control of Treg generation and

maintenance by active demethylation of the Foxp3 CNS2[27,36], and overexpression of the Tet1 catalytic domain improves the stability of Foxp3 expression in induced Tregs[57]. Here, we document rapid demethylation of the Foxp3 CNS2 in the early phase of the Treg induction, using highly naive CD4$^+$T cells as the starting population. This effect was absent in cells with low levels of Foxp3, indicating that CNS2 demethylation does not result from TCR stimulation per se. In line with these observations, we found Tet2 expression to be increased following TCR stimulation. Importantly, the inhibition of miR142-3p in vitro resulted in increased levels of Tet2 mRNA and protein, further supporting the concept of Tet2 as a direct target of miR142-3p.

Given the key impact of Tet2 in a variety of cell types, we analyzed the role of miR142-3p signaling in nonhematopoietic

3T3 fibroblasts, which do not express miR142-3p endogenously. Here, the gain-of-function experiment using the application of a miR142-3p mimic caused a diminished Tet2 expression in line with the biochemical and molecular findings, indicating a direct miR142-3p/Tet2 axis. In addition, increased Tet2 mRNA and protein levels of in vitro-activated T cells isolated from miR142$^{-/-}$ compared to miR142 competent mice, provide compelling mechanistic evidence for direct targeting of Tet2 by miR142-3p.

Furthermore, the diminished Treg induction efficacy in the presence of a Tet2 siRNA provides additional conceptual support for the functional relevance of the miR142-3p/Tet2 axis for murine and human Treg induction, and homeostasis. The fact that reducing Tet2 expression alone was sufficient to interfere with Treg induction potential underscores the critical contribution of Tet2 when compared to previous studies, which employed combined Tet targeting[36].

From an immunopathological perspective, the analysis of murine and human T cell samples with recent onset of autoimmunity/T1D directly links Tet2 abundance and the methylation status of the Foxp3 CNS2 with islet autoimmunity in vivo. Thus, in NOD mice with islet autoimmunity CD4$^+$ T cells express low levels of Tet2 and pancreas-resident Foxp3$^+$ Tregs were significantly reduced. Similarly, CD4$^+$ T cells isolated from peripheral blood of human subjects with recent onset of T1D showed diminished Tet2 expression compared to subjects without T1D. In addition, there was a significant increase in Foxp3 CNS2 methylation of Tregs upon onset of islet autoimmunity and in T1D. These results further support our hypothesis that dysregulated DNA methylation remodeling mediates impaired Treg homeostasis and consequently can contribute to the promotion of islet autoimmunity. Importantly, the increased Foxp3 CNS2 DNA methylation in NOD mice <30 days of age with a very early onset of autoimmunity suggests a potential causative role of miR142-3p/Tet2 signaling in promoting autoimmune activation and progression.

Of note, the in vivo inhibition of miR142-3p reduced murine islet autoimmunity and islet autoantibody levels, while increasing frequencies of both Foxp3$^+$ Tregs and Tet2$^+$ T cells in the pancreas. These effects were accompanied by reduced methylation of the Foxp3 CNS2, suggesting that the observed effect of miR142-3p inhibition is due to Tet2-mediated DNA demethylation and improved Treg stability. Moreover, the results of a pilot experiment with humanized NSG mice reconstituted with PBMCs from individuals with established T1D suggests that miR142-3p inhibition might have a beneficial effect in improving Treg homeostasis even in the context of established T1D. Future studies, including dose titration and variations in application duration will be necessary to validate these findings in Tregs in the experimental setting of established T1D.

In conclusion, we identify a miR142-3p/Tet2/Foxp3 axis in murine and human CD4$^+$ T cells that during islet autoimmunity interferes with the efficient induction of Tregs and leads to impairments in Treg stability. These results offer a mechanistic model where during islet autoimmunity miR142-3p/Tet2-mediated Treg instability can contribute to autoimmune activation and progression, and suggest that targeting miR142-3p could contribute to the development of intervention strategies, aiming at improved Treg induction and stability to interfere with islet autoimmunity.

## Methods

**Human subjects.** All human studies comply with the relevant ethical regulations for work with human participants and all study participants gave written consent prior to inclusion in the Munich Bioresource project (approval number #5049/11, Technische Universität München, Munich, Germany). Venous blood was collected using sodium heparin tubes and blood volumes were based on EU guidelines with a

maximal blood volume of 2.4 ml per kg of body weight. All subjects have been already enrolled in the BABYDIAB study[3,58,59] and the DiMelli study[60] with the documented age of T1D onset. Subjects have been stratified based on the presence or absence of multiple islet autoantibodies and T1D. No islet autoimmunity and no T1D: $n = 6$; median age at sampling = 8 years, IQR (interquartile range) = 6–12 years; all male. Recent onset of T1D: $n = 10$; median age at sampling = 4 years, IQR = 3–5 years; median HbA1c = 8.9 mg/dl, IQR = 8.4–10.6 mg/dl; median time from diagnosis to sampling = 7 days, IQR = 1–11 days; all male.

**In vitro studies with primary human T cells.** Freshly isolated human CD4$^+$ T cells were cultured in X-Vivo15 Medium (Lonza) supplemented with 2 mM glutamine, 100 U/ml human recombinant IL-2 (ReproTech), 50 U/ml penicillin, 50 mg/ml streptomycin (Sigma Aldrich), and 5% heat-inactivated human AB serum (Invitrogen) at 37 °C in a humidified CO$_2$ incubator. Cell culture-treated 96-well U bottom plates were used (Bio-Greiner one).

**Mice.** CBy.PL(B6)-Thy1$^a$/ScrJ (CD90.1 BALB/c), Balb/cByJ (CD90.2 BALB/c), Balb/c.Cg-Foxp3tm2Tch/J (BALB/c Foxp3GFP), and NOD/ShiLtJ mice were obtained from the Jackson Laboratory and NOD/ShiLtJ mice were stratified according to their IAA status. Humanized mice, NOD.Cg-Prkdc$^{scid}$ H2-Ab1$^{tm1Gru}$ Il2rg$^{tm1Wjl}$ Tg(HLA-DQA1,HLA-DQB1) 1Dv//Sz mice lack mouse MHC class II and transgenically express human HLA-DQ8. These mice were obtained from and developed by Dr. Leonard D. Shultz. To develop this stock, B10M-HLA-DQ8 mice were kindly provided by Dr. Chella David[61]. The DQ8 transgene was backcrossed for ten generations on the NSG strain background. The NSG-DQ8 mice were then intercrossed with NSG mice lacking mouse MHC class II (NOD.Cg-Prkdc$^{scid}$ H2-Ab1$^{tm1Gru}$ Il2rg$^{tm1Wjl}$)[62]. miR142$^{-/-}$ mice were developed by excision of the miR142 region in C57BL/6 N mouse embryonic stem cells[50] and obtained from the mouse facility of the German Center for Lung Research, Universities of Giessen and Marburg (Mfd Diagnostics, Wendelsheim, Germany). When possible littermate controls were used, otherwise age- and sex-matched mice. Mice were maintained under specific pathogen-free conditions on 12-h/12-h light dark cycle at 25 °C with ad libitum access to water and a standard diet at the animal facility of Helmholtz Zentrum München, Munich, Germany according to guidelines established by the Institutional Animal Committees, including all relevant ethical regulations for animal testing and research. Ethical approval for all mouse experimentations has been received by the District Government of Upper Bavaria, Munich, Germany (approval # ROB-55.2-2532.Vet_02-17-130).

**In vitro studies with primary murine T cells.** Freshly isolated murine CD4$^+$ T cells were cultured in RPMI media (Gibco by life technologies) supplemented with 10% FCS, 1 mM sodium pyruvate (Sigma Aldrich), 50 mM b-mercaptoethanol (Amimed), 1× nonessential amino acids (Merck Millipore), 100 U/ml human recombinant IL-2 (ReproTech), 100 U/ml penicillin, and 100 mg/ml streptomycin (Sigma Aldrich) at 37 °C in an humidified CO$_2$ incubator. Cell culture-treated 96-well U bottom plates were used (Bio-Greiner one).

**Murine insulin autoantibody assay.** Levels of NOD IAA were measured with a mouse high specificity/sensitivity competitive IAA assay in an enzyme-linked immunosorbent assay (ELISA) format and sera from NOD mice. In brief, high-binding 96-well plates (Costar) were coated with human recombinant insulin (100 U/ml; Humulin; Lilly) overnight at 4 °C. Unspecific blocking was performed with PBS containing 2% BSA for 2 h at room temperature. Preincubated NOD sera (diluted 1:10) with or without insulin competition were added and incubated for 2 h at room temperature. After four wash steps, biotinylated antimouse IgG1 (Abcam), diluted 1:10,000 in PBS/BSA was added for 30 min at room temperature. After washing horseradish peroxidase-labeled streptavidin was added for 15 min. After five additional washing steps, TMB substrate solution was added (OptEIA reagent set; BD). All samples were measured in duplicates with and without competition using human insulin.

In a second approach, to determine levels of IAA in NOD mice, a Protein A/G radiobinding assay based on $^{125}$I-labeled recombinant human insulin, was applied as previously described[59]. Serum from C57Bl/6 mice was used as negative control.

**Cell isolation.** Human PBMCs were isolated from fresh venous blood by density gradient centrifugation using Ficoll-Paque PLUS (GE Healthcare). CD4$^+$ T cells were purified from PBMCs by Magnetic Activated Cell Sorting (MACS) using CD4 microbeads (Miltenyi Biotec) following the manufacturer's protocol.

Murine lymph nodes and spleens were passed through 70 μm cell strainers, stained with a CD4-Biotin antibody (BD Bioscience) and MACS purified using Streptavidin Microbeads (Miltenyi Biotec) following the manufacturer's protocol.

**Cell staining and sorting.** To prevent unspecific signals, the isolated cells were incubated with Fc-Block (Biolegend) and afterward with fluorochrome-labeled antibodies for 30 min on ice in the dark. For FACS staining of human T cells the following monoclonal antibodies were used: from BD Biosciences (San Jose, CA): anti-CD25 APC (2A3), anti-CD45RO APC-H7 (UCHL1), anti-CD4 V500 (RPA-T4), and anti-HLA-DR PerCP-Cy5.5 (L243); from Biolegend (San Diego, CA):

anti-CD45RA FITC (HI100), anti-CD3 PerCP-Cy5.5 (HIT3a), anti-CD127 PE-Cy7 (A019D5), and anti-CD3 AlexaFluor700 (HIT3a); from eBioscience (San Diego, CA): anti-FOXP3 PE (236 A/E7). For murine FACS staining, the following monoclonal antibodies were used: from BD Biosciences (San Jose, CA): anti-CD4 Biotin (GK1.5); from Biolegend (San Diego, CA): CD25 PerCP-Cy5.5 (PC61) and CD44 PE (IM7) Ki67 APC (16A8); from eBioscience (San Diego, CA): CD4 AlexaFluor700 (RM4-5), CD62L APC (MEL-14), and Foxp3 FITC (FJK-16s); from Abiocode: Tet2 (10F1). After surface staining, the cells were fixed and permeabilized using the Foxp3 Staining Buffer Kit (eBioscience) to enable the detection of the intracellular abundance of Foxp3, Ki67, and Tet2. Cells were acquired on the BD FACS Aria III cell sorting system using FACS Diva software with optimal compensation and gain settings determined for each experiment based on unstained and single-color stained samples. Doublets were excluded based on SSC-A vs. SSC-W plots. Live cell populations were gated on the basis of cell side and forward scatter, and the exclusion of cells positive for Sytox Blue (Life Technologies) or Fixable Viability Dye eFluor450 (ebioscience). Flow cytometry data were analyzed using FlowJo software version 7.6.1 (TreeStar Inc., OR)

**HITS-CLIP**. Human CD4$^+$ T cells were homogenized subjected to UV-crosslinking three times at 400 mJ/cm$^2$ and extra RNA bound to the RISC complex was digested with RNase T1. The Argonaute complex was then immunoprecipitated 2–4 h at 4 °C using the monoclonal Argonaute antibody 2A8. Phosphate from the precipitated complex was removed and a 5′-32P-labeled RL3 linker was attached. The samples were separated by SDS-gel electrophoresis, transferred to a nitrocellulose membrane and visualized by autoradiography. The desired area with the complexed argonaute protein was cut and subjected to RNA extraction; afterward the 5′ RNA linker was ligated. In the following, RNAs were amplified by RT/PCR, the resulting products were separated on agarose gel and two respective populations of RNA (50–60 bp for small RNAs, 80–180 bp for mRNAs) extracted. After a re-PCR with Solexa Fusion Primers the desired bands on an agarose gel (miRNA, 150 bps; mRNA, smear ~200 bps) were again extracted, quality checked on an Agilent Bioanalyzer.

**HITS-CLIP—sequencing and alignment statistics**. Two libraries, mRNA- and miRNA-enriched, were sequenced to 100 bp on a HiSeq 2500. Two processing pipelines were used. The first considered only the nonchimeric reads in that the alignment program, bowtie, was looking for end-to-end alignments and did not allow significant parts of the read to "dangle". The second, applied to just the mRNA library used STAR to align to the transcriptome and did allow dangling ends. Dangling ends were then selected for lengths consistent with mature miRNAs and aligned to miRNA hairpins.

**HITS-CLIP—nonchimeric statistics**. The nonchimeric pipeline trims adapter sequence using a program written in the Kaestner lab[63]. There was some evidence of double ligation due to the enrichment for chimeric reads. We trimmed adapters from the 3′-end that (partially) match GTGTCAGTCACTTCCAG or TGTCAGTCACTTCCAG. We kept sequences that were at least 16 bp long. After trimming reads were aligned to the human genome, human RefSeq transcripts, and human miRNA hairpin sequences using bowtie allowing for multiple alignments. The mRNA library is enriched for long fragments, many of which span introns or are chimeric, so the genomic alignment rate is lower than the miRNA library. RefSeq alignment for the mRNA fraction is also lower due to the chimeric fragments. The miRNA fraction of the miRNA library is relatively high. Roughly, 81 million mRNA fragments and 68 million miRNA reads were used in the following steps. We identified locations of RISC occupancy on RefSeq transcripts as follows: the alignments were processed to count the number of times each position in the transcripts occurred at the start of an alignment. Then these weighted positions were clustered (from heavy to light) into bins of 10 bp to create the 5′-ends of the RISC complex footprints. We identified 265,406 footprints on 28,693 transcripts. To quantify miRNA occupancy in the RISC complex, we counted the number of read that overlapped with the annotated locations (miRbase v20) on the miRNA hairpins (Table 1).

**miRNA expression analysis (NGS)**. For high-throughput sequencing of miRNAs, total RNA of four-pooled samples of activated CD4$^+$ T cells from children with or without ongoing islet autoimmunity was extracted using the miRNeasy Micro Kit (Qiagen). cDNA libraries were obtained using the NEBnext Multiplex Small RNA Library Prep Set (New England Biolabs) according to the manufacturer's protocol. Sequencing was performed on a HiSeq2000 (Illumina) with 50 bp single end reads using Illumina reagents and following the manufacturer's instructions.

**NGS data processing and statistical analysis**. Unwanted adaptor sequences were trimmed from small RNA reads using BTrim[64] and quality of sequencing was assessed for trimmed read data with a mean phred quality score of 38, referring to a base call accuracy of 99.99%. Read data was filtered of unwanted RNA fragments by mapping on rRNA, tRNA, snRNA, and snoRNA sequences obtained from the Rfam database using bowtie[65]. Remaining reads were then mapped on mature human miRNA sequences obtained from mirBase (release 20)[66] and summed up to read count lists using SAMTools. mRNA read data was processed comparably without unnecessary trimming and filtering. Raw read data was mapped on the human genome (build 37.2) using a gapped alignment for paired end data with bowtie2[67]. Finally, read count lists were created by HTSeqcount 47[68]. Differential expression of miRNA was evaluated using DESeq[69], handling size factor correction and normalization.

**Isolation and analysis of miRNAs**. SmallRNAs/miRNAs were isolated using the miRNeasy Micro Kit (Qiagen). RNA concentration and purity were determined by nanodrop (Epoch, Biotech). For cDNA synthesis, the Universal cDNA Synthesis Kit II (Exiqon) was used according to the instructions. qPCR was performed using the ExiLENT SYBR Green PCR Master Mix (Exiqon) in combination with miR-CURY LNA primers for miR142-3p. For normalization, miRCURY LNA primers for the housekeeper 5s rRNA were used (Exiqon). For primer sequences see Supplementary Table 1. The reaction was performed on a CFX96 real time system (Biorad).

**Isolation and analysis of mRNAs**. mRNAs were isolated using the miRNeasy Micro Kit (Qiagen). cDNA synthesis was performed with the iScript cDNA Synthesis Kit (Biorad). For qPCR the SsoFast Evagreen Supermix (Biorad) and self-designed gene-specific primers were used. For normalization, the primers for the housekeeping gene Histone were used. For primer sequences see Supplementary Table 1. The reactions were performed on a CFX96 Real Time System (Biorad).

**Methylation analysis**. Up to 2000 CD4$^+$ T cells were subjected to a combined sample lysis and bisulfite conversion using the EZ DNA Methylation-Direct Kit (Zymo Research) according to the manufacturer's instructions. For bias-controlled quantitative methylation analysis, a combination of methylation-sensitive high resolution melting (MS–HRM) and subsequent pyrosequencing was performed. Utilizing the PyroMark Assay Design Software 2.0 (Qiagen), PCR primers and the according sequencing primers were designed to cover the area of differential methylation in the first Foxp3 intron initially reported by Baron et al.[23]. For primer sequences see Supplementary Table 1. MS–HRM was performed using the Sensi-FAST HRM Kit (Bioline) and the CFX96 real time system (Biorad). Pyrosequencing was performed on the PyroMark Q24 system (Qiagen) using PyroMark Gold Q24 Reagents (Qiagen) and following the manufacturer's instructions.

**Human in vitro Treg induction by limited TCR stimulation**. Human naive CD4$^+$ T cells (CD3$^+$, CD4$^+$, CD45RA$^+$, CD45RO$^-$, CD127$^+$ and CD25$^-$) were sorted with the BD FACS Aria III for purity and cultured (10,000 or 100,000 / well) in a 96-well plate precoated with 5 µg/ml anti-CD3 and 5 µg/ml anti-CD28 antibody with additional IL2 (100 U/ml). Limited TCR stimulation was achieved by pipetting the cells into uncoated wells, after 18 h, where they were cultured for additional 36 h without further TCR stimulation. Treg induction efficiency was measured using flow cytometry by analyzing Foxp3 expression in CD25$^+$CD127$^-$CD4$^+$ T cells.

**Murine in vitro Treg induction by limited TCR stimulation**. Murine naive CD4$^+$ T cells (CD4$^+$, CD25$^-$, and CD44$^-$) were sorted with the BD FACS Aria III for purity and cultured (10,000 or 100,000 / well) in a 96-well plate pre-coated with 5 µg/ml anti-CD3 and 5 µg/ml anti-CD28 antibody with additional IL2 (100 U/ml). Limited TCR stimulation was achieved by pipetting the cells into uncoated wells after 18 h, where they were cultured for additional 36 h without further TCR stimulation. Treg induction was measured by flow cytometry by analyzing Foxp3 expression in CD25$^+$CD4$^+$ T cells.

**Nanoparticles**. The nanoparticles were prepared by an emulsion-diffusion-evaporation method, first described by Kumar et al.[70] with slight modifications. The emulsion was prepared using a Harvard syringe pump for controlled dropping speed. Particles were passed through a sterile filter after preparation. The fluorescent-labeled particles were prepared with a poly(lactic-co-glycolic acid) (PLGA$^-$) fluoresceinamine (FA) conjugate. Particles were characterized with Dynamic light scattering (Zetasizer Nano ZS, Malvern). Chitosan PLGA nanoparticles and FA-labeled nanoparticles (in brackets) had a mean hydrodynamic

### Table 1

| Library | Raw reads | Trimmed reads | Genomic (%) | RefSeq (%) | RefSeq N | miRNA (%) | miRNA N |
|---|---|---|---|---|---|---|---|
| mRNA | 236,827,218 | 205,594,509 | 45.8 | 39.6 | 81,389,124 | 15.5 | |
| miRNA | 172,094,624 | 140,008,888 | 79.1 | 74.6 | | 48.6 | 67,980,650 |

diameter of $146.7 \pm 0.8$ nm ($152.8 \pm 1.2$ nm), polydispersity index $0.068 \pm 0.009$ ($0.056 \pm 0.007$) and a zeta potential of $+29.6 \pm 0.3$ mV ($+29.6 \pm 0.7$ mV).

**Application of miR142-3p inhibitor and mimic**. Chitosan-coated PLGA nanoparticles were loaded with the inhibitor or mimic (miRCURY LNA miRNA mimic/inhibitor, Exiqon) at a weight ratio of nanoparticles:inhibitor/mimic of 50:1 and incubated at room temperature for 30 min with gentle shaking. The loaded nanoparticles were added to the wells of a polyclonal Treg induction assay at a final concentration of 6 pmol/well (human) or 9 pmol/well (murine). As a control, nanoparticles loaded with miRNA mimic/inhibitor controls (Exiqon) were added to the cultures. For inhibitor/mimic sequences see Supplementary Table 1.

**3′ UTR luciferase reporter assay**. HEK-293 cells were cotransfected with a dual luciferase plasmid containing the wild-type or mutated full-length 3′ UTR from human TET2 (RefSeq NM_001127208.2) and a miR142-3p mimic (10 pmol/well) at 10,000 cells per well in a 96-well plate using Lipofectamine 3000 (Thermo Fisher Scientific) for 24 h. Luminescence was measured with the Dual Luciferase Reporter Assay Kit (Promega) following the manufacturer's protocol. The ratio of Firefly over Renilla luminescence was determined and compared to the transfection control. The mutations were introduced in the predicted miR142-3p-binding sites[45] in the TET2 3′ UTR (position 4135-4141 and 5392-5398) using site-directed mutagenesis by PCR. For primer sequences see Supplementary Table 1.

**miR142-3p activity assay**. A miR142-3p activity sensor plasmid was constructed by inserting a double-stranded oligonucleotide containing the miR142-3p target sequences (miR142-3p reverse complement, with a central bulged mismatch) into the 3′ UTR of a dual luciferase reporter plasmid. Jurkat T cells were cotransfected with the miR142-3p activity sensor plasmid and a miR142-3p inhibitor (5 pmol/well) at 50,000 cells per well in a 96-well plate, using Attractene transfection reagent (Qiagen) for 24 h. Luminescence was measured with the Dual Luciferase Reporter Assay Kit (Promega) following the manufacturer's protocol. The ratio of Firefly over Renilla luminescence was determined and compared to the transfection control.

**Application of Tet2 siRNA**. Tet2 siRNA (Silencer Select Pre-Designed siRNA Tet2, Ambion) or control siRNA (Silencer Select Negative control no.1, Ambion) were combined with siRfficient transfection reagent (MBL Life Science) following the manufacturer's protocol. A total of 20 pmol siRNA were added to murine and human Treg induction assays. For siRNA information see Supplementary Table 1.

**3T3 fibroblasts**. 3T3 fibroblasts were seeded in 96-well plates at a density of 10,000 cells/well. After 24 h the cells were transfected with 10 pmol of miR142-3p mimic or control mimic using Lipofectamine RNAiMAX Reagent, following the manufacturer's instructions.

**Restimulation assay**. After in vitro Treg induction using naive CD4[+] T cells from BALB/c Foxp3GFP reporter mice in the presence of a miR142-3p inhibitor, Foxp3[+]CD25[+]CD4[+] Tregs were sort-purified. The Tregs were then stimulated for 30 h with anti-CD3 and anti-CD28 antibodies in the presence of a miR142-3p inhibitor or a control inhibitor. Maintenance of the Treg phenotype was measured by flow cytometry analysis of Foxp3 expression in CD25[+]CD4[+] T cells.

**Engraftment of NSG mice with human PBMCs**. Murine MHCII deficient, HLA-DQ8 transgenic NOD.Cg-Prkdc[scid] Il2rg[tm1Wjl] (NSG) mice were reconstituted with PBMCs from an HLA-DQ8[+] donor with T1D. A total of $5 \times 10^4$ PBMCs per mouse were injected intravenously in 50 ml PBS into the retro orbital sinus without prior conditioning by irradiation or busulfan treatment. To avoid sex incompatibilities, the sex of the NSG-HLA-DQ8 mice for reconstitution was chosen in accordance with the blood donor.

**In vivo miR142-3p inhibitor application**. A miR142-3p inhibitor (Inhibitor Probe mmu-miR-142-3p, Exiqon) was injected i.p. into IAA[+] NOD mice or NSG humanized mice at 10 mg/kg every other day for 14 days. On day 15 Treg frequencies were analyzed in lymph nodes. CD25[high]Foxp3[high] Tregs were sort-purified for methylation analysis. Pancreata were embedded for cryosections and analysis of pancreas pathology. For miRNA inhibitor localization experiments, a FAM-labeled miR142-3p inhibitor (mmu-miR-142-3p inhibitor 5′FAM, Qiagen) was injected i.p. into BALB/c or NSG-humanized mice at 10 or 20 mg/kg. The FAM-labeled miR142-3p inhibitor was detected via flow cytometry after 4 h, 24 h or at the end of an application period of 14 days every other day.

**Immunofluorescence staining of NOD pancreata**. Immunofluorescence staining was carried out after acetone fixation, permeabilization, and blocking with Avidin/Protein blocking together with 5% goat serum using rabbit-anti-mouse insulin antibodies (Cell Signaling, 1:100) and donkey-anti-rabbit[AlexaFluor647] antibodies (Dianova, 1:400). For CD3 staining, arm.hamster anti-mouse antibodies (BD, 1:50) were used, followed by goat-anti-arm.hamster antibodies conjugated with

Dylight[488] (Dianova, 1:100). For Tet2 staining, rabbit-anti-Tet2 antibodies (ABclonal, clone A5682, 1:25) was used with biotinylated horse-anti-rabbit antibodies (Vector, 1:100) combined with SA[Dylight 549]. For Foxp3 staining, cells were incubated with rat-anti-mouse Foxp3 antibodies (eBioscience, clone FJK-16s, 1:50) and biotinylated goat anti-rabbit (BD), combined with SA[Dylight 549]. Nuclei were counterstained with DAPI (Diavona). For Tgfbr1 staining, rat-anti-mouse/human Tgfbr1 antibodies (R&D, clone 141231, 1:25) was used, followed with biotinylated goat-anti-rat antibodies (1:250) combined with SA[Dylight 549] (Diavona, 1:200). Negative control slides were incubated with secondary antibodies. Cells were analyzed by confocal microscopy (Zeiss LSM700).

**Histopathology of NOD pancreata**. Pancreata of NOD mice were embedded with Tissue-Tek O.C.T. Compound and frozen on dry ice. Serial sections were stained with hematoxylin and eosin. Insulitis scoring was performed as follows: 0: intact islets/no lesions; 1: peri-islet infiltrates; 2: < 25% islet destruction; 3: > 25% islet destruction; 4: complete islet destruction. Investigators were blinded for group allocations.

**Statistical analysis**. Results are presented as mean and s.e.m or as percentages, where appropriate. For normally distributed data, Student's $t$-test for unpaired values was used to compare means between independent groups and the Student's $t$-test for paired values was used to compare values for the same sample or subject tested under different conditions. For multiple testing, ordinary one-way analysis of variance (ANOVA) and Tukey's multiple comparisons test were used. For all tests, a two-tailed $P$ value of < 0.05 was considered to be significant. Statistical significance is shown as $* = P < 0.05$; $** = P < 0.01$; and $*** = P < 0.001$, or not significant (NS) $P > 0.05$. Analyses were performed using the program GraphPad Prism 7 (La Jolla, CA).

**Reporting summary**. Further information on research design is available in the Nature Research Reporting Summary linked to this article.

## Data availability
The authors declare that all data supporting the findings of the study are available within the article and its Supplementary Material or from the corresponding author upon reasonable request. The source data underlying Fig. 1g, h, 2a, c–f, 3a, c–e, 4b, d, f, 5a–j, 6a, c, e–g, 7b, c, e, g, h and Supplementary Figures 3, 6a–d, 7a–c, 8b, d, 11a, b, 12a–f, 13b–d, 14d, f, 15a, b, 16a–c, and 17c are provided as a Source Data file. HITS-CLIP library sequencing data have been deposited to GEO database under the accession number GSE124264. miRNA sequencing data have been deposited to GEO database under the accession number GSE140064.

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

## Acknowledgements

We thank C. Matzke for the support with the insulin autoantibody analyses, T. Telieps for the support with plasmid construction, S. Fiedler and S. Popp for the support with

immunohistochemistry, and K. Goth for the support with the miR142$^{-/-}$ mice. K.G. is supported by the grant J-50 (IZKF). B.W. is supported by Deutsche Forschungsgemeinschaft (DFG) KFO-257 (grant WE 4656/2) and DFG-CRC1811 (B02). C.D. is supported by a Research Group at Helmholtz Zentrum München, the German Center for Diabetes Research (DZD) and through a membership in the CRC1054 of the Deutsche Forschungsgemeinschaft (B11). K.H.K. is supported by the National Institute of Health Grant (UC4DK112217) and through a Hans Fischer Senior Fellowship from the Technische Universität München. This work was supported by grants from the Juvenile Diabetes Research Foundation (JDRF 2-SRA-2014-161- Q-R [C.D., A.G.Z.], JDRF 17−2012-16 [A.G.Z.], JDRF 6-2012-20 [A.G.Z]), the Kompetenznetz Diabetes mellitus (Competence Network for Diabetes mellitus), funded by the Federal Ministry of Education and Research (FKZ 01GI0805-07, FKZ 01GI0805) and the German Center for Diabetes Research (DZD).

## Author contributions

M.G.S. performed experiments, analyzed and interpreted data, and wrote the manuscript. I.S. performed experiments. A.M.Z. performed HITS-CLIP experiments. J.S. supported next-generation sequencing (NGS) and HITS-CLIP analyses. S.B. supported experiments with loss-of-function models. R.M. supported luciferase experiments. V.K.S. performed experiments. K.G. and B.W. performed immunofluorescent stainings. A.-G.Z. supported the study design, and is the principal investigator of the BABYDIAB, DiMelli, and the Munich Bioresource studies, which provided blood samples for the study. K.H.K. advised on HITS-CLIP and miRNA analyses and edited the manuscript. C.D. conceptualized and designed the study, analyzed and interpreted data, and wrote the manuscript.

## Competing interests

The authors declare no competing interests.
