## [Peer Review File · Nature Communications]

Editorial Note: Parts of these reports have been redacted.

Reviewers' comments:

Reviewer #1, expert in Tet family proteins (Remarks to the Author):

Comments to authors

In this study, Scherm et al. claimed that miRNA142-3p/Tet2 signaling impairs Treg induction and stability in type 1 diabetes (T1D) autoimmunity. The authors proposed an important scientific question to further understand molecular mechanisms that may trigger autoimmune progression in T1D. Also, the authors decipher epigenetic interactions as a mechanism to regulate Tregs. It appears that the authors have provided convinced experimental data for the conclusion. However, lack of novelty is a major drawback.

Major Concerns

1. The roles of microRNAs in regulating Treg function (PMIDs: 27438767, 26165721) and T1D islet autoimmunity (PMID: 27791035) have been reported. Specifically, miR-142-3p has been documented in Treg functional modulation in autoimmune diseases (PMIDs: 29459719, 23650616). These previous findings dampen the novelty of this study. Although the authors proposed a target link between miR-142-3p and Tet2, representing an epigenetic link between microRNA and DNA-methylation regulations, the functional peculiarity of this kind of regulation in autoimmunity progression is not clearly dissected in this manuscript.
2. Tet2-mediated Foxp3 regulation for Treg differentiation and immune homeostasis has also been reported (PMID: 26275994). While the importance of Tet2 in T1D autoimmunity and progression is not well understood, the authors failed to provide experimental evidence for understanding in vivo functional role of Tet2 downstream miR-142-3p regulation.
3. In Fig. 7, the authors applied systemic delivery of the microRNA inhibitor to treat islet autoimmunity in vivo. Nevertheless, the authors did not provide evidence for miR-142-3p knocking-down efficacy in islet tissue. Furthermore, therapeutic effects of miR-142-3p inhibition on NOD diabetic phenotypes are lacking.
4. For statistical analysis, Student's t tests are not appropriate for multiple group comparisons. Furthermore, if standard error of the mean (s.e.m) is used for representing the error, the authors should state clearly how the experiments were repeated: e.g. whether biological or technical replicates were involved, and whether average values of each repeat of experiments were used. Otherwise, standard deviation (s.d) should be used for representing the error.

Minor concerns

1. In Fig. 1, whether miR-142-3p expression is correlated with T1D disease severity remains to be revealed.
2. In Fig. 5, luciferase reporter assay should be used to confirm targeting of Tet2 by miR-142-3p.
3. Scale bars are missing in Fig. 6 and Fig. 7 islet images.

Reviewer #2, expert on micro-RNA and autoimmune disease (Remarks to the Author):

Scherm et al link miR-142 to Tet2 and the regulation of Treg differentiation in autoimmune disease. The subject matter and proposed "axis" of regulation that could contribute to loss of immunologic self-tolerance are of high interest. However, the presented evidence do not strongly support the existence and importance of this 'axis' due to numerous technical issues, incomplete data presentation, and the lack of validating data for several key techniques.

1. More information and data are needed to assess the HITS-CLIP experiment. How many mapped reads were obtained for mRNA fragments? Most protocols generate miRNA and protected mRNA fragments of similar small size in the same library. Why were larger fragments collected here, what was the size distribution of mapped reads, and how was the validity of the data assessed? Both global

analysis of concordance with target predictions for abundant miRNAs and examples of individual genes should be shown. The one provided example (a short region of Tet2 in Fig. 5a) indicates rather low read density and provides no further information about Ago2 binding to the transcript, which is a known target of miR-29. As the data are currently presented, they do not provide convincing support for miR-142 targeting of Tet2.

2. Labels for Fig. 1d are confusing. Does miR-142 refer to both miR-142-3p and miR-142-5p? What does miR-let-7bhg denote that is distinct from miR-let-7b? (Also, why miR-let-7 instead of just let-7?) What about miR-17hg – does that refer to the entire miRNA cluster that includes miR-17?

3. Fig. 1F should indicate the variability in replicates (n=4 per group), and indicate whether and how read counts were normalized for quantification. Control miRNAs should be shown, and full sequencing data including sufficient information to assess data quality should be provided.

4. The meaning “activated phenotype” should be specified for both mouse and human T cells for Fig. 1g-h. The peripheral blood usually contains very very few activated lymphocytes.

5. Functional blockade of miR-142 activity using nanoparticle delivery of inhibitors should be validated using an activity sensor (e.g. luciferase or fluorescent protein construct with miR-142 bindings sites) rather than PCR. PCR can be affected by inhibitor binding after lysis, and does not provide information about the fraction of cells that receive effective inhibition. It may be sufficient to cite a prior publication in which this technique for delivery has been fully assessed.

6. Why should miR-142 inhibitor and mimic have no effect on sorted naïve T cells from diabetic NOD mice, and in human subjects with recent onset or established T1D? Does this indicate some kind of extrinsic regulation of Treg differentiation that overrides the effect of miR-142 in diabetic individuals?

7. The presented data are not sufficient to assess the affect of miR-142 on Treg stability. The number of CD25+ and FoxP3+ cells in restimulated induced Treg cultures could be affected by differences in the growth and survival of the mixture of Tregs and other T cells in the culure, and/or it could be affected by changes in Treg stability. To test for the latter, FoxP3 lineage markers are needed (e.g. FoxP3;Rosa26-lsl-fluorescent protein mice, which are widely available).

8. Data availability section is incomplete. HITS-CLIP data “will be provided shortly”, and there is no mention of NGS miRNA sequencing data.

9. Fig. 5a is unclear regarding chimeric reads. Does the bottom panel indicate the position of a single miR-142 chimeric read, while the middle panel indicates the read density for all chimeric reads? Also, where in this region are there predicted miR-142 binding site(s)? Are they conserved in mice? This is important since many key experiments were performed in mice. Miranda is a very permissive target prediction algorithm with many more false positive predictions than other algorithms such as Targetscan.

10. Further experiments are needed to validate Tet2 as a miR-142 target. Reciprocal miRNA and putative target gene expression are consistent with direct targeting, but could easily be due to indirect effects of the miRNA on other genes. This is especially true for inferences about anti-correlated miRNA and putative target gene expression patterns and related biology in vivo (e.g. Fig. 6). Is the 3' UTR of Tet (or the putative region bound by Ago2:miR-142 complexes) responsive to miR-142 inhibitor and/or mimics in 3' UTR reporter assays?

11. Do the inhibitors affect miR-142 activity in T cells in the mice that are dosed systemically? Or might they be working by alternate means (in other cells to which it is easier to deliver inhibitors, or by some indirect mechanism?). This type of delivery system requires rigorous validation, as previous efforts to block miRNAs in lymphocytes in vivo have not been so successful. Much better in vivo loss of

function experiments could be performed using genetic miR-142-deficiency (e.g. Sun et, JCI 2015).

Reviewer #3, expert on Treg (Remarks to the Author):

In the submitted paper the authors unveil a pathway that links miR142-3p expression in T cells to a reduction in Tet2 expression and hence Treg instability. The studies are based on type 1 diabetes in both patients and mice, and suggest that increased miR142-3p may play a pathogenic role in diabetes development. In humans, the relationship between the different observations is induced, while in mice it is formally demonstrated up to the point of T cell infiltrate (while impacts on diabetes development are not shown, possibly because such experiments may be impractical using the tools generated here). Overall, while the paper is written a bit roughly, the data are novel and potentially important.

Major points:

- The higher number of Tregs with the miR-142-3p inhibitor could be increased stability, as stated in the text, but altered proliferation or apoptosis of either the Treg or Tconventional fractions was not excluded in Figure 4b.
- The studies on Foxp3 methylation patterns in Figure 4 should have been performed with and without miR-142-3p over-expression and inhibition. In the current format, it is not obvious what they contribute to the study.
- Are the changes to Foxp3 methylation in T1D patients compared to healthy control due to: 1) more contamination of transiently Foxp3+ T cells in the Treg isolation; 2) more Nrp1- pTreg within the Treg pool; or 3) partial methylation of tTreg?
- The suggested direct link between miR-142-3p, CNS3 methylation and Tet2 could be further tested in human cells by Tet2 siRNA

Minor points

- It is stylistic, but I would suggest that the insertion of number, standard deviation and p values into the text reduces readability, while not providing information that is not already present on the figure. In addition, the paper would be well served by implementation of the standard paragraphing format
- In Figures 4d and 4f, are the remethylated CNS2 Tregs at later time points still Foxp3+? The co-expression of Treg markers is stated in 4c and 4e, but a proper time course with the key markers would be more useful
- In Figure 6c, are Tet2+ T cells reduced as a proportion of CD3 T cells? Or are just total T cell numbers down, which means Tet2+ cells will be down as a result?

Point by Point Response

Nature Communications Manuscript number NCOMMS-18-24478

miRNA142-3p/Tet2 signaling impairs Treg induction and stability in type 1 diabetes autoimmunity

Reviewer #1, expert in Tet family proteins (Remarks to the Author):

Comments to authors

In this study, Scherm et al. claimed that miRNA142-3p/Tet2 signaling impairs Treg induction and stability in type 1 diabetes (T1D) autoimmunity. The authors proposed an important scientific question to further understand molecular mechanisms that may trigger autoimmune progression in T1D. Also, the authors decipher epigenetic interactions as a mechanism to regulate Tregs. It appears that the authors have provided convinced experimental data for the conclusion. However, lack of novelty is a major drawback.

Response: We thank the reviewer for the constructive criticism related to our findings presented in this manuscript and the positive feedback highlighting the importance of the scientific question and the convincing experimental data. We are also grateful for the constructive comments regarding novelty, which turned out to be extremely helpful in revising this manuscript and clarifying how our novel data and mechanistic insights add to the already existing studies mentioned by the reviewer.

Major Concerns

1. The roles of microRNAs in regulating Treg function (PMIDs: 27438767, 26165721) and T1D islet autoimmunity (PMID: 27791035) have been reported. Specifically, miR-142-3p has been documented in Treg functional modulation in autoimmune diseases (PMIDs: 29459719, 23650616). These previous findings dampen the novelty of this study. Although the authors proposed a target link between miR-142-3p and Tet2, representing an epigenetic link between microRNA and DNA-methylation regulations, the functional peculiarity of this kind of regulation in autoimmunity progression is not clearly dissected in this manuscript.

Response: We thank the reviewer for mentioning these important studies and we agree that they provided considerable insight into the roles of miRNAs in regulating Treg function and T1D islet autoimmunity. However, below we would like to respectfully clarify how our novel data is relevant for the field and how it integrates in the existing literature. In addition, we have performed a series of additional *in vitro* and *in vivo* studies including cellular and molecular dissection of the functional implication of the identified miR142-3p/Tet2/Foxp3 axis in the regulation of islet autoimmunity. Specially, we have included *in vivo* data in the setting of islet autoimmunity upon *in vivo* inhibition of miR142-3p demonstrating enhanced CD3⁺Tet2⁺T cells in the pancreas accompanied by improved stability of the Treg phenotype (see Fig. 8f-h). We have provided proof of principle studies to show miR142-3p inhibition in the setting of established T1D using PBMCs from individuals with T1D for the reconstitution of humanized mice, likewise in these humanized mice we identify increased numbers of CD3⁺Tet2⁺T cells in the pancreas following blockade of miR142-3p (now in Supplementary Fig. 11). In line with these findings, we have now also included novel analyses using T cells from individuals with recent onset of clinical T1D which presented with significantly reduced numbers of CD3⁺Tet2⁺T cells (now in Fig.7d and e). Of note, we have now performed novel experiments in the context of very early onset islet autoimmunity in NOD mice below 30 days of age. Importantly here, the identified increased Foxp3 CNS2 DNA methylation in NOD mice below 30 days of age with a very early onset of autoimmunity suggests a potential causative role of miR142-3p/Tet2 signaling in promoting autoimmune activation and progression (presented in Supplementary Fig. 9c). For further details please refer to the manuscript.

Below we discuss the indicated literature in the context of our novel findings:

PMID 27438767 addresses how inflammatory cues induce expression of miR-17 which targets Eos, a co-regulatory molecule of Foxp3, diminishing the suppressive capacity of Tregs. While these findings are of considerable importance for the understanding of Treg function, the paper does not address the contribution of miRNAs to impaired Treg induction from naive T cells and their stability during onset of islet autoimmunity, which is the major focus of our manuscript where we provide mechanistic evidence for a direct targeting of Tet2 by miRNA142-3p. Specifically, we find that aberrant miR142-3p expression in CD4⁺ T cells acting via Tet2 repression functions as one mechanism by which dysregulated DNA methylation at the Foxp3 locus mediates impaired Treg homeostasis, and consequently contributes to autoimmune activation.

PMID 26165721 identifies miR-31 as a regulator of Gprc5a (retinoic acid-inducible protein 3) expression, generation of peripheral Tregs and experimental autoimmune encephalomyelitis (EAE) severity. These findings are of interest for the regulation of Treg induction and its contribution to autoimmunity. However, the mechanisms described in our manuscript, in particular epigenetic remodeling of the Foxp3 CNS2 during onset of islet autoimmunity via miR142-3p/Tet2 signaling and the link to impaired Treg induction and stability are not addressed.

PMID 27791035 deals with TFH precursor cells, their role in the onset of islet autoimmunity and how their frequency is controlled by miR-92a. Even though this publication shows miRNA regulation of a T cell subset which is important for autoimmune pathogenesis, it does not address the crucial role of Tregs and specifically their induction and stability in autoimmunity.

PMID 29459719 identifies miR142-3p as a regulator of ATG16L1 and addresses its role in regulating autophagy, proliferation and function in thymic-derived Tregs as well as the implication of these findings for graft-versus-host disease. However, the manuscript does not address the impact of miR142-3p to de novo Treg induction from naive CD4⁺T cells and stability of induced Tregs during the onset of islet autoimmunity. Furthermore, the identification of Tet2 as a direct target of miR142-3p and the impaired demethylation of the Fox3 CNS2 are important and novel findings, contributing to our understanding of regulation of Treg induction and stability in autoimmunity.

PMID 23650616 identifies miR142-3p as a regulator of GARP expression on CD25⁺ CD4⁺ T cells and, as a result, their expansion in response to activation, which provides insight into the control of Treg expansion. While these findings are of importance for the field, they do not address the role of miR142-3p for Treg induction or stability as well as the critical role of the miR142-3p/Tet2 axis for the onset of islet autoimmunity.

2. Tet2-mediated Foxp3 regulation for Treg differentiation and immune homeostasis has also been reported (PMID: 26275994). While the importance of Tet2 in T1D autoimmunity and progression is not well understood, the authors failed to provide experimental evidence for understanding in vivo functional role of Tet2 downstream miR-142-3p regulation.

Response: We thank the reviewer for referring to this important study and we agree that it is of considerable importance for the understanding of Tet1- and Tet2-mediated regulation of Foxp3 expression, Treg differentiation and immune homeostasis. In addition to these important discoveries our novel findings provide mechanistic evidence for the role of Tet2 mediated Foxp3 CNS2 demethylation in the context of Treg induction from naive T cells in the setting of autoimmunity. Importantly, to dissect the potential causative contribution of impaired Treg stability in promoting autoimmune progression we now assessed Foxp3 CNS2 DNA methylation in NOD mice below 30 days of age with a very early onset of IAA⁺ positivity. Of note, these very young IAA⁺ NOD mice presented with distinctly increased Foxp3 CNS2 DNA methylation when compared to non-autoimmune prone BALB/c mice (presented in Supplementary Fig. 9c). These novel findings thereby support the concept that early onset Treg instability can be involved in promoting autoimmune activation and progression. In addition to these novel insights on Treg stability impairments in impacting autoimmune activation and progression we provide mechanistic evidence for a miRNA142-3p-Tet2 signaling axis and integrate their contribution in aberrant Treg induction and stability. Specifically, we identified miR142-3p, which is upregulated in autoimmunity, as an upstream regulator of Tet2, linking differential miRNA expression to reduced Tet2 abundance, incomplete Foxp3 CNS2 demethylation and impaired Treg induction and stability in autoimmunity.

In addition, we have performed a series of novel *in vitro* and *in vivo* experiments including loss of function models. Specifically, employing a combination of HITS-CLIP and various molecular and cellular approaches, including miR142-3p modulation and loss of function models, we have demonstrated that miR142-3p activity is linked to Tet2 abundance, Foxp3 CNS2 methylation, affecting Treg induction and stability. Concordantly, the inhibition of miR142-3p in NOD mice with ongoing islet autoimmunity significantly reduced Foxp3 CNS2 methylation in Tregs, increased the number of pancreatic Tregs and improved islet infiltration (now in novel Fig. 8). In addition we provide novel data showing that miR142-3p inhibition *in vivo* also leads to higher numbers of Tet2⁺ T cells in the pancreas (novel data in Fig. 8f and g) directly linking miR142-3p to Tet2 expression, epigenetic remodeling in Tregs and islet autoimmunity. In addition, we performed novel studies using humanized NSG mice. Humanized mice are immunodeficient mice that after reconstitution with human hematopoietic cells or tissues do develop a human immune system with a highly diverse TCR repertoire. These mice permit the assessment of human T cell responses *in vivo*. Specifically, we made use of the murine MHCII deficient, HLA-DQ8 transgenic NOD.Cg-Prkdc^{scid} Il2rg^{tm1Wjl} (NSG) mouse model¹. To assess a potential relevance of miR142-3p inhibition in T cells from T1D

phenotypes we employed PBMCs from individuals with recent onset T1D for a pilot experiment to reconstitute NSG mice for the establishment of humanized mice. Application of a miR142-3p inhibitor to these humanized NSG mice showed a trend towards increased frequencies of peripheral CD127^{low}CD25^{hi} Tregs and a significant increase in Tet2 expression in pancreatic T cells (Supplementary Fig. 11).

3. In Fig. 7, the authors applied systemic delivery of the microRNA inhibitor to treat islet autoimmunity in vivo. Nevertheless, the authors did not provide evidence for miR-142-3p knocking-down efficacy in islet tissue. Furthermore, therapeutic effects of miR-142-3p inhibition on NOD diabetic phenotypes are lacking.

Response: We thank the reviewer for this comment. We used an LNA in vivo miRNA inhibitor which has been shown to be effective in a broad range of tissues. When delivered systemically, LNA miRNA inhibitors distribute broadly into most tissues, including hematopoietic tissues such as lymph nodes, spleen and bone marrow ². We agree that besides the accumulation in the tissues, the knocking-down efficacy in the relevant tissue is highly relevant. Here, the efficient knock-down of miR142-3p using systemic administration of a LNA miRNA inhibitor has been shown in splenocytes ^{3,4}. A list of studies using in vivo miRNA inhibition with LNA miRNA inhibitors can be found here: <http://www.exiqon.com/ls/Documents/Scientific/mirna-inhibition-publications.pdf>.

To further strengthen the evidence for successful delivery of the inhibitor to pancreatic tissue we performed novel experiments and upon application of the miR142-3p inhibitor validated the expression of a well-established miR142-3p target Tgfbr1 ⁵ directly in pancreatic T cells. Specifically, we demonstrated delivery to pancreatic tissue and local immune cells by showing increased expression of Tgfbr1 in pancreatic T cells when compared to NOD mice that had received a control inhibitor (Supplementary Fig. 10a and b).

Regarding the therapeutic effects of miR142-3p on diabetic phenotypes we would like to respectfully draw the attention of the reviewer to the in vitro Treg induction data presented in Fig. 2 and 3 and the in vivo miR142-3p inhibition data shown in Fig. 8 of the manuscript. We showed that miR142-3p inhibition in vitro results in increased Treg induction efficacy in NOD mice with and without islet autoimmunity, while there was only a trend towards a comparable improvement in NOD mice with

established T1D. In human T cells miR142-3p inhibition improved Treg induction capacity in both subjects with recent onset of T1D and with long-term T1D. Here we wish to point to the differences of established T1D between NOD mice and humans. While in humans insulin replacement therapy controls blood glucose levels relatively well, in mice the lack of insulin can result in severe dysglycemia. We suggest that this difference also affects Treg induction which is in line with the finding that Treg induction capacity without any miRNA modulation differs significantly between IAA⁺ and diabetic NOD mice, but it does not between human subjects with recent onset and long-term diabetes. Furthermore, we showed that inhibition of miR142-3p in IAA⁺ NOD mice *in vivo* significantly improves islet autoimmunity, which is causative for the development of T1D. These improvements include reduced islet infiltration (Fig. 8a and b), lower IAA levels (Fig. 8c), increased abundance of Foxp3⁺ Tregs directly in the pancreas (Fig. 8d and e) and improved demethylation of the Foxp3 CNS2 in Tregs (Fig. 8h).

In addition we provide novel data showing that miR142-3p inhibition *in vivo* also leads to higher numbers of Tet2⁺ T cells in the pancreas (Fig. 8f and g) directly linking miR142-3p to Tet2 expression, epigenetic remodeling in Tregs and islet autoimmunity. In addition, we performed novel studies using humanized NSG mice. As outlined above humanized mice are immunodeficient mice that after reconstitution with human hematopoietic cells or tissues do develop a human immune system with a highly diverse TCR repertoire. Specifically, we made use of the murine MHCII deficient, HLA-DQ8 transgenic NOD.Cg-*Prkdc*^{scid} *Il2rg*^{tm1Wjl} (NSG) mouse model¹. To assess a potential relevance of miR142-3p inhibition in T cells from T1D phenotypes we employed PBMCs from individuals with recent onset T1D for a pilot experiment to reconstitute NSG mice for the establishment of humanized mice. Application of a miR142-3p inhibitor to these humanized NSG mice showed a trend towards increased frequencies of peripheral CD127^{low}CD25^{hi} Tregs and a significant increase in Tet2 expression in pancreatic T cells (novel data in Supplementary Fig. 11). Furthermore, we have now also included novel analyses using T cells from individuals with recent onset of clinical T1D which presented with significantly reduced numbers of CD3⁺Tet2⁺T cells (now in Fig.7d and e).

4. For statistical analysis, Student's t tests are not appropriate for multiple group comparisons. Furthermore, if standard error of the mean (s.e.m) is used for representing the error, the authors should state clearly how the experiments were repeated: e.g. whether biological or technical

replicates were involved, and whether average values of each repeat of experiments were used. Otherwise, standard deviation (s.d) should be used for representing the error.

Response: We thank the reviewer for this suggestion and agree on this point. We made changes in the manuscript, indicating the respective statistical test for multiple group comparison. In addition we specified how the experiments were repeated, as requested.

Minor concerns

1. In Fig. 1, whether miR-142-3p expression is correlated with T1D disease severity remains to be revealed.

Response: We thank the reviewer for this comment. To address the question whether aberrant miR142-3p/Tet2 signaling together with alterations in Treg induction and stability is correlated with T1D disease severity we performed novel studies where we specifically focused on early onset of autoimmunity. Specifically, to dissect the potential causative contribution of impaired Treg stability in promoting autoimmune progression and thereby early onset and severity of clinical disease we assessed Foxp3 CNS2 DNA methylation in NOD mice below 30 days of age with a very early onset of IAA⁺ positivity. Importantly, these very young IAA⁺ NOD mice presented with distinctly increased Foxp3 CNS2 DNA methylation when compared to non-autoimmune prone BALB/c mice. These novel findings are now presented in Supplementary Fig. 9c and support a potential causative role of miRNA142-3p/Tet2 signaling in promoting early onset Treg instability, autoimmune activation thereby impinging on progression and T1D disease severity.

2. In Fig. 5, luciferase reporter assay should be used to confirm targeting of Tet2 by miR-142-5p.

Response: We thank the reviewer for this suggestion. In order to confirm targeting of Tet2 by miRNA-142-3p we performed a series of novel mechanistic studies. Specifically, we performed luciferase reporter assays as requested. To this end, we co-transfected HEK-293 cells with a TET2 3'UTR reporter construct and a miR142-3p mimic and observed a lower Firefly/Renilla luminescence ratio compared to the transfection control. These novel data are now presented in Fig. 6e.

In line with the broad role of Tet2 in various cell types we first made use of the non-hematopoietic 3T3 fibroblast cell line, which was previously shown to lack miR142-3p expression almost completely ⁶. As a second approach we dissected the miRNA142-3p-Tet2 axis using T cells from miR142 knockout (miR142^{-/-}) mice ⁷. For further details please refer to the novel Supplementary Fig. 8.

Specifically, we transfected 3T3 fibroblasts with a miR142-3p mimic or a control mimic to introduce miRNA142-3p levels given the absence of endogenous miR142-3p expression. The introduction of miRNA142-3p expression in these cells following mimic application resulted in a significant decrease of Tet2 mRNA levels. These novel findings are now provided in Fig. 6f.

To further strengthen the mechanistic evidence for the direct miRNA142-3p-Tet2 relationship we used T cells from miRNA142 knockout animals. Stimulation of CD4⁺T cells from miRNA142 knockout mice showed significantly higher Tet2 mRNA levels when compared to T cells from miRNA142 competent mice, now presented in (Fig. 6g). The direct targeting of Tet2 by miRNA142 was also validated on the protein level. Specifically, T cells from miRNA142 deficient animals presented with significantly higher Tet2 protein expression following TCR stimulation when compared to T cells from miR142^{+/+} mice (Fig. 6h). These findings provide compelling evidence for direct targeting of Tet2 by miRNA142.

3. Scale bars are missing in Fig. 6 and Fig. 7 islet images.

Response: We thank the reviewer for this comment and apologize for the mistake. We have included scale bars in the micrographs in former Fig. 6 (currently Fig. 7) and former Fig. 7 (currently Fig. 8).

Reviewer #2, expert on micro-RNA and autoimmune disease (Remarks to the Author):

Scherm et al link miR-142 to Tet2 and the regulation of Treg differentiation in autoimmune disease. The subject matter and proposed “axis” of regulation that could contribute to loss of immunologic self-tolerance are of high interest. However, the presented evidence do not strongly support the

existence and importance of this ‘axis’ due to numerous technical issues, incomplete data presentation, and the lack of validating data for several key techniques.

Response: We thank the reviewer for the positive comments, constructive criticism and interest in our studies. We are specifically grateful for the suggestion of additional experiments, which turned out to be extremely helpful in improving our dataset and revising this manuscript. In order to address the reviewer’s concerns and strengthen the conclusions, we performed novel *in vitro* and *in vivo* experiments, reanalyzed data and provided additional information for clarification as requested.

1. More information and data are needed to assess the HITS-CLIP experiment. How many mapped reads were obtained for mRNA fragments? Most protocols generate miRNA and protected mRNA fragments of similar small size in the same library. Why were larger fragments collected here, what was the size distribution of mapped reads, and how was the validity of the data assessed? Both global analysis of concordance with target predictions for abundant miRNAs and examples of individual genes should be shown. The one provided example (a short region of Tet2 in Fig. 5a) indicates rather low read density and provides no further information about Ago2 binding to the transcript, which is a known target of miR-29. As the data are currently presented, they do not provide convincing support for miR-142 targeting of Tet2.

Response: We thank the reviewer for this important comment and the suggestion to provide additional information to strengthen our HITS-CLIP dataset and especially the chimeric reads. In accordance with the information provided here, we expanded the description of the HITS-CLIP analysis and specific requested details as mentioned above in the methods section of our manuscript. In addition we have included novel figures (Fig. 5) and supplementary figures (Supplementary Fig.6) to modify the presentation of the data accordingly in order to provide convincing support for miR142-3p targeting of Tet2. For further details please refer to the manuscript.

In relation to questions outlined above we also include additional specific information below:

HITS-CLIP – sequencing and alignment statistics

Two libraries, mRNA- and miRNA-enriched, were sequenced to 100bp on a HiSeq 2500. Two processing pipelines were used. The first considered only the non-chimeric reads in that the

alignment program, bowtie, was looking for end-to-end alignments and did not allow significant parts of the read to ‘dangle’. The second, applied to just the mRNA library used STAR to align to the transcriptome and did allow dangling ends. Dangling ends were then selected for lengths consistent with mature miRNAs and aligned to miRNA hairpins.

HITS-CLIP – non-chimeric statistics

The non-chimeric pipeline was described previously ⁸. The non-chimeric pipeline trims adapter sequence using a program written in the Kaestner lab. There was some evidence of double ligation due to the enrichment for chimeric reads. We trimmed adapters from the 3’ end that (partially) match GTGTCAGTCACTTCCAG or TGTCAGTCACTTCCAG. We kept sequences that were at least 16bp long. After trimming reads were aligned to the human genome, human RefSeq transcripts, and human miRNA hairpin sequences using bowtie allowing for multiple alignments. The mRNA library is enriched for long fragments, many of which will span introns or are chimeric, so the genomic alignment rate is lower than the miRNA library. RefSeq alignment for the mRNA fraction is also lower due to the chimeric fragments. The miRNA fraction of the miRNA library is relatively high. Roughly 81 million mRNA fragments and 68 million miRNA reads were used in the following steps. We identified locations of RISC occupancy on RefSeq transcripts as described previously ⁸. Briefly, the alignments were processed to count the number of times each position in the transcripts occurred at the start of an alignment. Then these weighted positions were clustered (from heavy to light) into bins of 10bp to create the 5’ ends of the RISC complex footprints. We identified 265,406 footprints on 28,693 transcripts. To quantify miRNA occupancy in the RISC-complex we counted the number of read that overlapped with the annotated locations (miRbase v20) on the miRNA hairpins.

Library	Raw Reads	Trimmed Reads	Genomic %	RefSeq %	RefSeq N	miRNA %	miRNA N
mRNA	236,827,218	205,594,509	45.8%	39.6%	81,389,124	15.5%	
miRNA	172,094,624	140,008,888	79.1%	74.6%		48.6%	67,980,650

HITS-CLIP – chimeric pipeline

The chimeric pipeline used cutadapt (DOI:10.14806/ej.17.1.200) to trim adapter sequence AGGGAGGACGATGCG from the 5' end and GTGTCAGTCACTTCCAGCGGTCGTATGCCGTCTTCTGCTTG from the 3' end from the mRNA reads. We allowed for up to 3 rounds of trimming. Starting with 236,827,218 raw reads as above cutadapt produced 205,052,818 trimmed reads. The trimmed reads were aligned to the RefSeq transcriptome using STAR using the following parameters, --outFilterMismatchNmax 2 and --outFilterMultimapNmax 10. We found 84,312,502 uniquely mapped reads and 1,530,815 reads that hit too many places and so were discarded. We further filtered the alignments down to 75,428,704 reads. The filtering rules were designed to select reads that were either non-chimeric well-aligned reads or chimeric reads liable to contain a miRNA fragment at the 5' end. Alignments had to be a clean end-to-end match, a match with 1 or 2 bp leading non-match, a long miRNA-like lead non-match (18bp or more), or an 8 bp trailing non-match. Alignments were considered bad if they had inserts or deletions in the alignment, or too short a match (less than 20bp). Of these reads, 3,912,266 had a potential miRNA at the 5' end. We then aligned the potential miRNA portion of the likely-chimeric reads to miRNA hairpins using bowtie. We found 728,596 (18%) reads that aligned to 290 miRNAs. We then merged the STAR and bowtie alignment information for the reads to identify mRNA/miRNA pairs.

Moreover, we evaluated the quality of the chimeric pairing by measuring the distribution of chimeric reads relative to mirRanda-predicted⁹ miRNA binding sites for a set of the most frequent miRNAs. The novel Supplementary Fig. 6 indicates that the chimeras generally agree well with miRanda predictions.

2. Labels for Fig. 1d are confusing. Does miR-142 refer to both miR-142-3p and miR-142-5p? What does miR-let-7bhg denote that is distinct from miR-let-7b? (Also, why miR-let-7 instead of just let-7?) What about miR-17hg – does that refer to the entire miRNA cluster that includes miR-17?

Response: We thank the reviewer for this comment and apologize for the unclear labeling of Fig. 1d. This was due to our HITS-CLIP pipeline which appends the 3p/5p suffix only if it needs to in order to infer the existence of a mature form that is not already included in the miRbase release we are using. We reanalyzed the data accordingly and added the suffix for all miRNAs.

3. Fig. 1F should indicate the variability in replicates (n=4 per group), and indicate whether and how read counts were normalized for quantification. Control miRNAs should be shown, and full sequencing data including sufficient information to assess data quality should be provided.

Response: To mechanistically dissect impaired Treg induction during onset of islet autoimmunity, we determined miRNA expression profiles by next generation sequencing (NGS) using CD4⁺T cells from non-diabetic children with or without islet autoimmunity. This pilot RNA-Seq experiment has been performed with pooled CD4⁺ T cell samples from four individuals with islet autoimmunity in comparison to four individuals without islet autoimmunity. To overcome this limitation validation experiments have been performed for miR142-3p with CD4⁺ T cells from children with recent onset of T1D and without T1D (Fig. 1g: here box-and-whisker plots indicating minimum to maximum values to demonstrate data distribution have been used). We apologize for not making this clearer in the first version of our manuscript. The processing, normalization and statistical analysis of the sequencing data were performed as follows: Unwanted adaptor sequences were trimmed from small RNA reads using BTrim¹⁰ and quality of sequencing was assessed for trimmed read data with a mean phred quality score of 38, referring to a base call accuracy of 99.99%. Read data was filtered of unwanted RNA fragments by mapping on rRNA, tRNA, snRNA and snoRNA sequences obtained from the Rfam database using bowtie¹¹. Remaining reads were then mapped on mature human miRNA sequences obtained from mirBase (release 20)¹² and summed up to read count lists using SAMTools. mRNA read data was processed comparably without unnecessary trimming and filtering. Raw read data was mapped on the human genome (build 37.2) using a gapped alignment for paired end data with bowtie2¹³. Finally, read count lists were created by HTSeqcount⁴⁷¹⁴. Differential expression of miRNA was evaluated using DESeq¹⁵, handling size factor correction and normalization. More information on NGS miRNA sequencing data, including a set of most abundant miRNAs relevant for T cell activation and/or Treg induction, can be found in a previous publication¹⁶.

4. The meaning “activated phenotype” should be specified for both mouse and human T cells for Fig. 1g-h. The peripheral blood usually contains very very few activated lymphocytes.

Response: We thank the reviewer for this comment and apologize for the lack of clarity. Human activated T cells, as in Fig. 1f and g, were sorted as CD4⁺CD3⁺CD45RA⁻

CD45RO⁺CD127⁺CD25^{intermediate} and murine activated T cells, as in Fig. 1h, were sorted as CD4⁺CD25⁻CD44^{high}. The corresponding gating strategies are now shown in Supplementary Fig. 1a and b.

5. Functional blockade of miR-142 activity using nanoparticle delivery of inhibitors should be validated using an activity sensor (e.g. luciferase or fluorescent protein construct with miR-142 bindings sites) rather than PCR. PCR can be affected by inhibitor binding after lysis, and does not provide information about the fraction of cells that receive effective inhibition. It may be sufficient to cite a prior publication in which this technique for delivery has been fully assessed.

Response: We thank the reviewer for this comment. The successful chitosan-coated PLGA nanoparticle-mediated miRNA uptake in CD4⁺ T cells including intracellular co-localization of the nanoparticles and the miRNA as well as effective miRNA inhibition have been shown for example in Serr et al. 2018 ¹⁶. We agree that qPCR can be affected by inhibitor binding after lysis and therefore performed additional experiments. To further strengthen the evidence for functional blockade of miR142-3p activity using nanoparticle delivery of a miR142-3p inhibitor to CD4⁺ T cells we analyzed the expression of two well-established miR142-3p targets Tgfb1 ⁵ and ATG16L1 ¹⁷. Specifically, we demonstrated functional blockade of miR142-3p activity in CD4⁺ T cells by showing increased expression of Tgfb1 and ATG16L1 in T cells stimulated in presence of a miR142-3p inhibitor compared to a control inhibitor (now presented in Supplementary Fig. 3b and c).

6. Why should miR-142 inhibitor and mimic have no effect on sorted naïve T cells from diabetic NOD mice, and in human subjects with recent onset or established T1D? Does this indicate some kind of extrinsic regulation of Treg differentiation that overrides the effect of miR-142 in diabetic individuals?

Response: We thank the reviewer for this important and constructive comment. We showed that miR142-3p inhibition *in vitro* results in increased Treg induction efficacy in NOD mice with and without islet autoimmunity, while there was only a trend towards a comparable improvement in NOD mice with established T1D. In humans miR142-3p inhibition improved Treg induction capacity in both subjects with recent onset of T1D and with long-term T1D. Here we wish to point to

the differences of established T1D between NOD mice and humans. While in humans insulin replacement therapy controls blood glucose levels relatively well, in mice the lack of insulin leads to severe dysglycemia. We suggest that this difference also affects Treg induction which is in line with the finding that Treg induction capacity without any miRNA modulation differs significantly between IAA⁺ and diabetic NOD mice, but there is only a trend towards a difference between human subjects with recent onset and long-term diabetes. Regarding the mimic we agree with the reviewer and suggest that regulation induced by the onset of diabetes could lead to multifactorial impairments in Tregs and their induction, including an increase in signaling strength of stimulation, as reported previously¹⁶. These differences in proliferative potential are in line with the impairments in Treg induction during onset of islet autoimmunity and T1D, which could override the effect of the mimic in this experimental setting. However, titration of both TCR stimulation and dose of the mimic might restore the effect.

7. The presented data are not sufficient to assess the affect of miR-142 on Treg stability. The number of CD25⁺ and FoxP3⁺ cells in restimulated induced Treg cultures could be affected by differences in the growth and survival of the mixture of Tregs and other T cells in the culture, and/or it could be affected by changes in Treg stability. To test for the latter, FoxP3 lineage markers are needed (e.g. FoxP3;Rosa26-lsl-fluorescent protein mice, which are widely available).

Response: We thank the reviewer for this important comment and the suggestion to study the aspect of Treg stability in our restimulation experiments in more detail. We addressed this point by performing additional Treg induction and restimulation experiments. These studies provided novel data regarding cell viability and proliferation. In addition we would like to highlight that we used BALB/c Foxp3GFP reporter mice for the restimulation experiments enabling the sorting of highly pure CD4⁺CD25^{hi}Foxp3^{hi} Tregs (purity > 95%) as a starting population for restimulation. As indicated in the novel Supplementary Fig. 4a and b, there were no significant differences in viability (percentage and number) or proliferation (Ki67⁺ cells) in the CD4⁺ or the CD4⁺CD25^{hi}Foxp3^{hi} Treg fraction after Treg induction in presence of the miR142-3p inhibitor or the control inhibitor. Similarly, the restimulation of the highly pure CD4⁺CD25^{hi}Foxp3^{hi} Treg fraction in presence of the miR142-3p inhibitor or the control inhibitor did not affect cell viability or proliferation (Supplementary Fig. 4c). These novel results support the notion that miR142-3p inhibition increases Treg stability *in vivo*.

8. Data availability section is incomplete. HITS-CLIP data “will be provided shortly”, and there is no mention of NGS miRNA sequencing data.

Response: We thank the reviewer for this comment and apologize for the incomplete data availability section which we completed as requested. All HITS-CLIP library sequencing data have been deposited into NCBI GEO under accession number GSE124264. More information on NGS miRNA sequencing data, including a set of most abundant miRNAs relevant for T cell activation and/or Treg induction, can be found in a previous publication ¹⁶.

9. Fig. 5a is unclear regarding chimeric reads. Does the bottom panel indicate the position of a single miR-142 chimeric read, while the middle panel indicates the read density for all chimeric reads? Also, where in this region are there predicted miR-142 binding site(s)? Are they conserved in mice? This is important since many key experiments were performed in mice. Miranda is a very permissive target prediction algorithm with many more false positive predictions than other algorithms such as Targetscan.

Response: We thank the reviewer for this comment which turned out to be extremely helpful in improving the HITS-CLIP related figure and apologize for the lack of clarity. We now provide a significantly revised version of former Fig. 5a which is now the novel Fig. 5 (for the information of this reviewer also provided below). As shown below, the figure first indicates the location of the human TET2 gene on chromosome 4. Next, we show two human TET2 transcripts (NM_017628 and NM_001127208) and below the corresponding RISC footprints (purple bars) found on these transcripts. Below we show transcript NM_001127208 zoomed in on a region of the 3'UTR between 7150 and 8050 and a more detailed view of the RISC footprints (grey bars) found in this region. The bottom panel shows the location of miR142-3p – Tet2 – chimeric reads with the colored portion indicating the miR142-3p part of the chimeras. We hope that this revised version of the HITS-CLIP related figure significantly improves the comprehensibility of our findings.

10. Further experiments are needed to validate Tet2 as a miR-142 target. Reciprocal miRNA and putative target gene expression are consistent with direct targeting, but could easily be due to indirect effects of the miRNA on other genes. This is especially true for inferences about anti-correlated miRNA and putative target gene expression patterns and related biology *in vivo* (e.g. Fig. 6). Is the 3' UTR of Tet (or the putative region bound by Ago2:miR-142 complexes) responsive to miR-142 inhibitor and/or mimics in 3' UTR reporter assays?

Response: We thank the reviewer for this helpful and constructive criticism. As requested we conducted novel *in vitro* and *in vivo* experiments to provide mechanistic evidence and validate Tet2 as a target of miR142-3p. These novel data have been included in Fig. 6e-g. Specifically, we performed luciferase reporter assays as requested. To this end, we co-transfected HEK-293 cells with a TET2 3'UTR reporter construct and a miR142-3p mimic and observed a lower Firefly/Renilla luminescence ratio compared to the transfection control. These novel data are now presented in Fig. 6e.

In addition, we used two loss-of-function models to mechanistically validate Tet2 as a direct target of miR142-3p. In line with the broad role of Tet2 in various cell types we first made use of the non-hematopoietic 3T3 fibroblast cell line, which was previously shown to lack miR142-3p expression almost completely ⁶. As a second approach we dissected the miR142-3p-Tet2 axis using T cells from miR142 knockout (miR142^{-/-}) mice ⁷. For further details please refer to the novel Supplementary Fig. 8.

Specifically, we transfected 3T3 fibroblasts with a miR142-3p mimic or a control mimic to introduce miR142-3p levels given the absence of endogenous miR142-3p expression. The introduction of miR142-3p expression in these cells following mimic application resulted in a significant decrease of Tet2 mRNA levels. These novel findings are now provided in Fig. 6f.

To further strengthen the mechanistic evidence for the direct miR142-3p-Tet2 relationship we used T cells from miR142 knockout animals. Stimulation of CD4⁺T cells from miR142 knockout mice showed significantly higher Tet2 mRNA levels when compared to T cells from miR142 competent mice, now presented in (Fig. 6g). The direct targeting of Tet2 by miR142 was also validated on the protein level. Specifically, T cells from miR142 deficient animals presented with significantly higher Tet2 protein expression following TCR stimulation when compared to T cells from miR142^{+/+} mice (Fig. 6h). These findings provide compelling evidence for direct targeting of Tet2 by miR142-3p.

11. Do the inhibitors affect miR-142 activity in T cells in the mice that are dosed systemically? Or might they be working by alternate means (in other cells to which it is easier to deliver inhibitors, or by some indirect mechanism?). This type of delivery system requires rigorous validation, as previous efforts to block miRNAs in lymphocytes in vivo have not been so successful. Much better in vivo loss of function experiments could be performed using genetic miR-142-deficiency (e.g. Sun et, JCI 2015).

Response: We thank the reviewer for this comment. We used an LNA in vivo miRNA inhibitor which has been shown to be effective in a broad range of tissues. When delivered systemically, LNA miRNA inhibitors distribute broadly into most tissues, including hematopoietic tissues such as lymph nodes, spleen and bone marrow ². We agree that besides the accumulation in the tissues, the knocking-down efficacy in the relevant tissue is highly relevant. Here, the efficient knock-down of

miR142-3p using systemic administration of a LNA miRNA inhibitor has been shown in splenocytes^{3,4}. A list of studies using *in vivo* miRNA inhibition with LNA miRNA inhibitors can be found here: <http://www.exiqon.com/ls/Documents/Scientific/mirna-inhibition-publications.pdf>.

To further strengthen the evidence for successful delivery of the inhibitor to pancreatic tissue we performed novel experiments and upon application of the miR142-3p inhibitor validated the expression of a well-established miR142-3p target *Tgfbr1*⁵ directly in pancreatic T cells. Specifically, we demonstrated delivery to pancreatic tissue and local immune cells by showing increased expression of *Tgfbr1* in pancreatic T cells when compared to NOD mice that had received a control inhibitor. These novel data are now presented in Supplementary Fig. 10a.

Reviewer #3, expert on Treg (Remarks to the Author):

In the submitted paper the authors unveil a pathway that links miR142-3p expression in T cells to a reduction in *Tet2* expression and hence Treg instability. The studies are based on type 1 diabetes in both patients and mice, and suggest that increased miR142-3p may play a pathogenic role in diabetes development. In humans, the relationship between the different observations is induced, while in mice it is formally demonstrated up to the point of T cell infiltrate (while impacts on diabetes development are not shown, possibly because such experiments may be impractical using the tools generated here). Overall, while the paper is written a bit roughly, the data are novel and potentially important.

Response: We thank the reviewer for his/her enthusiasm for our work, constructive criticism and positive comments on our manuscript highlighting novelty and potential importance. In response to these comments, which were very helpful for this revision, we conducted a series of additional *in vitro* and *in vivo* experiments to strengthen our conclusions and improve the quality of the manuscript as suggested.

Major points:

- The higher number of Tregs with the miR-142-3p inhibitor could be increased stability, as stated in the text, but altered proliferation or apoptosis of either the Treg or Tconventional fractions was not excluded in Figure 4b.

Response: We thank the reviewer for this important comment and his/her helpful suggestion. We would like to highlight that we used BALB/c Foxp3GFP reporter mice for the restimulation experiments enabling the sorting of highly pure CD4⁺CD25^{hi}Foxp3^{hi} Tregs (purity > 95%) as a starting population for restimulation. We performed novel Treg induction and restimulation experiments and analyzed cell viability and proliferation as requested. As indicated in Supplementary Fig. 4a and b, there were no significant differences in viability (percentage and number) or proliferation (Ki67⁺ cells) in the CD4⁺ or the CD4⁺CD25^{hi}Foxp3^{hi} Treg fraction after Treg induction in presence of the miR142-3p inhibitor or the control inhibitor. Similarly, the restimulation of the highly pure CD4⁺CD25^{hi}Foxp3^{hi} Treg fraction in presence of the miR142-3p inhibitor or the control inhibitor did not affect cell viability or proliferation (Supplementary Fig. 4c). These novel results support the notion that miR142-3p inhibition increases Treg stability *in vivo*.

- The studies on Foxp3 methylation patterns in Figure 4 should have been performed with and without miR-142-3p over-expression and inhibition. In the current format, it is not obvious what they contribute to the study.

Response: We thank the reviewer for this comment and apologize for the lack of clarity. To illustrate the importance of the findings presented in Fig. 4c-f we would like to highlight that the Foxp3 CNS2 is completely demethylated in Tregs but fully methylated in conventional T cells and *in vitro*-induced Tregs¹⁸. However, it was shown that Treg induction using limited TCR stimulation results in higher Treg induction efficacy and increased Treg stability compared to continuous TCR stimulation¹, suggesting that the early phase is critical for efficient Treg induction. Our FACS and DNA methylation analysis revealed that initial Treg induction from naive CD4⁺ T cells *in vitro* causes rapid CNS2 demethylation and Foxp3 expression. However, longer culture leads to methylation of the CNS, likely a result of the culture conditions. Of note, the demethylation was restricted to cells showing a Foxp3^{hi} Treg phenotype, while no changes in CNS2 methylation were observed in Foxp3⁻ cells (Supplementary Fig. 5a-d). These findings support the importance of this early phase of Treg induction and point to an involvement of the DNA demethylation machinery.

Regarding the requested studies on methylation patterns with and without miR142-3p inhibition we would like to refer to the limitations of our *in vitro* Treg induction system. Specifically, the rapid remethylation of the Foxp3 CNS2 *in vitro* interferes with the precise analysis of relatively small changes in DNA methylation. However, in addition we provide evidence by demonstrating that miRNA142-3p inhibition *in vivo* impinges on Foxp3 CNS2 methylation, as outlined now in Fig.8h.

- Are the changes to Foxp3 methylation in T1D patients compared to healthy control due to: 1) more contamination of transiently Foxp3+ T cells in the Treg isolation; 2) more Nrp1- pTreg within the Treg pool; or 3) partial methylation of tTreg?

Response: We thank the reviewer for this important question. To address this question we performed novel experiments to analyze the composition of the Treg pool in T1D patients compared to healthy controls with regard to thymus-derived and peripheral Tregs. Specifically, we measured the expression of Helios, which is a marker of thymic-derived Tregs¹⁹, in Tregs isolated from peripheral blood of human subjects without T1D and with recent onset of T1D. Helios expression did not differ between the two groups (Supplementary Fig. 9d), suggesting that the observed differences in Foxp3 CNS2 DNA methylation are not affected by the composition of the Treg pool.

- The suggested direct link between miR-142-3p, CNS3 methylation and Tet2 could be further tested in human cells by Tet2 siRNA

Response: We thank the reviewer for this helpful suggestion and performed novel experiments to further strengthen the important role of Tet2 for Treg induction also in the human system. We now performed novel Treg induction experiments using limited TCR stimulation of human naive CD4⁺ T cells in presence of a Tet2 siRNA and a control siRNA. In line with the results obtained in mice, reduced Tet2 abundance resulted in a significantly attenuated Treg induction efficacy. These novel data are now shown in Fig. 6j. Moreover and in line with these observations, we now performed additional experiments using T cells from individuals with recent onset of clinical T1D which presented with significantly reduced numbers of CD3⁺Tet2⁺T cells. These results are now presented in Fig.7d and e.

Minor points

- It is stylistic, but I would suggest that the insertion of number, standard deviation and p values into the text reduces readability, while not providing information that is not already present on the figure. In addition, the paper would be well served by implementation of the standard paragraphing format

Response: We thank the reviewer for this suggestion and agree on this point. To improve readability of the manuscript we removed number, standard deviation and p values from the text and complemented the figure legends if necessary.

- In Figures 4d and 4f, are the remethylated CNS2 Tregs at later time points still Foxp3+? The co-expression of Treg markers is stated in 4c and 4e, but a proper time course with the key markers would be more useful

Response: We thank the reviewer for this question and provide a more comprehensive time course for the key markers CD25 and Foxp3 as well as methylation of Foxp3⁺ and Foxp3⁻ cells after 12, 18 and 36 hours in Supplementary Fig 5a-d. Due to the requirement of cell lysis for methylation analysis it is not possible to analyze Treg markers of these cells at later time points. However, we would like to highlight the clear increase of Tregs over time as indicated by high levels of CD25 and Foxp3, especially after termination of TCR stimulation. As previously reported, the induced Tregs do not lose CD25 or Foxp3 expression during the Treg induction process²⁰ and we suggest that high Treg induction efficacy is related to the termination of TCR stimulation in the early phase of CNS2 demethylation.

- In Figure 6c, are Tet2⁺ T cells reduced as a proportion of CD3⁺ T cells? Or are just total T cell numbers down, which means Tet2⁺ cells will be down as a result?

Response: We thank the reviewer for this important question. To address this question we performed novel experiments and analyzed the numbers of total CD3⁺ T cells in the pancreas of IAA⁻ and IAA⁺ NOD mice, as requested. In line with previous observations, there was an increased number of pancreas infiltrating CD3⁺ T cells in IAA⁺ NOD mice, confirming a reduction of Tet2⁺ T cells as a proportion of total CD3⁺ T cells. These novel data are shown in Supplementary Fig. 9b.

1. Serr, I. *et al.* Type 1 diabetes vaccine candidates promote human Foxp3⁺Treg induction in humanized mice. *Nat. Commun.* **7**, (2016).
2. Straarup, E. M. *et al.* Short locked nucleic acid antisense oligonucleotides potently reduce apolipoprotein B mRNA and serum cholesterol in mice and non-human primates. **38**, 7100–7111 (2010).
3. Sun, Y. *et al.* Mature T cell responses are controlled by microRNA-142. *J. Clin. Invest.* **125**, 2825–2840 (2015).
4. Sun, Y. *et al.* Targeting of microRNA-142-3p in dendritic cells regulates endotoxin-induced mortality. *Blood* **117**, 6172–6183 (2011).
5. Talebi, F. *et al.* MicroRNA-142 regulates inflammation and T cell differentiation in an animal model of multiple sclerosis. 1–14 (2017). doi:10.1186/s12974-017-0832-7
6. Sun, Y. *et al.* PU.1-Dependent Transcriptional Regulation of miR-142 Contributes to Its Hematopoietic Cell – Specific Expression and Modulation of IL-6. (2013). doi:10.4049/jimmunol.1202911
7. Shrestha, A., Carraro, G., Agha, E. El, Mukhametshina, R. & Chao, C. Generation and Validation of miR-142 Knock Out Mice. 1–14 (2015). doi:10.1371/journal.pone.0136913
8. Schug, J. *et al.* Dynamic recruitment of microRNAs to their mRNA targets in the regenerating liver. *BMC Genomics* **14**, (2013).
9. Betel, D., Koppal, A., Agius, P., Sander, C. & Leslie, C. Comprehensive modeling of microRNA targets predicts functional non-conserved and non-canonical sites. (2010).
10. Kong, Y. Btrim: A fast, lightweight adapter and quality trimming program for next-generation sequencing technologies. *Genomics* **98**, 152–153 (2011).

11. Langmead, B., Trapnell, C., Pop, M. & Salzberg, S. L. Ultrafast and memory-efficient alignment of short DNA sequences to the human genome. *Genome Biol.* **10**, R25 (2009).
12. Kozomara, A. & Griffiths-Jones, S. MiRBase: Annotating high confidence microRNAs using deep sequencing data. *Nucleic Acids Res.* **42**, 68–73 (2014).
13. Langmead, B. & Salzberg, S. L. Fast gapped-read alignment with Bowtie 2. *Nat. Methods* **9**, 357–9 (2012).
14. Anders, S., Pyl, P. T. & Huber, W. HTSeq-A Python framework to work with high-throughput sequencing data. *Bioinformatics* **31**, 166–169 (2015).
15. Love, M. I., Huber, W. & Anders, S. Moderated estimation of fold change and dispersion for RNA-seq data with DESeq2. *Genome Biol.* **15**, 1–21 (2014).
16. Serr, I. *et al.* A miRNA181a/NFAT5 axis links impaired T cell tolerance induction with autoimmune type 1 diabetes. *Sci. Transl. Med.* **10**, (2018).
17. Lu, Y. *et al.* miR-142-3p regulates autophagy by targeting ATG16L1 in thymic-derived regulatory T cell (tTreg). *Cell Death Dis.* (2018). doi:10.1038/s41419-018-0298-2
18. Baron, U. *et al.* DNA demethylation in the human FOXP3 locus discriminates regulatory T cells from activated FOXP3(+) conventional T cells. *Eur. J. Immunol.* **37**, 2378–89 (2007).
19. Thornton, A. M. *et al.* Expression of Helios, an Ikaros Transcription Factor Family Member, Differentiates Thymic-Derived from Peripherally Induced Foxp3 + T Regulatory Cells. (2010). doi:10.4049/jimmunol.0904028
20. Sauer, S. *et al.* T cell receptor signaling controls Foxp3 expression via PI3K, Akt, and mTOR. *Proc. Natl. Acad. Sci. U. S. A.* **105**, 7797–7802 (2008).

Reviewers' comments:

Reviewer #1 (Remarks to the Author):

The authors have appropriately addressed this reviewer's concern and questions.

Reviewer #2 (Remarks to the Author):

The updated manuscript contains some improvements, and new data that strengthen some parts of the study. However, my previous comments about the HITS-CLIP data and the validation of miRNA antagonism in T cells (especially in tissues in vivo) have not been addressed adequately. These are important issues, and both techniques require further rigorous analysis and validation by orthogonal methods.

Evidence for miR-142 targeting of Tet2 remains unconvincing, and the HITS-CLIP data do not appear robust. After considerable effort, I was able to find the location of the 5 apparently identical miR-142 chimeric reads from the HITS-CLIP dataset. Could these 5 reads be amplified from the same cloning event? I see no mention of the use of unique molecular identifiers. In any case, there is no evidence for a miR-142 binding sequence in this region using miRANDA (at microrna.org) or Targetscan. Instead, this region contains one of several deeply conserved miR-29 binding sites. The new figure highlights just how sparse CLIP reads are in this region of the Tet2 3' UTR as compared to other parts of the mRNA, although it remains unclear whether those reads coincide with binding sites for other miRNAs that have been previously validated to target Tet2. Overall, the data provided indicates that the mRNA part of the HITS-CLIP data are not of sufficient quality to support specific miRNA:target interactions. It's not clear that any known miRNA binding sites are detected, much less how well these data agree with target predictions and prior data on miRNA/Ago binding to human transcripts. Supplementary Figure 6 shows only 6 unnamed regions with miRanda predicted binding sites.

The updated manuscript still lacks evidence for delivery of functional antagomirs in vivo and even in vitro. If this reagent is working as proposed, very efficient delivery must be achieved, since miRNA miR-142 is the most abundant miRNA in T cells. Measurement of two known targets is not sufficient, as these genes may be affected by the treatment through indirect pathways. I am aware that this method for antagomir delivery to T cells was used in a prior publication, but that reference also lacks evidence for direct effects on specific miRNA activity. An activity sensor should be used to determine what fraction of miR-142 activity in T cells is relieved by delivery of the miR-142 specific antagomir compared with controls. Genetic deletion of miR-142 can be achieved with existing mutant mice. They would provide a better means of loss of function analysis.

Reviewer #3 (Remarks to the Author):

The authors have adequately addressed all of the concerns raised in my initial review. The paper is of sufficient scope and quality to be published at Nature Communications.

Point by Point Response

Nature Communications Manuscript number NCOMMS-18-24478A

miRNA142-3p/Tet2 signaling impairs Treg induction and stability in type 1 diabetes autoimmunity

Reviewer #1 (Remarks to the Author):

The authors have appropriately addressed this reviewer's concern and questions.

Response: We thank the reviewer and are delighted that we have addressed all his/her concerns and questions satisfyingly and that the reviewer finds our newly added data critical to strengthen and clarifying our manuscript.

Reviewer #2, expert on micro-RNA and autoimmune disease (Remarks to the Author):

The updated manuscript contains some improvements, and new data that strengthen some parts of the study. However, my previous comments about the HITS-CLIP data and the validation of miRNA antagonism in T cells (especially in tissues *in vivo*) have not been addressed adequately. These are important issues, and both techniques require further rigorous analysis and validation by orthogonal methods.

Response: We thank the reviewer for the positive comments and constructive criticism. In response to the reviewer's remaining comments and questions, which were very helpful for this revision, we here provide additional information for clarification as requested. Moreover, to further strengthen the main claims of our manuscript: that Tet2 is a direct target of miR142-3p, and that inhibitor delivery *in vivo* and *in vitro* is sufficiently robust to account for the observed effects, we directly addressed these comments of the reviewer by performing a series of additional novel *in vitro* as well as *in vivo* experiments.

Evidence for miR-142 targeting of Tet2 remains unconvincing, and the HITS-CLIP data do not appear robust. After considerable effort, I was able to find the location of the 5 apparently identical

miR-142 chimeric reads from the HITS-CLIP dataset. Could these 5 reads be amplified from the same cloning event? I see no mention of the use of unique molecular identifiers. In any case, there is no evidence for a mir-142 binding sequence in this region using miRANDA (at microrna.org) or Targetscan. Instead, this region contains one of several deeply conserved mir-29 binding sites. The new figure highlights just how sparse CLIP reads are in this region of the Tet2 3' UTR as compared to other parts of the mRNA, although it remains unclear whether those reads coincide with binding sites for other miRNAs that have been previously validated to target Tet2. Overall, the data provided indicates that the mRNA part of the HITS-CLIP data are not of sufficient quality to support specific miRNA:target interactions. It's not clear that any known miRNA binding sites are detected, much less how well these data agree with target predictions and prior data on miRNA/Ago binding to human transcripts. Supplementary Figure 6 shows only 6 unnamed regions with miRanda predicted binding sites.

Response: We thank the reviewer for his comments. Our understanding is that in the view of reviewer 2 the HITS-CLIP data and the quality of the accompanied analyses need further clarification (1) and that within the HITS-CLIP data analysis the evidence for miR142-3p targeting of Tet2 is not fully convincing (2). Please find our responses to these concerns detailed below:

(1) While we understand the reviewer's concern regarding the absence of a predicted miR142-3p binding site in the Tet2 3'UTR, we would like to respectfully draw the attention of the reviewer to Supplementary Fig. 6, showing the quality of the chimeric pairing by measuring the distribution of chimeric reads relative to mirRanda-predicted miRNA binding sites for a set of the most frequent miRNAs. The Supplementary Fig. 6 indicates that the chimeras generally agree well with miRanda predictions, however, we also found a few miRNAs where the distribution of chimeric reads did not correlate with predicted binding sites. To the best of our knowledge miRNA target prediction tools first of all rely on the well characterized 6 nucleotide seed sequence at the 5' end of the miRNA (positions 2-7) ¹. The complementary pairing of miRNA and mRNA is widely used for prediction of miRNA-mRNA pairs, often in combination with evolutionary conservation ², secondary structure ³, or neighboring context information ⁴. While these rules have been valuable to identify multitudes of miRNA targets and prediction tools have been improved continuously, both false positive and false negative predictions cannot be fully excluded and some non-canonical target sites might not be identified ^{5,6}. The functional relevance of imperfectly matched miRNA seeds has been shown in various studies ⁷⁻⁹. This highlights the importance of methods like HITS-CLIP that can directly

assess functional miRNA-mRNA interactions *in vivo*, regardless and despite of *in silico* predictions. The validity and robustness of HITS-CLIP and comparable methods, including the analysis of chimeric reads, has been shown in several studies^{6,10-12}.

Therefore, while we agree that the HITS-CLIP data alone are not sufficient to define Tet2 as an undoubted direct target of miR142-3p, they are a valuable source of information for miRNA-mRNA pairs *in vivo* and provide important and sufficient experimental evidence to dissect the identified miR142-3p/Tet2 relationship and to follow up with additional molecular and cellular experimental approaches as evidenced in this manuscript.

(2) In order to provide additional convincing mechanistic evidence to validate Tet2 as a direct target of miR142-3p we performed a series of molecular and cellular *in vitro* and *in vivo* experiments.

Since some of these experiments rely on the functional delivery of the miR142-3p inhibitor we demonstrated the high efficacy and specificity of the miRNA inhibitor in several independent experimental settings: First, the chitosan-coated PLGA nanoparticle-mediated miRNA uptake in CD4⁺ T cells including intracellular co-localization of the nanoparticles and the miRNA was shown using fluorescently-labeled nanoparticles and miRNAs (Figure 1 for the reviewers). Second, to demonstrate the efficacy of the miR142-3p inhibitor in T cells *in vitro*, the stimulation of CD4⁺ T cells in presence of the miR142-3p inhibitor reduced miR142-3p abundance by more than 99% after three hours of incubation (Supplementary Figure 3a). Third, the co-transfection of Jurkat T cells with a luciferase reporter plasmid containing the miR142-3p target sequences in the 3'-UTR and the miR142-3p inhibitor significantly de-repressed luciferase expression (Supplementary Figure 3b). Fourth, the functional inhibition of miR142-3p activity in CD4⁺ T cells was demonstrated by showing increased expression of Tgfbr1 and ATG16L1, two established target genes of miR142-3p, in T cells stimulated in presence of a miR142-3p inhibitor compared to a control inhibitor (Supplementary Fig. 3c and d).

The delivery of functional miR142-3p inhibitors *in vivo* was demonstrated using an LNA *in vivo* miRNA inhibitor which has been shown to accumulate in a broad range of tissues and to efficiently inhibit its targets in several independent studies^{13,14}. In order to provide experimental evidence for the delivery of functional miR142-3p inhibitors specifically to CD4⁺ T cells *in vivo* we used a fluorescently-labeled miR142-3p inhibitor. Importantly, the successful delivery of the inhibitor to

CD4⁺ T cells in relevant draining lymph nodes including liver-draining lymph nodes, mesenteric lymph nodes as well as pancreatic lymph nodes and directly in pancreas-residing CD4⁺ T cells was shown after 4 hours (Supplementary Figure 10a) and 24 hours (Supplementary Figure 10b). Previous miRNA inhibition experiments *in vivo* were assessed with a miRNA inhibitor application for 14 days every other day. Therefore, the successful delivery of the inhibitor to CD4⁺ T cells in lymph nodes was likewise analyzed at the end of the 14-days application period (Supplementary Figure 10c and d). In addition, the efficacy of the successfully delivered inhibitors was demonstrated by showing increased expression of *Tgfb1*, a well-established target of miR142-3p, directly in pancreatic T cells (Supplementary Figure 10e and f).

In order to provide further evidence for the miR142-3p targeting of Tet2, we performed Treg induction experiments using naive CD4⁺ T cells from BALB/c mice in the presence of the miR142-3p inhibitor or control inhibitor and analyzed Tet2 mRNA and protein expression. Inhibition of miR142-3p resulted in significantly higher Tet2 mRNA levels after 6 hours of TCR stimulation (Figure 6b), and increased Tet2 protein abundance after 12 hours of TCR stimulation (Figure 6d). Importantly, there was no effect of miR142-3p inhibition on Tet1 or Tet3 expression (Supplementary Figure 7b).

To support the notion that miR142-3p directly targets the Tet2 mRNA, we co-transfected HEK-293 cells with a TET2 3'UTR reporter construct and a miR142-3p mimic and observed a significant reduction in luciferase activity compared to the transfection control (Supplementary Figure 3b).

Furthermore, to support the direct miR142-3p – Tet2 target relationship, we employed two loss-of-function models, 3T3 fibroblasts and miR142 knockout (miR142^{-/-}) mice. As virtually all non-hematopoietic cells the 3T3 fibroblast cell line lacks miR142-3p expression almost completely (Supplementary Figure 8a), providing a suitable experimental system to study the effect of miR142-3p in absence of endogenous miR142-3p expression. Here, the introduction of miR142-3p in these cells via transfection with a miR142-3p mimic resulted in significantly decreased Tet2 mRNA levels (Figure 6f).

Next, we used T cells from miR142 deficient animals to validate the direct targeting of Tet2 by miR142-3p directly in a mouse model. The stimulation of CD4⁺ T cells from miR142^{-/-} mice resulted

in significantly increased Tet2 mRNA levels when compared to T cells from miR142^{+/+} mice (Figure 6g). In line with the increased expression of Tet2 mRNA, Tet2 protein levels were likewise elevated in T cells from miR142 deficient mice following TCR stimulation when compared to T cells from mice expressing miR142 (Figure 6h).

As described above, stimulation of BALB/c T cells in presence of a miR142-3p inhibitor resulted in increased expression of Tet2 (Figure 6b) when compared to a control inhibitor. To confirm that these changes are due to a direct effect of miR142-3p inhibition and that Tet2 is a direct target of miR142-3p, we performed corresponding experiments with T cells from miR142^{-/-} mice which indicated the absence of any difference between T cells stimulated in presence of a miR142-3p inhibitor and a control inhibitor (Supplementary Figure 8c).

In line with a direct targeting of Tet2 by miR142-3p its inhibition *in vivo* resulted in significantly elevated Tet2 levels in pancreatic T cells of IAA⁺ NOD mice (Figure 8f, g). In addition, this effect was accompanied by an elevated Tet2 expression (Supplementary Figure 11d) in peripheral T cells of miR142-3p inhibitor treated IAA⁺ NOD mice.

Finally, the application of a miR142-3p inhibitor to humanized NSG mice showed a significant increase in Tet2 expression in pancreatic T cells (Supplementary Figure 13b, c).

Taken together the above-mentioned *in vitro* and *in vivo* experiments provide compelling evidence for direct targeting of Tet2 by miR142-3p.

The updated manuscript still lacks evidence for delivery of functional antagomirs *in vivo* and even *in vitro*. If this reagent is working as proposed, very efficient delivery must be achieved, since miRNA miR-142 is the most abundant miRNA in T cells. Measurement of two known targets is not sufficient, as these genes may be affected by the treatment through indirect pathways. I am aware that this method for antagomir delivery to T cells was used in a prior publication, but that reference also lacks evidence for direct effects on specific miRNA activity. An activity sensor should be used to determine what fraction of miR-142 activity in T cells is relieved by delivery of the miR-142 specific antagomir compared with controls. Genetic deletion of miR-142 can be achieved with existing mutant mice. They would provide a better means of loss of function analysis.

Response: We thank the reviewer for this important and constructive comment and performed additional novel experiments to further validate the delivery of functional inhibitors *in vitro* and *in vivo* as requested.

miR142-3p inhibitor delivery *in vitro*

The successful chitosan-coated PLGA nanoparticle-mediated miRNA uptake in CD4⁺ T cells including intracellular co-localization of the nanoparticles and the fluorescently-labeled miRNA has been shown in Serr et al. 2018¹⁵. Naive CD4⁺ T cells were stimulated with anti-CD3 and anti-CD28 antibodies for 18 hours in the presence of FA-labeled nanoparticles and/or a fluorescently-labeled miRNA. As shown in a previous study (Serr et al. 2018¹⁵), FA-labeled nanoparticles and the miRNA were efficiently taken up by the cells (Fig.S5 A and B) and confocal microscopy images show the intracellular co-localization of nanoparticles and miRNA (Fig.S5 C).”

[REDACTED]

In addition, to demonstrate the efficacy of the miR142-3p inhibitor in T cells *in vitro*, we now performed additional experiments with a miR142-3p activity sensor as requested. The miR142-3p activity sensor plasmid was constructed by inserting a double-stranded oligonucleotide containing the miR142-3p target sequences (miR142-3p reverse complement, with a central bulged mismatch) into the 3'-UTR of a dual luciferase reporter plasmid. The co-transfection of Jurkat T cells with the miR142-3p sensor plasmid and the miR142-3p inhibitor significantly de-repressed luciferase expression (Figure 2 for the reviewers/ Supplementary Figure 3b in the revised manuscript).

Figure 2 for the reviewers. miR142-3p inhibitor efficiently blocks its target miRNA. Normalized luciferase activity of Jurkat T cells co-transfected with a miR142-3p inhibitor and a miR142-3p activity sensor plasmid containing the miR142-3p target sequences in the 3'-UTR. $n = 4$.

Taken together, we show the successful miRNA uptake in CD4⁺ T cells including intracellular co-localization of the nanoparticles and the fluorescently-labeled miRNA (Figure 1 for the reviewers). Moreover, we now demonstrate reduced miR142-3p activity in T cells by delivery of the miR142-3p inhibitor compared with a control inhibitor using experiments with a miR142-3p activity sensor as suggested (Figure 2 for the reviewers / Supplementary Figure 3b in the revised manuscript).

miR142-3p inhibitor delivery *in vivo*

For the inhibition of miR142-3p *in vivo* we used an LNA *in vivo* miRNA inhibitor which has been shown to accumulate in a broad range of tissues and to efficiently inhibit its targets in several

independent studies ^{13,14}. Specifically, important studies using LNA miRNA inhibitors to successfully inhibit miRNAs *in vivo*, among others in T cells, are listed in Table 1 and an even more extensive list of studies can be found here: <http://www.exiqon.com/ls/Documents/Scientific/mirna-inhibition-publications.pdf>.

Table 1. *In vivo* miRNA inhibition using LNA miRNA inhibitors. List of selected publication.

microRNA	Tissue	Organism	Process/Disease	Reference
miR-221	Various	Mouse (plus toxicology studies in non-human primates)	Multiple Myeloma	Gallo Cantafio et al. , Mol. Ther. Nucl. Acids, 2016
miR-802	Liver	Mouse	Diabetes	Kornfeld et al. , Nature 2013
miR-33	Liver	Mouse	Cholesterol regulation	Najafi-Shoushtari et al. , Science 2010
miR-142-3p	T-cells	Mouse	Graft-versus-host disease	Sun et al. , JCI 2015
miR-192	Kidney	Mouse	Diabetic Nephropathy	Putta et al. , J Am Soc Nephrol 2012
miR-134	Brain	Mouse	Epilepsy	Jimenez-Mateos et al. , 2012, Nature Medicine
miR-212	Brain	Rat	Addiction	Hollander et al. , Nature 2010
miR-21	Lung	Mouse	Fibrosis	Liu et al. , J Exp Med 2010
miR-34a	Heart	Mouse	Cardiac decline/ myocardial infarction	Boon et al. , Nature 2013
miR-21-5p	Heart	Mouse	Obesity	Seeger et al. , Obesity 2014
miR-199a, miR-1908	Melanoma	Mouse	Melanoma metastasis	Pencheva et al. , Cell 2012
miR-712	Aorta endothelium	Mouse	Atherosclerosis	Son et al. , Nature Commun 2013
miR-33	Eye	Mouse	Age related macular degeneration	Sene et al. , Cell Metabolism 2013
miR-21-5p	Bone marrow – hemapoietic stem cells	Mouse	Myelodysplastic	Bhagat et al. , Blood 2013

When delivered systemically, a series of studies have documented that LNA miRNA inhibitors distribute broadly into most tissues, including hematopoietic tissues such as lymph nodes, spleen and bone marrow^{13,14}. The results of two studies show the accumulation of miRNA inhibitors in various tissues (Figure 2, Straarup *et al.* 2010 14; Figure 4 Cantafio *et al.* 2016 13).

[REDACTED]

[REDACTED]

We now performed additional novel experiments using a fluorescently-labeled miR142-3p inhibitor in order to provide further experimental evidence for the delivery of functional miR142-3p inhibitors specifically to CD4⁺ T cells *in vivo*. Importantly, we confirmed the successful delivery of the inhibitor to CD4⁺ T cells in relevant draining lymph nodes including liver-draining lymph nodes, mesenteric lymph nodes as well as pancreatic lymph nodes and directly in pancreas-residing CD4⁺ T cells as assessed after 4 hours (Figure 5a for the reviewers / Supplementary Figure 10a in the revised manuscript) and 24 hours (Figure 5b for the reviewers / Supplementary Figure 10b in the revised manuscript).

Figure 5 for the reviewers. miR142-3p inhibitor accumulates in CD4⁺ T cells *in vivo*. (a) Representative histograms showing the accumulation of the FAM-labelled miR142-3p inhibitor in CD4⁺ T cells isolated from relevant lymph nodes and the pancreas 4 hours after inhibitor application. (b) Representative histograms showing the accumulation of the FAM-labelled miR142-3p inhibitor

in CD4⁺ T cells isolated from relevant lymph nodes and the pancreas 24 hours after inhibitor application. **(c)** Representative histograms showing the accumulation of the FAM-labelled miR142-3p inhibitor in CD4⁺ T cells isolated from lymph nodes after 14 days of inhibitor application every other day. **(d)** miR142-3p inhibitor abundance (median fluorescence intensity) in CD4⁺ T cells isolated from lymph nodes after 14 days of inhibitor application every other day. *n* = 4.

In addition, we likewise demonstrated the successful delivery of the inhibitor to CD4⁺ T cells in lymph nodes at the end of the 14-days application period (Figure 5c and for the reviewers / Supplementary Figure 10c and d in the revised manuscript). Moreover, we validated the efficiency of the delivered miR142-3p inhibitor in pancreatic tissue and local immune cells by showing increased expression of *Tgfr1*, a well-established target of miR142-3p, in pancreatic T cells (Supplementary Figure 10e and f in the revised manuscript).

Direct effect of the miR142-3p inhibitor *in vitro* and *in vivo*

We now also performed additional *in vivo* experiments using miR142 deficient animals to further validate the specific effect of the miR142-3p inhibitor *in vitro* and *in vivo*. In a previous set of experiments outlined in the manuscript we have demonstrated increased expression of *Tet2* (Figure 6 in the revised manuscript) and two well-established miR142-3p targets, *Tgfr1* and *ATG16L1* (Supplementary Figure 3 in the revised manuscript), in BALB/c T cells stimulated in presence of a miR142-3p inhibitor compared to a control inhibitor. Now, we performed corresponding experiments with T cells from miR142^{-/-} mice which indicated the absence of any difference between T cells stimulated in presence of a miR142-3p inhibitor and a control inhibitor (Figure 6a-c for the reviewers / Supplementary Figure 8c-e in the revised manuscript). The findings in the miR142 knockout animals again underline the direct effect of the miR142-3p inhibitor and provide compelling evidence that *Tet2* is a direct target of miR142-3p.

Figure 6 for the reviewers. miR142-3p inhibitor has no effect in miR142^{-/-} CD4⁺ T cells *in vitro*. (a) Tet2 mRNA abundance after 6 hours of TCR stimulation of CD4⁺ T cells isolated from lymph nodes of miR142^{-/-} mice, in presence of a miR142-3p inhibitor and a control inhibitor respectively. *n* = 4. (b) Tgfbr1 mRNA abundance after 6 hours of TCR stimulation of CD4⁺ T cells isolated from lymph nodes of miR142^{-/-} mice, in presence of a miR142-3p inhibitor and a control inhibitor respectively. *n* = 4. (c) ATG16L1 mRNA abundance after 6 hours of TCR stimulation of CD4⁺ T cells isolated from lymph nodes of miR142^{-/-} mice, in presence of a miR142-3p inhibitor and a control inhibitor respectively. *n* = 4.

In addition and to confirm that the observed effects of miR142-3p inhibitor application on Treg induction, stability and Tet2 in NOD mice *in vivo* were directly mediated by reduced miR142-3p activity, we now performed novel *in vivo* experiments and applied the miR142-3p inhibitor to miR142^{-/-} mice. As expected, the inhibitor had no effect on Treg frequency (Figure 7a for the reviewers / Supplementary Figure 12a in the revised manuscript), Tet2 protein abundance (Figure 7b for the reviewers / Supplementary Figure 12b in the revised manuscript) or Foxp3 CNS2 methylation (Figure 7c for the reviewers / Supplementary Figure 12c in the revised manuscript) in miR142 deficient mice, confirming the delivery of functional miR142-3p inhibitors *in vivo* and Tet2 as a direct target of miR142-3p.

Figure 7 for the reviewers. miR142-3p inhibitor has no effect in *miR142*^{-/-} mice *in vivo*. (a) *Ex vivo* CD25^{hi}Foxp3⁺ Tregs isolated from lymph nodes of *miR142*^{-/-} mice treated with a miR142-3p inhibitor or control inhibitor for 14 days with 10 mg/kg ip every other day. *n* = 4. (b) *Ex vivo* Tet2 protein abundance (median fluorescence intensity) in CD4⁺ T cells isolated from lymph nodes of *miR142*^{-/-} mice treated with a miR142-3p inhibitor or control inhibitor as described in (a). *n* = 4. (c) Methylation of the Fxp3 CNS2 (mean of all sites) in Tregs isolated from lymph nodes of *miR142*^{-/-} mice treated with a miR142-3p inhibitor or control inhibitor as described in (a). *n* = 4.

These results clearly indicate that the effect observed *in vitro* and *in vivo* are due to the specific inhibition of miR142-3p by means of the miR142-3p inhibitor. Altogether we provide compelling evidence for a miR142-3p/Tet2/Foxp3 axis in murine and human CD4⁺ T cells that during islet autoimmunity interferes with the efficient induction of Tregs and leads to impairments in Treg stability. Importantly, we strengthen the claims that Tet2 is a direct target of miR142-3p, and that inhibitor delivery *in vivo* and *in vitro* is sufficiently robust to account for the observed effects.

Reviewer #3 (Remarks to the Author):

The authors have adequately addressed all of the concerns raised in my initial review. The paper is of sufficient scope and quality to be published at Nature Communications.

Response: We thank the reviewer and are delighted that we have addressed his/her concerns satisfyingly and that the reviewer finds our revised manuscript of sufficient scope and quality to be published at Nature Communications.

1. Bartel, D. P. MicroRNAs: Target Recognition and Regulatory Functions. *Cell* **136**, 215–233 (2009).
2. Lewis, B. P., Burge, C. B. & Bartel, D. P. Conserved seed pairing, often flanked by adenosines, indicates that thousands of human genes are microRNA targets. *Cell* **120**, 15–20 (2005).
3. Long, D. *et al.* Potent effect of target structure on microRNA function. *Nat. Struct. Mol. Biol.* **14**, 287–294 (2007).
4. Grimson, A. *et al.* MicroRNA Targeting Specificity in Mammals: Determinants beyond Seed Pairing. *Mol. Cell* **27**, 91–105 (2007).
5. Chi, S. W., Hannon, G. J. & Darnell, R. B. An alternative mode of microRNA target recognition. *Nat. Struct. Mol. Biol.* **19**, 321–327 (2012).
6. Mittal, N. & Zavolan, M. Seq and CLIP through the miRNA world. *Genome Biology* **15**, (2014).
7. Vella, M. C., Choi, E. Y., Lin, S. Y., Reinert, K. & Slack, F. J. The *C. elegans* microRNA let-7 binds to imperfect let-7 complementary sites from the lin-41 3'UTR. *Genes Dev.* **18**, 132–137 (2004).
8. Tay, Y., Zhang, J., Thomson, A. M., Lim, B. & Rigoutsos, I. MicroRNAs to Nanog, Oct4 and Sox2 coding regions modulate embryonic stem cell differentiation. *Nature* **455**, 1124–1128 (2008).
9. Didiano, D. & Hobert, O. Perfect seed pairing is not a generally reliable predictor for miRNA-target interactions. *Nat. Struct. Mol. Biol.* **13**, 849–851 (2006).
10. Chi, S. W., Zang, J. B., Mele, A. & Darnell, R. B. Argonaute HITS-CLIP decodes microRNA–mRNA interaction maps. *Nature* **460**, 479–486 (2009).
11. Kameswaran, V. *et al.* Epigenetic regulation of the DLK1-MEG3 MicroRNA cluster in human type 2 diabetic islets. *Cell Metab.* **19**, 135–145 (2014).
12. Schug, J. *et al.* Dynamic recruitment of microRNAs to their mRNA targets in the regenerating liver. *BMC Genomics* **14**, (2013).
13. Cantafio, M. E. G. *et al.* Pharmacokinetics and pharmacodynamics of a 13-mer LNA-inhibitor-miR-221 in mice and non-human primates. *Mol. Ther. - Nucleic Acids* **5**, (2016).

14. Straarup, E. M. *et al.* Short locked nucleic acid antisense oligonucleotides potently reduce apolipoprotein B mRNA and serum cholesterol in mice and non-human primates. **38**, 7100–7111 (2010).
15. Serr, I. *et al.* A miRNA181a/NFAT5 axis links impaired T cell tolerance induction with autoimmune type 1 diabetes. *Sci. Transl. Med.* **10**, (2018).

Reviewers' comments:

Reviewer #2 (Remarks to the Author):

The authors decline to provide further analysis of the HITS-CLIP data or to discuss specific criticisms of the sparse and possibly jackpotted data at the Tet2 3' UTR site featured in the paper. In the current form, the manuscript does not provide sufficient information to support the inclusion of the HITS-CLIP data.

The authors provide some additional experiments that are consistent with their assumption that in vivo delivered antagonists act in T cells to functionally inhibit miR-142 in those cells with downstream biological consequences. However, there plausible alternative explanations for these observations. The authors decline to directly test the activity of miR-142 in T cells in vivo (or even in vitro). Cited prior publications that used this technique and suggested direct effects in T cells also failed to provide this absolutely critical validatory data.

More detailed responses to the authors's rebuttal is provided in the attached file (all new material is in black, bold print).

Reviewer #3 (Remarks to the Author):

Excellent job, all remaining issues from reviewer 2 dealt with. Adrian Liston.

Point by Point Response

Nature Communications Manuscript number NCOMMS-18-24478B

miRNA142-3p/Tet2 signaling impairs Treg induction and stability in type 1 diabetes autoimmunity

Reviewers' comments:

Reviewer #2 (Remarks to the Author):

The authors decline to provide further analysis of the HITS-CLIP data or to discuss specific criticisms of the sparse and possibly jackpotted data at the Tet2 3' UTR site featured in the paper. In the current form, the manuscript does not provide sufficient information to support the inclusion of the HITS-CLIP data.

The authors provide some additional experiments that are consistent with their assumption that *in vivo* delivered antagonists act in T cells to functionally inhibit miR-142 in those cells with downstream biological consequences. However, there plausible alternative explanations for these observations. The authors decline to directly test the activity of miR-142 in T cells *in vivo* (or even *in vitro*). Cited prior publications that used this technique and suggested direct effects in T cells also failed to provide this absolutely critical validatory data.

More detailed responses to the authors's rebuttal is provided in the attached file (all new material is in black, bold print).

Response for revision 3: We thank the reviewer for this comment and the detailed assessment of our HITS-CLIP data in this and the previous revisions. Because of the remaining concerns of the reviewer regarding the data quality of the chimeric reads, we follow the reviewer's and the editorial board's suggestion and removed the chimeric reads part of the HITS-CLIP analysis from the manuscript and focus more on our molecular and cellular approaches to confirm Tet2 as a direct target of miR142-3p. Specifically, we now removed the information regarding the chimeric reads from Figure 5 (lower panel) and we relocated the remaining part of this figure to the supplementary figures (Supplementary Figure 7). In line with these novel

modifications we have updated the manuscript text accordingly and removed the chimeric reads paragraph from the results (pages 10 and 11 of the manuscript) and the methods section (pages 27 and 28 of the manuscript).

Altogether, we provide compelling evidence for the successful delivery of the miR142-3p inhibitor to T cells *in vitro* and *in vivo* using labelled inhibitors (Supplementary Figure 12a-d, added in revision 2) and/or nanoparticles (Figure 1 for the reviewers, added in revision 2). In addition we show an increased abundance of two established targets of miR142-3p in presence of the inhibitor *in vitro* and *in vivo* and the absence of this effect of miR142^{-/-} T cells *in vitro* and *in vivo* (Supplementary Figure 4c, d, added in revision 1; Supplementary Figure 10 e, f added in Revision 2; Supplementary Figure 12e, f, added in revision 1).

Regarding the miR142-3p activity sensor we would like to respectfully disagree. As requested by the reviewer we directly tested the activity of miR142-3p in T cells *in vitro* in revision 2 using a luciferase reporter with an artificial 3'UTR containing the complementary sequence to miR142-3p. The experiment is described in the results section on page 7 and in the methods section on page 31 of the current manuscript and the corresponding results are shown in Supplementary Figure 4b.

Results: “*Second, application of the miRNA inhibitor significantly de-repressed luciferase activity of a miR142-3p sensor plasmid in Jurkat T cells (Supplementary Fig. 4b).*” Page 7 of the manuscript.

Methods: “*A miR142-3p activity sensor plasmid was constructed by inserting a double-stranded oligonucleotide containing the miR142-3p target sequences (miR142-3p reverse complement, with a central bulged mismatch) into the 3'-UTR of a dual luciferase reporter plasmid.*” Page 31 of the manuscript.

In addition we now provide the critical controls with mutated binding sites for the Tet2 3'UTR luciferase experiments, as requested by the reviewer. We mutated both, the potential binding site identified by the chimeric reads of HITS-CLIP (mut 1) and two predicted binding sites (TargetsScan, Agarwal 2015) (mut 2). These results are shown in Figure 1 for the editors/reviewers (page 3 of this response letter) / Supplementary Figure 10a and described on pages 11 and 12 of the manuscript.

Results: *“Moreover, a miR142-3p mimic induced a significant reduction in luciferase activity in HEK-293 cells transfected with a wildtype TET2 3’UTR reporter construct while there was no effect of the mimic in cells transfected with a reporter construct containing the TET2 3’UTR with mutated miR142-3p binding sites (Fig. 5e, Supplementary Fig. 10a).”* Pages 11 and 12 of the manuscript.

In addition, a more detailed response to the additional comments of reviewer #2, including the references to the respective figures and sections of the updated manuscript, is provided below.

Reviewer #3 (Remarks to the Author):

Excellent job, all remaining issues from reviewer 2 dealt with. Adrian Liston.

Response for revision 3: We thank the reviewer for his comment and are delighted that, in his opinion, we have addressed all concerns satisfyingly and that the reviewer finds our revised manuscript of sufficient scope and quality to be published at Nature Communications.

Additional comments of Reviewer #2

Comments of the reviewer in revision 2: black

Responses of the authors in revision 2: blue

New comments of the reviewer in revision 3: black, bold type

New responses of the authors in revision 3: red

Reviewer #2, expert on micro-RNA and autoimmune disease (Remarks to the Author):

The updated manuscript contains some improvements, and new data that strengthen some parts of the study. However, my previous comments about the HITS-CLIP data and the validation of miRNA antagonism in T cells (especially in tissues *in vivo*) have not been addressed adequately. These are important issues, and both techniques require further rigorous analysis and validation by orthogonal methods.

Response of authors provided in Revision 2: We thank the reviewer for the positive comments and constructive criticism. In response to the reviewer's remaining comments and questions, which were very helpful for this revision, we here provide additional information for clarification as requested. Moreover, to further strengthen the main claims of our manuscript: that Tet2 is a direct target of miR142-3p, and that inhibitor delivery *in vivo* and *in vitro* is sufficiently robust to account for the observed effects, we directly addressed these comments of the reviewer by performing a series of additional novel *in vitro* as well as *in vivo* experiments.

Evidence for miR-142 targeting of Tet2 remains unconvincing, and the HITS-CLIP data do not appear robust. After considerable effort, I was able to find the location of the 5 apparently identical miR-142 chimeric reads from the HITS-CLIP dataset. Could these 5 reads be amplified from the same cloning event? I see no mention of the use of unique molecular identifiers. In any case, there is no evidence for a miR-142 binding sequence in this region using miRANDA (at microrna.org) or Targetscan. Instead, this region contains one of several deeply conserved mir-29 binding sites. The new figure highlights just how sparse CLIP reads are in this region

of the Tet2 3' UTR as compared to other parts of the mRNA, although it remains unclear whether those reads coincide with binding sites for other miRNAs that have been previously validated to target Tet2. Overall, the data provided indicates that the mRNA part of the HITS-CLIP data are not of sufficient quality to support specific miRNA:target interactions. It's not clear that any known miRNA binding sites are detected, much less how well these data agree with target predictions and prior data on miRNA/Ago binding to human transcripts. Supplementary Figure 6 shows only 6 unnamed regions with miRanda predicted binding sites.

Response of authors for revision 2: We thank the reviewer for his comments. Our understanding is that in the view of reviewer 2 the HITS-CLIP data and the quality of the accompanied analyses need further clarification (1) and that within the HITS-CLIP data analysis the evidence for miR142-3p targeting of Tet2 is not fully convincing (2). Please find our responses to these concerns detailed below:

(1) While we understand the reviewer's concern regarding the absence of a predicted miR142-3p binding site in the Tet2 3'UTR, we would like to respectfully draw the attention of the reviewer to Supplementary Fig. 6, showing the quality of the chimeric pairing by measuring the distribution of chimeric reads relative to miRanda-predicted miRNA binding sites for a set of the most frequent miRNAs. The Supplementary Fig. 6 indicates that the chimeras generally agree well with miRanda predictions, however, we also found a few miRNAs where the distribution of chimeric reads did not correlate with predicted binding sites. To the best of our knowledge miRNA target prediction tools first of all rely on the well characterized 6 nucleotide seed sequence at the 5' end of the miRNA (positions 2-7) ¹. The complementary pairing of miRNA and mRNA is widely used for prediction of miRNA-mRNA pairs, often in combination with evolutionary conservation ², secondary structure ³, or neighboring context information ⁴. While these rules have been valuable to identify multitudes of miRNA targets and prediction tools have been improved continuously, both false positive and false negative predictions cannot be fully excluded and some non-canonical target sites might not be identified ^{5,6}. The functional relevance of imperfectly matched miRNA seeds has been shown in various studies ⁷⁻⁹. This highlights the importance of methods like HITS-CLIP that can directly assess

functional miRNA-mRNA interactions *in vivo*, regardless and despite of *in silico* predictions. The validity and robustness of HITS-CLIP and comparable methods, including the analysis of chimeric reads, has been shown in several studies^{6,10–12}.

Therefore, while we agree that the HITS-CLIP data alone are not sufficient to define Tet2 as an undoubted direct target of miR142-3p, they are a valuable source of information for miRNA-mRNA pairs *in vivo* and provide important and sufficient experimental evidence to dissect the identified miR142-3p/Tet2 relationship and to follow up with additional molecular and cellular experimental approaches as evidenced in this manuscript.

Response of reviewer 2 to revision 2: This response offers nothing new to validate the quality of the HITS-CLIP data or its analysis in this manuscript. Supplementary Figure 6 shows six unnamed loci as validation for a transcriptome-wide dataset. Many of these loci show a very high background in addition to what appear to be specific chimeric reads near an identified predicted binding site. This selective analysis does not provide sufficient information for readers (or this reviewer) to assess how to interpret the presence of just 5 chimeric reads in Tet2, particularly when they do not correspond with a predicted site, and particularly when they are all identical reads (unlike those shown in Supplementary Figure 6). There is no doubt that HITS-CLIP is a valuable tool for identifying bona fide miRNA:mRNA target interactions, but only when sufficient data quality can be obtained and demonstrated to readers.

Response of authors for revision 3: We thank the reviewer for the very detailed assessment of our HITS-CLIP data in this and the previous revision. On grounds of the remaining concerns of the reviewer regarding the data quality of the chimeric reads, we follow the reviewer's and the editorial board's suggestion and removed the chimeric reads part of the HITS-CLIP analysis from the manuscript and focus more on our molecular and cellular approaches to confirm Tet2 as a direct target of miR142-3p. Specifically, we now removed the information regarding the chimeric reads from Figure 5 (lower panel) and we relocated the remaining part of this figure

to the supplementary figures (Supplementary Figure 7). In line with these novel modifications we have updated the manuscript text accordingly and removed the chimeric reads paragraph from the results (pages 10 and 11 of the manuscript) and the methods section (page 27 and 28 of the manuscript).

Response of authors for revision 2: (2) In order to provide additional convincing mechanistic evidence to validate Tet2 as a direct target of miR142-3p we performed a series of molecular and cellular *in vitro* and *in vivo* experiments.

Since some of these experiments rely on the functional delivery of the miR142-3p inhibitor we demonstrated the high efficacy and specificity of the miRNA inhibitor in several independent experimental settings: First, the chitosan-coated PLGA nanoparticle-mediated miRNA uptake in CD4⁺ T cells including intracellular co-localization of the nanoparticles and the miRNA was shown using fluorescently-labeled nanoparticles and miRNAs (Figure 1 for the reviewers). Second, to demonstrate the efficacy of the miR142-3p inhibitor in T cells *in vitro*, the stimulation of CD4⁺ T cells in presence of the miR142-3p inhibitor reduced miR142-3p abundance by more than 99% after three hours of incubation (Supplementary Figure 3a). Third, the co-transfection of Jurkat T cells with a luciferase reporter plasmid containing the miR142-3p target sequences in the 3'-UTR and the miR142-3p inhibitor significantly de-repressed luciferase expression (Supplementary Figure 3b). Fourth, the functional inhibition of miR142-3p activity in CD4⁺ T cells was demonstrated by showing increased expression of Tgfb1 and ATG16L1, two established target genes of miR142-3p, in T cells stimulated in presence of a miR142-3p inhibitor compared to a control inhibitor (Supplementary Fig. 3c and d).

Response of Reviewer 2 to Revision 2: The Jurkat experiment is new in this revision. Using a Tet3 3'UTR reporter makes for circular reasoning here. It also reduces the dynamic range of the possible effect. The provided data show ~20-25% increase in the sensor activity. Much better sensors for the activity of miRNAs (including miR-142) have been published many times in the past. The typical approach is to deliver by plasmid transfection or retrovirus transduction a gene encoding luciferase, GFP, or some other reporter protein with an artificial 3'UTR containing multiple perfect complementary sequence

matches to the full length miRNA. A sensor like this should be used in the primary cells to validate functional blocking of the miRNA in these cells directly. Looking directly at 2 known targets in primary cells is/was not adequate, as these effects could be indirect. However, RNA-seq and full analysis of all miR-142 targets and predicted targets would be another useful approach.

Response of authors for Revision 3: We thank the reviewer for this comment and would also like to use this comment to respectfully disagree. As described above and in the reviewer's comments in revision 2 the reviewer had suggested using an activity sensor to test the activity of miR142-3p in T cells. According to his suggestions we have performed exactly these experiments in Jurkat T cells in revision 2. This specific experiment is described in the results section on page 7 and in the methods section on page 32 of the current manuscript and the corresponding results are shown in Supplementary Figure 4b. The respective sections of the manuscript are provided below:

Results: *“Second, application of the miRNA inhibitor significantly de-repressed luciferase activity of a miR142-3p sensor plasmid in Jurkat T cells (Supplementary Fig. 4b).”* Page 7 of the manuscript.

Methods: *“A miR142-3p activity sensor plasmid was constructed by inserting a double-stranded oligonucleotide containing the miR142-3p target sequences (miR142-3p reverse complement, with a central bulged mismatch) into the 3'-UTR of a dual luciferase reporter plasmid.”* Page 32 of the manuscript.

We would like to emphasize that the luciferase reporter plasmid used in this activity sensor experiment (Supplementary Figure 4b) does not contain the TET2 3'UTR. We delivered, by plasmid transfection, a luciferase reporter gene with an artificial 3'UTR containing the perfect complementary sequence to miR142-3p, as suggested by the reviewer in this revision and in revision 2. We used this sensor plasmid in Jurkat T cells to validate the functional blocking of miR142-3p by the inhibitor directly in T cells.

Response of authors for revision 2: The delivery of functional miR142-3p inhibitors *in vivo* was demonstrated using an LNA *in vivo* miRNA inhibitor which has been shown to accumulate in a broad range of tissues and to efficiently inhibit its targets in several independent studies^{13,14}. In order to provide experimental evidence for the delivery of functional miR142-3p inhibitors specifically to CD4⁺ T cells *in vivo* we used a fluorescently-labeled miR142-3p inhibitor. Importantly, the successful delivery of the inhibitor to CD4⁺ T cells in relevant draining lymph nodes including liver-draining lymph nodes, mesenteric lymph nodes as well as pancreatic lymph nodes and directly in pancreas-residing CD4⁺ T cells was shown after 4 hours (Supplementary Figure 10a) and 24 hours (Supplementary Figure 10b). Previous miRNA inhibition experiments *in vivo* were assessed with a miRNA inhibitor application for 14 days every other day. Therefore, the successful delivery of the inhibitor to CD4⁺ T cells in lymph nodes was likewise analyzed at the end of the 14-days application period (Supplementary Figure 10c and d). In addition, the efficacy of the successfully delivered inhibitors was demonstrated by showing increased expression of Tgfr1, a well-established target of miR142-3p, directly in pancreatic T cells (Supplementary Figure 10e and f).

Response of reviewer 2 for revision 2: These experiments show the association of a FAM-labeled LNA inhibitor with T cells. What is needed is an activity reporter in T cells in vivo. The presence of the LNA attached to or inside cells does not provide clear evidence for inhibition of miRNA activity. As stated above, neither does measuring the abundance of a single (or small number of) miRNA target genes.

Response of authors for revision 3: We thank the reviewer for this comment and would like to highlight the extensive set of novel experiments we had provided in revision 1 and 2 to strengthen the successful inhibitor delivery *in vitro* and *in vivo* (Supplementary Figure 4a, b (added in revision 2), c and d (added in revision 1); Supplementary Figure 10 d-f (added in revision 2); Supplementary Figure 12a-d (added in revision 2), Supplementary Figure 14a-c (added in revision 2) and Figure 1 for the reviewers, added in revision 2). With the above mentioned set of experiments using the FAM-labeled LNA inhibitor, we demonstrated the successful *in vivo*

delivery of the miR142-3p inhibitor to CD4⁺ T cells in relevant draining lymph nodes including liver-draining lymph nodes, mesenteric lymph nodes as well as pancreatic lymph nodes and directly in pancreas-residing CD4⁺ T cells (Supplementary Figure 12a-d, added in revision 2). In addition, we clearly showed that the miR142-3p inhibitor blocks miR142-3p activity *in vitro* using various approaches, including the suggested miR142-3p activity sensor (described in the results section on page 7 and in the methods section on page 32 of the current manuscript, the corresponding results are shown in Supplementary Figure 4b, added in revision 2) and the increased abundance of two validated miR142-3p target genes (Supplementary Figure 4c, d, added in revision 1). These results, in addition to the confirmed localization of the inhibitor to CD4⁺ T cells *in vivo* and the likewise increased expression of a miR142-3p target gene *in vivo* (Supplementary Figure 12e, f), provides compelling evidence for the functionality of the miR142-3p inhibitor both *in vitro* and *in vivo*.

Response of authors from revision 2: In order to provide further evidence for the miR142-3p targeting of Tet2, we performed Treg induction experiments using naive CD4⁺ T cells from BALB/c mice in the presence of the miR142-3p inhibitor or control inhibitor and analyzed Tet2 mRNA and protein expression. Inhibition of miR142-3p resulted in significantly higher Tet2 mRNA levels after 6 hours of TCR stimulation (Figure 6b), and increased Tet2 protein abundance after 12 hours of TCR stimulation (Figure 6d). Importantly, there was no effect of miR142-3p inhibition on Tet1 or Tet3 expression (Supplementary Figure 7b).

To support the notion that miR142-3p directly targets the Tet2 mRNA, we co-transfected HEK-293 cells with a TET2 3'UTR reporter construct and a miR142-3p mimic and observed a significant reduction in luciferase activity compared to the transfection control (Supplementary Figure 3b).

Response from Reviewer 2 for revision 2: This experiment is referenced above. Was it performed in HEK-293 or Jurkat? Do both cell types express miR-142? The result is consistent with miR-142 inhibition of this plasmid. Vector

controls should be provided, and the presumed binding site in the 3'UTR should be mutated to disrupt this activity.

Response of authors for revision 3: We thank the reviewer for this comment and apologize for any potential unclear description of this experiment. The experiment is described in the results section on pages 11 and 12 as well as in the methods section on pages 31 and 32 of the current manuscript and the corresponding results are shown in Figure 1 for the reviewer/editors, Figure 5e and Supplementary Figure 10a. Additional details are provided below:

Results: “Moreover, a miR142-3p mimic induced a significant reduction in luciferase activity in HEK-293 cells transfected with a wildtype TET2 3'UTR reporter construct while there was no effect of the mimic in cells transfected with a reporter construct containing the TET2 3'UTR with mutated miR142-3p binding sites (Fig. 5e, Supplementary Fig. 10a).” Pages 11 and 12 of the manuscript.

Methods: “HEK-293 cells were co-transfected with a dual-luciferase plasmid containing wild-type or mutated full-length 3'UTR from human TET2 (RefSeq NM_001127208.2) and a miR142-3p mimic (10 pmol/well) at 10,000 cells per well in a 96-well plate using Lipofectamine 3000 (Thermo Fisher Scientific) for 24 h. Luminescence was measured with the Dual Luciferase Reporter Assay Kit (Promega) following the manufacturer’s protocol. (...)” Pages 31 and 32 of the manuscript.

As described above, the Tet2 3'UTR luciferase reporter experiment was performed in HEK-293 cells because these cells do not express miR142-3p, enabling a very precise analysis of the effect of the miR142-3p mimic. As requested by the editor and the reviewer, for this third round of revision we now performed additional experiments with the Tet2 3'UTR luciferase reporter. Specifically, we used site directed mutagenesis by PCR to mutate the potential miR142-3p binding sites in the Tet2 3'UTR. We first mutated the potential binding site identified by the chimeric reads of the HITS-CLIP data (mut 1) and second two other binding sites in the Tet2 3'UTR (position 4135-4141 and 5392-5398; mut 2), which were predicted using

Targetscan (Agarwal 2015). As before, we co-transfected HEK-293 cells with the respective luciferase reporter construct and a miR142-3p mimic. In mut 1 the mimic showed an effect comparable to the wildtype construct. Importantly, this effect was abrogated using mut 2, indicating that the binding sites at positions 4135-4141 and 5392-5398 are the ones targeted by miR142-3p and providing further compelling evidence of the direct targeting of Tet2 by miR142-3p.

Response of authors for revision 2: Furthermore, to support the direct miR142-3p – Tet2 target relationship, we employed two loss-of-function models, 3T3 fibroblasts and miR142 knockout (miR142^{-/-}) mice. As virtually all non-hematopoietic cells the 3T3 fibroblast cell line lacks miR142-3p expression almost completely (Supplementary Figure 8a), providing a suitable experimental system to study the effect of miR142-3p in absence of endogenous miR142-3p expression. Here, the introduction of miR142-3p in these cells via transfection with a miR142-3p mimic resulted in significantly decreased Tet2 mRNA levels (Figure 6f).

Next, we used T cells from miR142 deficient animals to validate the direct targeting of Tet2 by miR142-3p directly in a mouse model. The stimulation of CD4⁺ T cells from miR142^{-/-} mice resulted in significantly increased Tet2 mRNA levels when compared to T cells from miR142^{+/+} mice (Figure 6g). In line with the increased expression of Tet2 mRNA, Tet2 protein levels were likewise elevated in T cells from miR142 deficient mice following TCR stimulation when compared to T cells from mice expressing miR142 (Figure 6h).

As described above, stimulation of BALB/c T cells in presence of a miR142-3p inhibitor resulted in increased expression of Tet2 (Figure 6b) when compared to a control inhibitor. To confirm that these changes are due to a direct effect of miR142-3p inhibition and that Tet2 is a direct target of miR142-3p, we performed corresponding experiments with T cells from miR142^{-/-} mice which indicated the absence of any difference between T cells stimulated in presence of a miR142-3p inhibitor and a control inhibitor (Supplementary Figure 8c).

In line with a direct targeting of Tet2 by miR142-3p its inhibition *in vivo* resulted in significantly elevated Tet2 levels in pancreatic T cells of IAA⁺ NOD mice (Figure 8f, g). In addition, this effect was accompanied by an elevated Tet2 expression (Supplementary Figure 11d) in peripheral T cells of miR142-3p inhibitor treated IAA⁺ NOD mice.

Finally, the application of a miR142-3p inhibitor to humanized NSG mice showed a significant increase in Tet2 expression in pancreatic T cells (Supplementary Figure 13b, c).

Taken together the above-mentioned *in vitro* and *in vivo* experiments provide compelling evidence for direct targeting of Tet2 by miR142-3p.

Response of reviewer 2 for revision 2: It would be helpful if the authors could clarify which, if any, of the above is new to this revision. The manuscript was reviewed in full before, and all of the provided evidence was taken into account. The data show that miR-142 alters expression of miR-142, and they are consistent with the possibility that miR-142 directly targets Tet2, but they do not conclusively show that by any of the generally expected means. CLIP data should be excluded if they cannot be validated any further than what is shown. Overall, the manuscript focuses in on one potential direct target with inadequate evidence for a functional interaction.

Response of authors for revision 3: We thank the reviewer for this comment and apologize for any lack of clarity.

The application of the miR142-3p mimic to 3T3 fibroblasts was added in revision 1, the results can be found on page 12 of the manuscript and in Figure 5f.

The miR142^{-/-} mice experiments, showing increased Tet2 mRNA and protein abundance following TCR stimulation, compared to miR142^{+/+} mice were newly added in revision 2, the results can be found on page 12 of the manuscript and in Figure 5 g and h.

The experiments showing no effect of the miR142-3p inhibitor on Tet2, Tgfb1 or ATG16L1 expression in miR142^{-/-} mice were added in revision 2, the results can be found on pages 12 and 13 of the manuscript and in Supplementary Figure 10 d-f.

The *in vivo* inhibition of miR142-3p in IAA⁺ NOD mice was part of the initially submitted manuscript (pages 15 and 16 of the manuscript, Figure 7, Supplementary Figure 13), the application of a miR142-3p inhibitor to humanized NSG mice was added in revision 1 (pages 16 and 17 of the manuscript, Supplementary Figure 15).

As suggested by the reviewer and the editors, in revision 3 we now removed the chimeric reads data and relocated the remaining part of Figure 5 to the Supplementary Figures (Supplementary Figure 7). In addition, in revision 3 we now removed Supplementary Figure 6.

In summary and building up on an extensive number of newly added data in revisions 1-3, we provided an extensive set of experiments showing compelling evidence for the miR142-3p – Tet2 target relationship. The functionality of the miR142-3p inhibitor was shown by multiple *in vitro* and *in vivo* experiments, including a miR142-3p activity sensor in T cells (results: page 7, methods: page 31 Supplementary Figure 4b, added in revision 2), as suggested by the reviewer. We showed the effect of miR142-3p inhibitor and mimic application on Tet2 mRNA and protein abundance in various experimental settings *in vitro* (Figure 5b, d, f-h, added in revision 1, Supplementary Figure 10d, added in revision 2), and *in vivo* (Figure 7f, g, added in revision 1, Supplementary Figures 13d and 15b, c, added in revision 1), including knockout models and validated the direct targeting of Tet2 by miR142-3p using a Tet2 3'UTR luciferase reporter system. For this experiment we now provide additional controls using mutated Tet2 binding sites, as suggested by the reviewer and the editors (Figure 1 for the editors/reviewers, Supplementary Figure 10a, pages 11 and 12 of the manuscript).

The updated manuscript still lacks evidence for delivery of functional antagomirs *in vivo* and even *in vitro*. If this reagent is working as proposed, very efficient delivery must be achieved, since miRNA miR-142 is the most abundant miRNA in T cells.

Measurement of two known targets is not sufficient, as these genes may be affected by the treatment through indirect pathways. I am aware that this method for antagomir delivery to T cells was used in a prior publication, but that reference also lacks evidence for direct effects on specific miRNA activity. An activity sensor should be used to determine what fraction of miR-142 activity in T cells is relieved by delivery of the miR-142 specific antagomir compared with controls. Genetic deletion of miR-142 can be achieved with existing mutant mice. They would provide a better means of loss of function analysis.

Response of authors for revision 2: We thank the reviewer for this important and constructive comment and performed additional novel experiments to further validate the delivery of functional inhibitors *in vitro* and *in vivo* as requested.

miR142-3p inhibitor delivery *in vitro*

The successful chitosan-coated PLGA nanoparticle-mediated miRNA uptake in CD4⁺ T cells including intracellular co-localization of the nanoparticles and the fluorescently-labeled miRNA has been shown in Serr et al. 2018¹⁵. Naive CD4⁺ T cells were stimulated with anti-CD3 and anti-CD28 antibodies for 18 hours in the presence of FA-labeled nanoparticles and/or a fluorescently-labeled miRNA. As shown in a previous study (Serr et al. 2018¹⁵), FA-labeled nanoparticles and the miRNA were efficiently taken up by the cells (Fig.S5 A and B) and confocal microscopy images show the intracellular co-localization of nanoparticles and miRNA (Fig.S5 C).”

Response of reviewer 2 for revision 2: As mentioned above, these data do not show functional inhibition of miR-142. Several other papers have shown this degree of evidence for delivery of various nucleic acids, including miRNA antagonists of several different chemistries, to T cells and other cells. What is lacking is evidence that this actually results in a reduction in miRNA activity in the cells.

Response of authors for revision 3: We thank the reviewer for this comment and would also like to use this comment to respectfully disagree. As described above and in the reviewer's comments in revision 2 the reviewer had suggested using an activity sensor to test the activity of miR142-3p in T cells. According to his suggestions we have performed exactly these experiments in Jurkat T cells in revision 2. The experiment is described in the results section on page 7 and in the methods section on page 32 of the current manuscript and the corresponding results are shown in Supplementary Figure 4b. The respective sections of the manuscript are provided below:

Results: *“Second, application of the miRNA inhibitor significantly de-repressed luciferase activity of a miR142-3p sensor plasmid in Jurkat T cells (Supplementary Fig. 4b).”* Page 7 of the manuscript.

Methods: *“A miR142-3p activity sensor plasmid was constructed by inserting a double-stranded oligonucleotide containing the miR142-3p target sequences (miR142-3p reverse complement, with a central bulged mismatch) into the 3'-UTR of a dual luciferase reporter plasmid.”* Page 32 of the manuscript.

We would like to emphasize that the luciferase reporter plasmid used in this activity sensor experiment does not contain the Tet2 3'UTR. We delivered, by plasmid transfection, a luciferase reporter gene with an artificial 3'UTR containing the perfect complementary sequence to miR142-3p, as suggested by the reviewer in this revision and revision 2. We used this sensor plasmid in Jurkat T cells to validate the functional blocking of miR142-3p by the inhibitor directly in T cells.

[REDACTED]

In addition, to demonstrate the efficacy of the miR142-3p inhibitor in T cells *in vitro*, we now performed additional experiments with a miR142-3p activity sensor as requested. The miR142-3p activity sensor plasmid was constructed by inserting a double-stranded oligonucleotide containing the miR142-3p target sequences (miR142-3p reverse complement, with a central bulged mismatch) into the 3'-UTR of a dual luciferase reporter plasmid. The co-transfection of Jurkat T cells with the miR142-3p sensor plasmid and the miR142-3p inhibitor significantly de-repressed luciferase expression (Figure 2 for the reviewers/ Supplementary Figure 3b in the revised manuscript).

Figure 2 for the reviewers. miR142-3p inhibitor efficiently blocks its target miRNA. Normalized luciferase activity of Jurkat T cells co-transfected with a miR142-3p inhibitor and a miR142-3p activity sensor plasmid containing the miR142-3p target sequences in the 3'-UTR. $n = 4$.

Response of reviewer 2 to revision 2: See comments above regarding miRNA activity sensors. Using all (or part?) of the 3'UTR of Tet2, which is incompletely validated as a miR-142 sites, doesn't make sense, and can't

possibly provide the dynamic range of a normal miRNA activity sensor. Also as described above, experiments of this type require controls not provided here.

Response of authors for revision 3: We thank the reviewer for this comment and would also like to use this comment to respectfully disagree. As described above and in the reviewer's comments in revision 2 the reviewer had suggested using an activity sensor to test the activity of miR142-3p in T cells. According to his suggestions we have performed exactly these experiments in Jurkat T cells in revision 2. The experiment is described in the results section on page 7 and in the methods section on page 32 of the current manuscript and the corresponding results are shown in Supplementary Figure 4b. The respective sections of the manuscript are provided below:

Results: "Second, application of the miRNA inhibitor significantly de-repressed luciferase activity of a miR142-3p sensor plasmid in Jurkat T cells (Supplementary Fig. 4b)." Page 7 of the manuscript.

Methods: "A miR142-3p activity sensor plasmid was constructed by inserting a double-stranded oligonucleotide containing the miR142-3p target sequences (miR142-3p reverse complement, with a central bulged mismatch) into the 3'-UTR of a dual luciferase reporter plasmid." Page 32 of the manuscript.

We would like to emphasize that the luciferase reporter plasmid used in this activity sensor experiment does not contain the Tet2 3'UTR. We delivered, by plasmid transfection, a luciferase reporter gene with an artificial 3'UTR containing the perfect complementary sequence to miR142-3p, as suggested by the reviewer in this revision and revision 2. We used this sensor plasmid in Jurkat T cells to validate the functional blocking of miR142-3p by the inhibitor directly in T cells.

Taken together, we show the successful miRNA uptake in CD4⁺ T cells including intracellular co-localization of the nanoparticles and the fluorescently-labeled miRNA (Figure 1 for the reviewers). Moreover, we now demonstrate reduced miR142-3p

activity in T cells by delivery of the miR142-3p inhibitor compared with a control inhibitor using experiments with a miR142-3p activity sensor as suggested (Figure 2 for the reviewers / Supplementary Figure 3b in the revised manuscript).

miR142-3p inhibitor delivery *in vivo*

For the inhibition of miR142-3p *in vivo* we used an LNA *in vivo* miRNA inhibitor which has been shown to accumulate in a broad range of tissues and to efficiently inhibit its targets in several independent studies^{13,14}. Specifically, important studies using LNA miRNA inhibitors to successfully inhibit miRNAs *in vivo*, among others in T cells, are listed in Table 1 and an even more extensive list of studies can be found here: <http://www.exiqon.com/Is/Documents/Scientific/mirna-inhibition-publications.pdf>.

Table 1. *In vivo* miRNA inhibition using LNA miRNA inhibitors. List of selected publication.

microRNA	Tissue	Organism	Process/Disease	Reference
miR-221	Various	Mouse (plus toxicology studies in non-human primates)	Multiple Myeloma	Gallo Cantafio et al. , Mol. Ther. Nucl. Acids, 2016
miR-802	Liver	Mouse	Diabetes	Kornfeld et al. , Nature 2013
miR-33	Liver	Mouse	Cholesterol regulation	Najafi-Shoushtari et al. , Science 2010
miR-142-3p	T-cells	Mouse	Graft-versus-host disease	Sun et al. , JCI 2015
miR-192	Kidney	Mouse	Diabetic Nephropathy	Putta et al. , J Am Soc Nephrol 2012
miR-134	Brain	Mouse	Epilepsy	Jimenez-Mateos et al. , 2012, Nature Medicine
miR-212	Brain	Rat	Addiction	Hollander et al. , Nature 2010
miR-21	Lung	Mouse	Fibrosis	Liu et al. , J Exp Med 2010
miR-34a	Heart	Mouse	Cardiac decline/ myocardial infarction	Boon et al. , Nature 2013
miR-21-5p	Heart	Mouse	Obesity	Seeger et al. , Obesity 2014
miR-199a, miR-1908	Melanoma	Mouse	Melanoma metastasis	Pencheva et al. , Cell 2012
miR-712	Aorta endothelium	Mouse	Atherosclerosis	Son et al. , Nature Commun 2013
miR-33	Eye	Mouse	Age related macular degeneration	Sene et al. , Cell Metabolism 2013
miR-21-5p	Bone marrow – hemapoietic stem cells	Mouse	Myelodysplastic	Bhagat et al. , Blood 2013

When delivered systemically, a series of studies have documented that LNA miRNA inhibitors distribute broadly into most tissues, including hematopoietic tissues such

as lymph nodes, spleen and bone marrow^{13,14}. The results of two studies show the accumulation of miRNA inhibitors in various tissues (Figure 2, Straarup et al. 2010 14; Figure 4 Cantafio et al. 2016 13).

[REDACTED]

[REDACTED]

We now performed additional novel experiments using a fluorescently-labeled miR142-3p inhibitor in order to provide further experimental evidence for the delivery of functional miR142-3p inhibitors specifically to CD4⁺ T cells *in vivo*. Importantly, we confirmed the successful delivery of the inhibitor to CD4⁺ T cells in relevant draining lymph nodes including liver-draining lymph nodes, mesenteric lymph nodes as well as pancreatic lymph nodes and directly in pancreas-residing CD4⁺ T cells as assessed after 4 hours (Figure 5a for the reviewers / Supplementary Figure 10a in the revised manuscript) and 24 hours (Figure 5b for the reviewers / Supplementary Figure 10b in the revised manuscript).

Figure 5 for the reviewers. miR142-3p inhibitor accumulates in CD4⁺ T cells *in vivo*. (a) Representative histograms showing the accumulation of the FAM-labelled miR142-3p inhibitor in CD4⁺ T cells isolated from relevant lymph nodes and the pancreas 4 hours after inhibitor application. (b) Representative histograms showing the accumulation of the FAM-labelled miR142-3p inhibitor in CD4⁺ T cells isolated from relevant lymph nodes and the pancreas 24 hours after inhibitor application. (c) Representative histograms showing the accumulation of the FAM-labelled miR142-3p inhibitor in CD4⁺ T cells isolated from lymph nodes after 14 days of inhibitor application every other day. (d) miR142-3p inhibitor abundance (median fluorescence intensity) in CD4⁺ T cells isolated from lymph nodes after 14 days of inhibitor application every other day. *n* = 4.

In addition, we likewise demonstrated the successful delivery of the inhibitor to CD4⁺ T cells in lymph nodes at the end of the 14-days application period (Figure 5c and for the reviewers / Supplementary Figure 10c and d in the revised manuscript). Moreover, we validated the efficiency of the delivered miR142-3p inhibitor in pancreatic tissue and local immune cells by showing increased expression of *Tgfb1*, a well-established target of miR142-3p, in pancreatic T cells (Supplementary Figure 10e and f in the revised manuscript).

Response of reviewer 2 to revision 2: I am very aware of these prior publications, and my laboratory has used these same inhibitors in our own studies. Only a few of these publications study T cells as a target cell type. As stated above, none of them provide evidence for functional delivery to T cells *in vivo*. They should have! I cannot explain why they were published without providing this critical information.

Response of authors for revision 3: We thank the reviewer for this comment and would like to highlight the extensive set of novel experiments we had provided in revision 1 and 2 to strengthen the successful inhibitor delivery *in vitro* and *in vivo* (Supplementary Figure 4a, b (added in revision 2), c and d (added in revision 1); Supplementary Figure 10 d-f (added in revision 2); Supplementary Figure 12a-d

(added in revision 2), Supplementary Figure 14a-c (added in revision 2) and Figure 1 for the reviewers, added in revision 2). With the above mentioned set of experiments using the FAM-labeled LNA inhibitor, we demonstrated the successful *in vivo* delivery of the miR142-3p inhibitor to CD4⁺ T cells in relevant draining lymph nodes including liver-draining lymph nodes, mesenteric lymph nodes as well as pancreatic lymph nodes and directly in pancreas-residing CD4⁺ T cells (Supplementary Figure 12a-d, added in revision 2). In addition, we clearly showed that the miR142-3p inhibitor blocks miR142-3p activity *in vitro* using various approaches, including the suggested miR142-3p activity sensor (described in the results section on page 7 and in the methods section on page 32 of the current manuscript, the corresponding results are shown in Supplementary Figure 4b, added in revision 2) and the increased abundance of two validated miR142-3p target genes (Supplementary Figure 4c, d, added in revision 1). These results, in addition to the confirmed localization of the inhibitor to CD4⁺ T cells *in vivo* and the likewise increased expression of a miR142-3p target gene *in vivo* (Supplementary Figure 12e, f), provides compelling evidence for the functionality of the miR142-3p inhibitor both *in vitro* and *in vivo*.

Response of authors for revision 2:

Direct effect of the miR142-3p inhibitor *in vitro* and *in vivo*

We now also performed additional *in vivo* experiments using miR142 deficient animals to further validate the specific effect of the miR142-3p inhibitor *in vitro* and *in vivo*. In a previous set of experiments outlined in the manuscript we have demonstrated increased expression of Tet2 (Figure 6 in the revised manuscript) and two well-established miR142-3p targets, Tgfbr1 and ATG16L1 (Supplementary Figure 3 in the revised manuscript), in BALB/c T cells stimulated in presence of a miR142-3p inhibitor compared to a control inhibitor. Now, we performed corresponding experiments with T cells from miR142^{-/-} mice which indicated the absence of any difference between T cells stimulated in presence of a miR142-3p inhibitor and a control inhibitor (Figure 6a-c for the reviewers / Supplementary Figure 8c-e in the revised manuscript). The findings in the miR142 knockout animals again

underline the direct effect of the miR142-3p inhibitor and provide compelling evidence that Tet2 is a direct target of miR142-3p.

Figure 6 for the reviewers. miR142-3p inhibitor has no effect in miR142^{-/-} CD4⁺ T cells *in vitro*. (a) Tet2 mRNA abundance after 6 hours of TCR stimulation of CD4⁺ T cells isolated from lymph nodes of miR142^{-/-} mice, in presence of a miR142-3p inhibitor and a control inhibitor respectively. *n* = 4. (b) Tgfb1 mRNA abundance after 6 hours of TCR stimulation of CD4⁺ T cells isolated from lymph nodes of miR142^{-/-} mice, in presence of a miR142-3p inhibitor and a control inhibitor respectively. *n* = 4. (c) ATG16L1 mRNA abundance after 6 hours of TCR stimulation of CD4⁺ T cells isolated from lymph nodes of miR142^{-/-} mice, in presence of a miR142-3p inhibitor and a control inhibitor respectively. *n* = 4.

In addition and to confirm that the observed effects of miR142-3p inhibitor application on Treg induction, stability and Tet2 in NOD mice *in vivo* were directly mediated by reduced miR142-3p activity, we now performed novel *in vivo* experiments and applied the miR142-3p inhibitor to miR142^{-/-} mice. As expected, the inhibitor had no effect on Treg frequency (Figure 7a for the reviewers / Supplementary Figure 12a in the revised manuscript), Tet2 protein abundance (Figure 7b for the reviewers / Supplementary Figure 12b in the revised manuscript) or Foxp3 CNS2 methylation (Figure 7c for the reviewers / Supplementary Figure 12c in the revised manuscript) in miR142 deficient mice, confirming the delivery of functional miR142-3p inhibitors *in vivo* and Tet2 as a direct target of miR142-3p.

Figure 7 for the reviewers. *miR142-3p* inhibitor has no effect in *miR142*^{-/-} mice *in vivo*. (a) *Ex vivo* CD25^{hi}Foxp3⁺ Tregs isolated from lymph nodes of *miR142*^{-/-} mice treated with a *miR142-3p* inhibitor or control inhibitor for 14 days with 10 mg/kg ip every other day. *n* = 4. (b) *Ex vivo* Tet2 protein abundance (median fluorescence intensity) in CD4⁺ T cells isolated from lymph nodes of *miR142*^{-/-} mice treated with a *miR142-3p* inhibitor or control inhibitor as described in (a). *n* = 4. (c) Methylation of the Fxp3 CNS2 (mean of all sites) in Tregs isolated from lymph nodes of *miR142*^{-/-} mice treated with a *miR142-3p* inhibitor or control inhibitor as described in (a). *n* = 4.

Response of reviewer 2 to revision 2: These additional data are consistent with the idea that the inhibitors's *in vivo* effects occur due to targeting of *miR-142* in some cell type(s). Showing direct effects in T cells remains an important validation.

Response of authors for revision 3: We thank the reviewer for this comment and would like to respectfully highlight the extensive set of experiments we performed to show the effects of *miR142-3p* inhibition in T cells both *in vitro* and *in vivo* which have been newly added in response to revisions 1-3. The functionality of the *miR142-3p* inhibitor and the direct targeting of Tet2 by *miR142-3p* were shown by *in vitro* luciferase assays (*miR142-3p* activity sensor: page 7 of the manuscript, Supplementary Figure 4b; Tet2 3'UTR luciferase assay: pages 11 and 12 of the manuscript, Figure 5e, Supplementary Figure 10a). Using FAM-labeled LNA *in vivo*

inhibitors, we demonstrated the successful *in vivo* delivery of the miR142-3p inhibitor to CD4⁺ T cells in relevant draining lymph nodes including liver-draining lymph nodes, mesenteric lymph nodes as well as pancreatic lymph nodes and directly in pancreas-residing CD4⁺ T cells (page 15 of the manuscript, Figure 12a-d). Here we wish to respectfully highlight that all the above mentioned analyses (Figure 6 for the reviewers and Figure 7 for the reviewers of revision 2, added in revision 2) were all performed in CD4⁺ T cells and the same applies for the *in vivo* experiments. Therefore we would like to highlight that these experiments provide compelling evidence for a direct effect of the inhibitor in T cells.

Response of authors for revision 2: These results clearly indicate that the effect observed *in vitro* and *in vivo* are due to the specific inhibition of miR142-3p by means of the miR142-3p inhibitor. Altogether we provide compelling evidence for a miR142-3p/Tet2/Foxp3 axis in murine and human CD4⁺ T cells that during islet autoimmunity interferes with the efficient induction of Tregs and leads to impairments in Treg stability. Importantly, we strengthen the claims that Tet2 is a direct target of miR142-3p, and that inhibitor delivery *in vivo* and *in vitro* is sufficiently robust to account for the observed effects.

Response of reviewer 2 to revision 2: I respectfully disagree. The authors are urged to perform the recommended rigorous and simple (though admittedly time- and resource-consuming) experiments to test whether miR-142 antagonists can really block miR-142 activity in T cells in vivo. If they do not, the authors are further urged to consider alternate explanations for their observations of altered biology in these experiments. Other equally interesting and plausible possibilities exist.

Response of authors for revision 3: As described above in more detail, we provided an extensive set of novel experiments to validate the effect of the miR142-3p inhibitor. These additional data have been included in response to revision 1-3. We confirmed its functionality *in vitro* using a miR142-3p activity sensor as requested by the reviewer (page 7 of the manuscript, Supplementary Figure 4b). We showed that the miR142-3p inhibitor increased the abundance of validated miR142-3p

targets *in vitro* (page 7 of the manuscript, Supplementary Figure 4c and d) and *in vivo* (page 15 of the manuscript, Supplementary Figure 12e, f) and the absence of this effect in miR142^{-/-} T cells *in vitro* (pages 12 and 13 of the manuscript, Supplementary Figure 10d-f). We confirmed the successful delivery of the same inhibitor to CD4⁺ T cells in relevant draining lymph nodes including liver-draining lymph nodes, mesenteric lymph nodes as well as pancreatic lymph nodes and directly in pancreas-residing CD4⁺ T cells (page 15 of the manuscript, Supplementary Figure 12a-d and the *in vivo* effects resembled the effects observed *in vitro*.

REVIEWERS' COMMENTS:

Reviewer #2 (Remarks to the Author):

My prior concerns about this manuscript were focused on two major issues: The evidence for miR-142 targeting of Tet2, and the evidence that in vivo treatments act through miR-142 in T cells.

The first of these concerns has been adequately addressed by removal of the HITS-CLIP crosslinking data that seemed to suggest miR-142 interaction with a non-canonical binding site. New luciferase assays added in this revision indicate that two different canonical predicted miR-142 binding sites can mediate miR-142 regulation of the Tet2 3'UTR. I applaud the authors persistence in investigating how miR-142 affects Tet2 expression.

On the second point, the authors and I will have to agree to disagree. I eagerly await a clear demonstration that miRNA inhibitors can be effectively delivered to T cells and alter miRNA activity in vivo. However, I don't dispute that the reported treatments had the described effects in vivo, be it through inhibition of miRNA activity in T cells or other cells, or through some other unexpected mechanism.

Overall, the revised manuscript will be of significant value to the scientific community.

K. Mark Ansel

Point by Point Response

Nature Communications Manuscript number NCOMMS-18-24478C

miRNA142-3p targets Tet2 and impairs Treg differentiation and stability in models of type 1 diabetes

REVIEWERS' COMMENTS:

Reviewer #2 (Remarks to the Author):

My prior concerns about this manuscript were focused on two major issues: The evidence for miR-142 targeting of Tet2, and the evidence that in vivo treatments act through miR-142 in T cells.

Response: We thank the reviewer for his comments and constructive criticism during the previous rounds of revision, which were very helpful to improve the quality of the manuscript. We are pleased that we have addressed his concerns sufficiently as outlined below.

The first of these concerns has been adequately addressed by removal of the HITS-CLIP crosslinking data that seemed to suggest miR-142 interaction with a non-canonical binding site. New luciferase assays added in this revision indicate that two different canonical predicted miR-142 binding sites can mediate miR-142 regulation of the Tet2 3'UTR. I applaud the authors persistence in investigating how miR-142 affects Tet2 expression.

Response: We thank the reviewer for his comment and are delighted that we adequately addressed his concerns regarding the miR142-3p – Tet2 – target relationship. As detailed in revision #3 we removed the chimeric reads part of the HITS-CLIP analysis and now focused on molecular and cellular approaches to confirm Tet2 as a direct target of miR142-3p. We are happy that the reviewer agrees that the newly added luciferase assay data convincingly confirm Tet2 as a direct target of miR142-3p in line with the two predicted binding sites mediating regulation of Tet2 by miR142-3p.

On the second point, the authors and I will have to agree to disagree. I eagerly await a clear demonstration that miRNA inhibitors can be effectively delivered to T cells and alter miRNA activity in vivo. However, I don't dispute that the reported treatments had the described effects in vivo, be it through inhibition of miRNA activity in T cells or other cells, or through some other unexpected mechanism.

Response: We thank the reviewer for this comment and are glad that we agree on the in vivo effects of the miR142-3p inhibitor, described in our study. Regarding the delivery and efficacy, we would like to refer to the extensive set of experiments we had provided in revisions 1 and 2 in order to strengthen the successful inhibitor delivery in vitro and in vivo. Using the FAM-labeled LNA inhibitor, we demonstrated the successful in vivo delivery of the miR142-3p inhibitor to CD4⁺ T cells in relevant draining lymph nodes including liver-draining lymph nodes, mesenteric lymph nodes as well as pancreatic lymph nodes and directly in pancreas-residing CD4⁺ T cells. In addition, we clearly showed that the miR142-3p inhibitor blocks miR142-3p activity in vitro using various experimental approaches, including the suggested miR142-3p activity sensor and the increased abundance of two validated miR142-3p target genes. These results, in addition to the confirmed localization of the inhibitor to CD4⁺ T cells in vivo and the likewise increased expression of a miR142-3p target gene in vivo, provides compelling evidence for the functionality of the miR142-3p inhibitor both in vitro and in vivo.

Overall, the revised manuscript will be of significant value to the scientific community.

K. Mark Ansel

Response: We thank the reviewer for his comment and are delighted that the reviewer finds our revised manuscript of significant value to the scientific community and sufficient scope and quality to be published at Nature Communications.